

# An adaptive two-stage analog/regression model for probabilistic prediction of local precipitation in France.

Jérémy Chardon[1], Benoit Hingray[1], and Anne-Catherine Favre[1]

[1]Univ. Grenoble Alpes, CNRS, IRD, Grenoble INP, IGE, F-38000 Grenoble, France

*Correspondence to:* Benoit Hingray (benoit.hingray@univ-grenoble-alpes.fr)

**Abstract.** Statistical Downscaling Methods (SDMs) are often used to produce local weather scenarios from large scale atmospheric information. SDMs include transfer functions which are based on a statistical link identified from observations between local weather and a set of large scale predictors. As physical processes generating surface weather vary in time, the most relevant predictors and the regression link are likely to also vary in time. This is well known for precipitation for instance and the link is thus often estimated after some seasonal stratification of the data. In this study, we present a hybrid model where the regression link is estimated from atmospheric analogs of the current prediction day. Atmospheric analogs are first identified from geopotential fields at 1000 and 500 hPa. For the regression stage, two Generalized Linear Models are further used to model the probability of precipitation occurrence and the distribution of non-zero precipitation amounts respectively. The hybrid model is evaluated for the probabilistic prediction of local precipitation over France. It noticeably improves the skill of the prediction for both precipitation occurrence and quantity. As the analog days vary from one prediction day to another, the atmospheric predictors selected in the regression stage and the value of the corresponding regression coefficients vary from one prediction day to another. The hybrid approach allows thus for a day-to-day adaptive and tailored downscaling. It can also reveal specific predictors for peculiar and non-frequent weather configurations.

## 1 Introduction

Global meteorological reanalyses of the Earth system for the whole 20[th] century (e.g. Compo et al., 2011), numerical weather forecasting models and general circulation models (GCM) provide useful information on weather at synoptic scale for historical, near future and far future contexts respectively. However, both their coarse spatial resolution and the bias often obtained for simulated weather variables at local scales make their simulation outputs of limited use for a number of regional studies without post-processing. Forced by GCM or reanalysis data as boundary conditions, dynamic downscaling models allow the simulation of climate conditions over a limited spatial domain with a higher resolution of typically a few dozen kilometers (e.g. Jacob et al., 2014). Despite a much better representation of land surface features and atmospheric processes, large biases are still obtained for a number of simulated weather variables, such as precipitation, as a result of the parametrizations required to simulate sub-grid atmospheric processes.

Statistical Downscaling Models (SDMs), including bias correction methods, have been developed to tackle these resolution and bias issues (see Maraun et al., 2010, for a review). Among the different SDM approaches presented over the last decades,





Perfect Prog (PP) approaches make use of the physical relationships that exist between some large scale atmospheric parameters and local weather variables. Local weather scenarios can then be produced for any prediction day conditionally on the large scale atmospheric configuration observed or simulated for this day. Time series scenarios of local weather can also be produced from time series of large scale atmospheric predictors in a way which is consistent with the dynamic of the atmosphere at large

scale.

PP SDMs are widely used to generate weather scenarios for past or future climates from outputs of climate models (e.g. Wilby et al., 1999; Hanssen-Bauer et al., 2005; Boé et al., 2007; Lafaysse et al., 2014). They can also be used to reconstruct past weather conditions from atmospheric reanalysis data (e.g. Auffray et al., 2011; Wilby and Quinn, 2013; Kuentz et al., 2015), to produce local weather forecasts from weather numerical models (e.g. Obled et al., 2002; Gangopadhyay et al., 2005;

Marty et al., 2013; Ben Daoud et al., 2016) or for detection/attribution of climate trends (e.g. Vautard and Yiou, 2009; Stott et al., 2016).

PP SDMs include transfer functions, weather-type based models and methods based on atmospheric analogs (see Maraun et al., 2010, for a review). In the latter approach, atmospheric analog days of the current prediction day are searched for on the basis of some atmospheric similarity criterion in the historical database. The weather variables observed for one or for a

selection of the $k$-most similar days are then used as a weather scenario for the current prediction day.

A major advantage of resampling approaches and thus analog based approaches is that they do not require restrictive assumptions on the distribution of the predictands. They can be thus easily applied for the generation of non-normally distributed data, precipitation especially, or for the generation of multisite multivariate scenarios (e.g. Dayon et al., 2015). Surface weather variables of different predictands can indeed be sampled simultaneously from historical records for a given analog day, thus in-

suring the physical realism and consistency of generated fields within each day (because already observed, e.g. Chardon et al., 2014; Raynaud et al., 2016). A frequently reported limitation however is that for any given simulation time step (e.g. any given day when analog days are looked for), analog based methods are unable to give scenarios which were not already observed. Extrapolation from what has been experienced in the past is therefore not possible, even when the current meteorological situation suggests and/or requires some extrapolated estimate.

Transfer functions do not suffer from this extrapolation limitation. Transfer functions are mainly regression models where the expected value of the predictand for time $t$ is expressed as a linear or non-linear function of a set of predictors. For precipitation, the regression can be achieved with Generalized Linear Models (GLMs) or Vector Generalized Linear Models (VGLMs) which allow to extent the linear regression to non-Gaussian data (e.g. Maraun et al., 2010).

Transfer functions have also been used to produce deterministic, stochastic or probabilistic predictions. In the stochastic

case, a random noise component is classically added to the expected value of the predictand (e.g. Mezghani and Hingray, 2009). The stochastic process allows not underestimating the local scale variability of the predictand, a well-known limitation of deterministic predictions. It also allows the generation of non-observed and even extreme weather scenarios in a way which is consistent with the statistical distribution applied to model the data. The downscaling relationship and its parametrization are classically established empirically between a selection of large-scale predictors and the predictand (e.g. precipitation occur-

rence) from a set of observations or pseudo observations available for recent decades. The relationship is however obviously





expected to slightly change from one large scale weather configuration to the other. When inferred from all observations available for a given period, the relationship – which is likely inferred from a heterogeneous ensemble of precipitation events – is thus expected to lack physical relevance and possibly also statistical robustness. To reduce this potential limitation, the parameterization is often estimated after some data stratification. In the classical calendar stratification, one parameter set is optimized

for each calendar month or season. The stratification can also be based on some weather type information. In this case, a set of parameters is classically estimated for each weather type of a given pre-established weather type classification (e.g. Enke and Spegat, 1997). Widely applied, this weather-type based approach is expected to allow for a more physically based downscaling. An obvious limitation however remains for prediction days that do not clearly belong to one specific weather type (e.g. prediction days that are close to the "weather frontiers" delimiting two or more weather types). Those days are indeed likely to

be rather dissimilar to the weather configurations that each weather type is expected to characterize, making the downscaling relationships to be used not suited anymore or, at least, sub-optimal.

A smoother weather type like approach could consist in defining the weather type from all atmospheric situations that are similar to the situation of the prediction day. The ensemble of days from which the downscaling link can be identified is thus expected to be statistically and physically homogeneous. This is in turn expected to make the link much more relevant

and robust and to improve the prediction. Such an approach can be actually achieved in hybridizing the two popular SDM approaches discussed above, based on atmospheric analogs and on transfer functions respectively. To our knowledge, this hybrid approach has been only explored in few previous studies. In Ribalaygua et al. (2013), it was found to improve the probabilistic prediction of local surface temperature in the Spanish Iberian Peninsula. The multiple linear regression uses for each prediction day forward and backward stepwise selection of predictors from a set of four potential predictors (thickness of

the air column, three temperature indexes of previous days). However for precipitation, the authors did not test the potential of the hybridization, building directly the predictions from the precipitation observations of the 30 most similar atmospheric analogs. In the deterministic approach presented by Ibarra-Berastegi et al. (2011), incorporating the regression stage (with 79 potential atmospheric predictors) was found to allow a clear though not overwhelming improvement over the classical analog based predictions. The regression stage used a multiple linear regression model which is not really adapted to the strong

asymmetry of the distribution of precipitation (leading to a non normal distribution) and which leaves likely room for prediction skill improvement.

In the present study, we explore the interest of a hybrid analog/regression downscaling model for the probabilistic prediction of daily precipitation over France: for each prediction day, the statistical downscaling link between some large scale atmospheric predictors and local precipitation is estimated from large scale and local scale observations available from an ensemble

of atmospheric days analog to the prediction day. The analog model (AM) used for the analog stage is based on developments from different studies initially focusing on the probabilistic quantitative precipitation forecasts in southern France (e.g. Bontron and Obled, 2005; Marty et al., 2012) and extended to the prediction of precipitation on larger spatial domains (e.g. Chardon et al., 2014). For the regression stage, we use a 2-part GLM approach where the probability of precipitation occurrence and the distribution of wet-days amounts are modeled separately following Furrer and Katz (2007) and Mezghani and Hingray (2009).





The paper structures as follows: Section 2 describes the data and the hybrid downscaling model. Section 4 presents the skill of the model for the prediction of both precipitation occurrence and amount. Results are discussed in section 5 and section 6 concludes.

## 2   Data

The predictand is the daily local precipitation estimated over 8,981 grid cells of $8 \times 8 \ \mathrm{km}^2$ covering the metropolitan French territory. Precipitation data are obtained from the SAFRAN analysis produced for several surface variables at hourly time step by MeteoFrance (Quintana-Segui et al., 2008; Vidal et al., 2010). They will be considered as pseudo-observations in the following.

Atmospheric predictors are taken from the European Centre for Medium-Range Weather Forecasts (ECMWF) Re-Analysis

(ERA-40, Uppala et al., 2005). This global meteorological re-analysis is available on a $1.125° \times 1.125°$ grid with a 6-hour temporal resolution.

For the analog stage, predictors are 1000 and 500 HGT geopotential fields over a large spatial domain (roughly Lat = $10°$, Lon = $8°$) centered on the target location. These predictors have been found to be the most informative large scale predictors to be used in this context for France (e.g. Guilbaud and Obled, 1998; Obled et al., 2002; Radanovics et al., 2013). They also

correspond to the best large scale predictors of daily precipitation for different regions in Europe with contrasted meteorological regimes (Raynaud et al., 2016).

For the regression stage, 13 more predictors were considered. The selection gathers most predictors considered in previous studies over Europe (e.g. Hanssen-Bauer et al., 2005; Wetterhall et al., 2009; Horton et al., 2012; Raynaud et al., 2016). They include predictors characterizing the thermal state of the atmosphere, its dynamics, the water atmosphere content, its

thermo-dynamical instability. As potential predictor, we also consider the occurrence of precipitation for the previous day. All predictors are here scalar variables. Atmospheric predictors are estimated on a daily time step (mean of the four values available at 6, 12, 18 and 24h00 UTC) from the four ERA-40 grid cells surrounding the prediction grid cell (inverse distance interpolation).

To avoid the multi-colinearity in the predictors for the regression, we identified a subset of uncorrelated predictors. The

cross-correlations between all predictor pairs were first estimated on an annual basis from all available data. The correlation structure may however differ from one atmospheric configuration to the other. The set of uncorrelated predictors could thus differ from one prediction day to the other. We thus repeated the correlation analysis for each prediction day, using for this estimation the predictor values observed for the 100 nearest atmospheric analogs identified for this day. The main features of the inter-variable correlations were found to be roughly independent on the day (not shown). The final subset of uncorrelated

predictors is listed in Table 1. These predictors are tested for the prediction of both precipitation occurrence and quantity.

Predictors considered for the analog and regression stages obviously inform about different features of the atmosphere state for different scales. Geopotential fields, by their spatial extent, characterize the large-scale atmospheric circulation configuration (the spatial domain of several thousands of kilometers, includes a part of North-Eastern Atlantic and covers France and





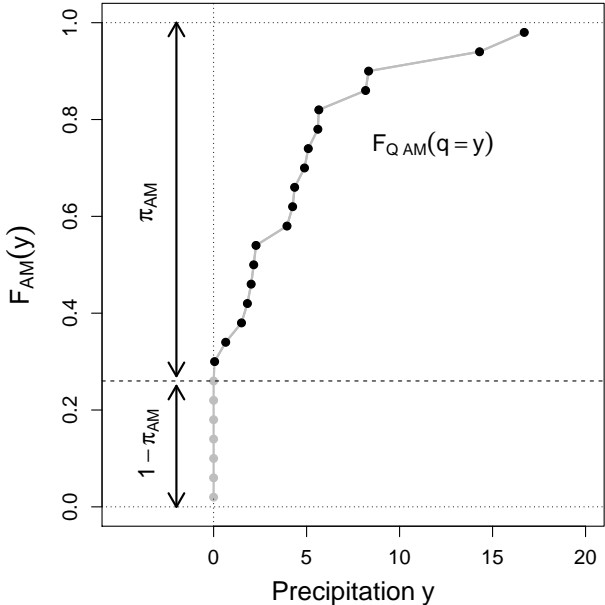

**Figure 1.** Empirical cumulative distribution function (in grey) of one day obtained from the AM for a given prediction day. The empirical cdf of the precipitation quantity $F_{Q,AM}$, weighted by the occurrence probability of precipitation $\pi_{AM}$ is highlighted in black (c.f. Eq. 1).

a part of neighboring countries) whereas GLM scalar predictors are descriptive of a more local (and mostly thermodynamic) state of the atmosphere (the spatial domain of several hundreds of kilometers is roughly centered above the target location).

## 3   The hybrid analog/regression model

As illustrated in Fig. 1, the cumulative distribution function (cdf) $F_Y$ of precipitation $Y$ can be expressed for any given day as

5    the composition of the no-precipitation occurrence probability $1 - \pi$ and the cdf $F_Q$ of the precipitation quantity $Q$ for non-zero precipitation:

$$F_Y(y) = (1 - \pi) + \pi \cdot F_Q(q = y), \tag{1}$$

where $\pi$ is the precipitation occurrence probability, $y$ and $q$ correspond to the precipitation value with regard to the whole precipitation distribution and to the non-zero precipitation distribution respectively.

10    In the present work, the cdf of precipitation is modeled for each prediction day with GLMs (Coe and Stern, 1982; Stern and Coe, 1984), estimated for this specific day from atmospheric analogs of the day. The probability of precipitation occurrence and the cdf of the non-zero precipitation amount are modeled separately. In the following, we describe the GLMs applied in the regression stage, the Analog Model (AM) used to identify atmospheric analogs and the way these different models are combined to give, for the current prediction day, a probabilistic prediction of precipitation. In the following, the hybrid

15    analog/regression model is further referred to as SCAMP.





### 3.1 Regression stage with GLMs

GLMs make the cdf depending on some covariates, atmospheric predictors in the present case. For each prediction day, the probability of precipitation occurrence $\pi$ is modeled with a GLM in the form of a logistic regression as:

$$\log\left(\frac{\pi}{1-\pi}\right) = \mathbf{x^{o T}}\beta^{\mathbf{o}}, \tag{2}$$

where $\mathbf{x^o}$ is the scalar vector of the $K_o$ predictors and $\beta^{\mathbf{o}}$ the scalar vector of the $K_o$ corresponding regression coefficients.

For the non-zero precipitation amount, we use a GLM with the gamma distribution and the log link function. The expected amount $\mu$ of non-zero precipitation is here expressed as:

$$\log(\mu) = \mathbf{x^{q T}}\beta^{\mathbf{q}}, \tag{3}$$

where $\mathbf{x^q}$ denotes the scalar vector of the $K_q$ predictors and $\beta^{\mathbf{q}}$ the scalar vector of the corresponding regression coefficients.

The shape parameter $\nu$ of the gamma distribution is calculated from the variance $\sigma^2$ of non-zero precipitation amounts estimated from Pearson's residuals (McCullagh and Nelder, 1989) as:

$$\sigma^2 = \frac{1}{\{N_q - (K_q + 1)\}} \sum_{i=1}^{N_q} \frac{(q_i - \mu)^2}{\mu^2}, \tag{4}$$

where $N_q$ is the number of non-zero precipitation data $q_i$ considered in the analysis. As the shape parameter $\nu$ equals the inverse of the variance $1/\sigma^2$, the issued distribution $F_Q$ of the quantity distribution thus follows a gamma distribution $\Gamma(\nu, \alpha = \mu/\nu)$.

For any given prediction day, the estimation of both GLM models practically proceeds as follows: the precipitation state (wet or dry), the precipitation amount and the value of the different potential predictors are extracted for the $N_d$ nearest analogs of the day. The precipitation state of a given day is considered to be wet if the precipitation amount for this day is higher or equal to 0.1 mm. It is described with a binary precipitation occurrence variable $\mathbf{O}$, set to 1 for the wet case, 0 for the dry case.

The occurrence GLM is then estimated from the predictors/occurrence values available for the $N_d$ analogs. The quantity
GLM is similarly estimated from the predictors/quantity values, but only from those available for the wet analog days ($N_q$, the number of days considered for the regression, is therefore smaller or equal to $N_d$ and a priori varies from one target day to another).

For each prediction day, the best predictor set has to be identified for the occurrence and quantity GLMs respectively. In the following, GLMs are fitted in turn for different predefined predictor sets (see details in section 4.2). The predictor set
which minimizes the Bayesian Information Criterion is retained for the prediction (Schwarz, 1978; Akaike, 1974). Regression coefficients of GLMs are estimated using the Iterative Re-weighted Least Squares algorithm (IRLS, Nelder and Wedderburn, 1972). The significance of the regression coefficients is assessed by the $Z$-test (resp. the Student $t$-test). If some predictors are not significant, the least significant is removed and the regression model is re-estimated (this process is repeated until all remaining predictors are significant).

Once the best GLMs are identified for the considered prediction day, the occurrence probability and the expected amount of precipitation are estimated based on the values of the predictors observed for this day. The final distribution of precipitation $F_Y$ is finally obtained by combining the issued occurrence probability $\pi$ and the quantity distribution $F_Q$ according to Eq. 1.



## 3.2 Atmospheric analogs

Atmospheric analogs are identified with an Analog Model defined from the developments of several past studies in France (e.g. Obled et al., 2002; Marty et al., 2012; Radanovics et al., 2013).

The atmospheric analog days here retained for the regression stage are the $N_d$ days that are most similar to the prediction day, in terms of large scale atmospheric circulation. The similarity is assessed by the Teweless-Wobus Score (TWS, Teweless and Wobus, 1954) applied to the geopotential 1000 hPa and 500 hPa respectively at +12h and +24h UTC. The TWS compares the shapes of geopotential fields, and thus informs on the localization of low and high pressures and on the origin of air masses. Note that the $N_d$ analog days are identified within a restricted pool of candidate analog days, namely all days of the archive that are included in a calendar window of $\pm$ 30 calendar days centered on the prediction day. The prediction day and its 5 preceding and following days are excluded from the candidates.

Following Chardon et al. (2014), the domain considered to estimate the atmospheric similarity was optimized for each target location. 8,981 different analog models are thus here considered for the 8,981 SAFRAN grid cells of the study respectively. For each prediction day, the analog days thus likely differ from one SAFRAN grid cell to the next (see Chardon et al., 2014, for illustration).

To estimate the GLMs in the regression stage, we use the 100-nearest atmospheric analog days identified with AM. The $N_d$-nearest analog days can also be directly used, without further regression stage, for a probabilistic prediction. In the following, we will also consider predictions obtained with the 25 nearest analog days (25 was found to give the best prediction skill for France in this configuration by Chardon et al. (2014)). In this case, the precipitation cdf for the prediction day is simply the empirical distribution of the precipitation values observed for these 25 analogs. The predictions obtained with this analog model, further called AM$_{25}$, will give a benchmark to assess the prediction skill of the hybrid analog/regression approach. They will be additionally used as backup prediction for days for which the regression stage failed in the hybrid approach.

## 3.3 The Analog Model as backup prediction model

One may actually face the situation where no GLM satisfies the significance conditions required for the regression coefficients. This can occur for precipitation occurrence probability, for non-zero precipitation amount or for both predictands simultaneously. In such cases, AM$_{25}$ is applied as backup prediction model.

If the significance conditions cannot be satisfied for the precipitation occurrence GLM, the occurrence probability $\pi$ is set to that obtained with AM$_{25}$. It thus simply corresponds to the empirical probability $\pi_{\text{AM}_{25}}$ of precipitation occurrence derived from the 25 analog days of AM$_{25}$ as:

$$\pi \equiv \pi_{\text{AM}_{25}} = \frac{1}{25} \sum_{i=1}^{25} o_i. \tag{5}$$





Similarly, if the significance conditions cannot be satisfied for the precipitation quantity GLM, the distribution $F_Q$ is estimated with the empirical distribution $F_{Q,\mathrm{AM}_{25}}$ derived with $\mathrm{AM}_{25}$ as:

$$F_Q(q) \equiv F_{Q,\mathrm{AM}_{25}}(q) = \frac{F_{\mathrm{AM}_{25}}(q) - (1 - \pi_{\mathrm{AM}_{25}})}{\pi_{\mathrm{AM}_{25}}}, \tag{6}$$

where $F_{\mathrm{AM}_{25}}$ corresponds to the empirical cdf estimated from all precipitations (null and positive) related to the 25 analog days. Note also that if the number $N_q$ of humid analog days is low ($N_q < 10$), the estimation of a GLM is really not robust. When this case appears, $F_Q$ is also set to the cdf obtained with $\mathrm{AM}_{25}$.

As illustrated in Fig. 2, four prediction cases are thus achieved with the hybrid approach. They correspond respectively to cases where $\mathrm{AM}_{25}$ is used to backup the prediction of the whole precipitation distribution (case 1), $\mathrm{AM}_{25}$ is applied to backup the quantity cdf prediction (case 2), $\mathrm{AM}_{25}$ is used to backup the occurrence probability prediction (case 3) and the regression stage could be activated for both occurence and quantity (case 4).

Note that the regression stage achieved with GLMs can also be seen as a way to refine the estimation of the cdf that could have been obtained directly with the backup (and possibly benchmark) $\mathrm{AM}_{25}$ analog model. The refinement leads to update the occurrence probability and/or the cdf of non-zero precipitation amount.

As described previously, the two-stage analog/regression prediction process is repeated for each prediction day in turn. As the analog days vary from one prediction day to another, the predictors selected in the regression stage and the value of the corresponding regression coefficients are expected to vary from one prediction day to the other. The hybrid model SCAMP allows thus for a day-to-day adaptive and tailored downscaling.

## 3.4 Model evaluation

The prediction skill of the downscaling model is assessed with probabilistic scores classically used to evaluate some Ensemble Prediction System (EPS).

The Brier Score (Brier, 1950; Murphy, 1973) first evaluates the ability of the considered EPS $\mathcal{P}$ to predict precipitation occurrence. When estimated over $M$ prediction days, the mean Brier Score $\overline{\mathrm{BS}}$ reads:

$$\overline{\mathrm{BS}} = \frac{1}{M} \sum_{i=1}^{M} [p_i - o_i]^2, \tag{7}$$

where, for a given prediction day $i$, $p_i$ is the occurrence probability issued by EPS $\mathcal{P}$ and $o_i$ is the effective precipitation occurrence for this day ($o_i = 1$ for a wet day, $= 0$ otherwise).

The ability of EPS $\mathcal{P}$ to estimate the precipitation amount is evaluated with the Continuous Ranked Probability Score (CRPS, Brown, 1974; Matheson and Winkler, 1976). When estimated over $M$ prediction days, the mean CRPS reads:

$$\overline{\mathrm{CRPS}} = \frac{1}{M} \sum_{i=1}^{M} \int_{-\infty}^{+\infty} [F_i(x) - H_{y_i}(x)]^2 dx, \tag{8}$$

where, for a given prediction day $i$, $H_{y_i}$ and $F_i$ denote respectively the cdf of the observation $y_i$ and the cdf derived from $\mathcal{P}$. $x$ denotes the predictand quantiles of the cdfs. Note that $H_{y_i}$ corresponds to the Heaviside function where $H_{y_i} = 1$ if $x \geq y_i$ and





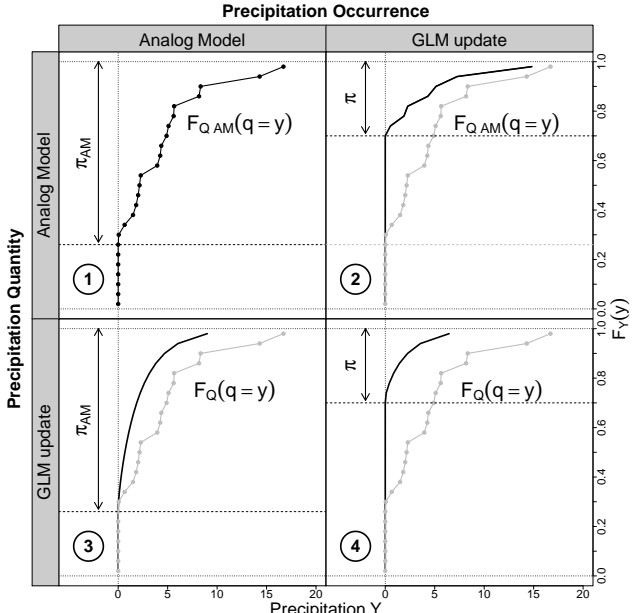

**Figure 2.** Illustrations of the four cases met for the issue of $F_Y(y)$ by the hybrid model. Case 1: None of the occurrence and quantity GLMs could be retained during the regression stage: $AM_{25}$ is used to predict the whole precipitation distribution. Case 2: Only the occurrence GLM could be retained. It gives the estimated occurrence probability. The distribution of non-zero precipitation comes from $AM_{25}$. Case 3: Only the quantity GLM could be retained. It gives the distribution of non-zero precipitation. The occurrence probability is the empirical occurrence probability from $AM_{25}$. Case 4: Both occurrence and quantity GLMs could be estimated: they respectively give the occurrence probability and the distribution of non-zero precipitation, to be further combined for the full distribution of precipitation.

$H_{y_i} = 0$ otherwise. A graphical interpretation of the CRPS for a given prediction $i$ is illustrated in Chardon et al. (2014, Fig. 1).

For this evaluation, the probabilistic prediction of the predictand $y$ is here practically described, for each prediction day, with a discretized cdf composed of $N$ values, with $N = 25$. When $AM_{25}$ is used as backup model, the $N$ values are the precipitation observations of the 25-first analog days. When the prediction is issued with SCAMP , the $N$ values are those of the 25 percentiles $(k - 0.5)/25$, $k$ in $[1, N]$, of the predicted cdf $F_Y$.

In the following, we discuss the prediction skill for precipitation occurrence and amount with the Brier Skill Score (BSS) and the Continuous Ranked Probability Skill Score (CRPSS) respectively. Both scores normalize the prediction skill of $\mathcal{P}$ with that obtained with a reference prediction model $\mathcal{P}_\varphi$. $\mathcal{P}_\varphi$ is here a climatological prediction based on a calendar climatology defined for each prediction day by the precipitation distribution of all days belonging to a seasonal window ($\pm$ 30 days) centered on the corresponding calendar day. In this context, the BSS and CRPSS respectively read:

$$\text{BSS} = 1 - \frac{\overline{\text{BS}}}{\overline{\text{BS}}_\varphi}, \tag{9}$$





and

$$\text{CRPSS} = 1 - \frac{\overline{\text{CRPS}}}{\overline{\text{CRPS}}_{\mathcal{P}_\varphi}}, \tag{10}$$

where $\overline{\text{BS}}_\varphi$ and $\overline{\text{CRPS}}_{\mathcal{P}_\varphi}$ correspond to the mean BS and the mean CRPS obtained with $\mathcal{P}_\varphi$. For both scores, a negative value indicates that the prediction obtained with $\mathcal{P}$ is worse than the prediction obtained with the climatological EPS $\mathcal{P}_\varphi$. A

score of 1 conversely denotes a perfect EPS $\mathcal{P}$.

In the following, to assess the added value of the hybrid SCAMP model when compared to the benchmark $\text{AM}_{25}$ analog model, we additionally estimate the gain in prediction skill as: $\Delta S = S_{\text{SCAMP}} - S_{\text{AM}_{25}}$ where $S$ corresponds either to the BSS or the CRPSS.

## 4   Results

### 4.1   Predictor sets

The hybrid model is used for the probabilistic prediction of local precipitation over the French metropolitan territory for each day of the 1982-2001 period (20 years). For each prediction day, predictors used in the regression stage are identified in turn for each SAFRAN grid cell. They are thus expected to be both day and location specific.

For each day, the 13 atmospheric variables described in Table 1 were considered as potential predictors for the regression

stage. A large number of different possible predictor sets can be built from those variables. For the sake of robustness, we considered that a maximum of five predictors could be integrated in a given GLM. In the present work, for the sake of simplicity and readability, we additionally restricted the number of predictor sets considered. This allows us to reduce the degrees of freedom's number in the model and to better highlight its skill and adaptive behavior.

The different predictor sets considered for the prediction of precipitation occurrence were selected as follows. We first

considered the simplified case where a unique predictor (i.e., a same predictor for all prediction days) is used in the regression stage. From the 13 potential predictors considered in turn, only nine allow a better BSS than that obtained with $\text{AM}_{25}$ (on average for the 8,981 grids cells in the whole French territory). These only nine predictors were retained for further analyses. We next considered the configuration where the predictor set is the same for all prediction days. The best predictor set was identified from a classical iterative forward/backward algorithm. It was identified for a selection of 12 SAFRAN grid cells

uniformly distributed over the French territory. For these cells, the optimal number of predictors was generally smaller than five and the predictor $R_{700}$ was always retained. We thus further worked with all sets of four predictors that include $R_{700}$. The skill score obtained with these $\binom{8}{3} = 56$ sets was evaluated. The best sets were further evaluated for the 8,981 SAFRAN grid cells. For the sake of simplicity, we retained only one of these for the present work, namely the four predictor combination which obtained in average the best BSS over France. The corresponding best predictors are the relative humidity $R_{700}$, the

helicity $H$, the vertical velocity $W_{700}$ and the precipitation occurrence $Occ - 1$ of the day before the prediction day.

The same selection process was followed for the quantity predictor set. In the fixed predictor sets configurations, the vertical velocity $W_{700}$ turned out to be always selected and the optimal number of predictors was also found to be equal to four. The





quantity predictor set finally retained is similar to the one identified for occurrence except that the occurrence of the previous day $Occ - 1$ is replaced by the 700 hPa air temperature $T_{700}$. Note that the selection of predictors $R_{700}$, $W_{700}$ and $T_{700}$ is consistent with results of several past studies in the region (e.g. Ben Daoud et al., 2016).

We further present the prediction skill obtained for occurrence and quantity with these two four predictors sets respectively. As discussed later in section 5, the four predictors are not necessary all used, the predictors really explaining the predicatant vary from one day to the other.

## 4.2 Performance of SCAMP

Figure 3a presents the BSS skill score of SCAMP for precipitation occurrence prediction. The highest BSS values – up to 0.5 – are found in the western part of the Massif Central, in the Alps and along the Atlantic coast. Lower skill (BSS from 0.45 to 0.5) is obtained in northern and western lowlands. The lowest skill (0.35) is obtained for few cells located along the Mediterranean coast.

The BSS gain obtained with SCAMP over $AM_{25}$ is rather important (up to 0.1 BSS points) but very sensitive to the topography (Fig. 3b). Most of the gains (between 0.05 and 0.1) are obtained in the Pyrenees and East of France (i.e. Morvan massif, Alps, Rhone valley, East of Massif Central). The highest gains are found along the Mediterranean coast and in the Southern Alps where the BSS of SCAMP was the lowest. This highlights the weakness of the $AM_{25}$ in these regions – characterized by more frequent convective precipitation and thus a weaker link with large-scale atmospheric circulation – and the interest for thermodynamic and more local predictors. Conversely, lower gains are observed in the western part of France characterized by more frontal precipitation and thus a stronger link with large-scale circulation. We also found that the spatial distribution of $\Delta$BSS is very close (even if it has higher values) to the one obtained by SCAMP with $R_{700}$ as unique predictor (not shown here).

Figure 4a similarly presents the CRPSS obtained with SCAMP. The CRPSS values also depend on topography. The highest values, up to 0.45, are obtained in the western part of the Massif Central, the northern Alps, the Jura and the Vosges massifs. Lower values, between 0.32 and 0.45, are obtained in lowlands. The lowest skill (below 0.30) is again obtained along the Mediterranean coast.

The CRPSS gain obtained over $AM_{25}$ is significant (up to 0.07) for most grid cells, with the highest value (up to 0.10) obtained in the Rhone valley and in north-eastern France (Fig. 4b). Similarly to what was obtained with the BSS gain for occurrence, a lower CRPSS gain is here also obtained in lowlands and western France. We also found that the spatial distribution of $\Delta$CRPSS is very close (even if it has higher values) to the one obtained by SCAMP with $W_{700}$ as unique predictor for quantity (not shown here).

Despite the large dependency to regional features such as topography or proximity to the sea, adding local and thermodynamic information in SCAMP greatly and thus significantly improves the prediction skill over that of $AM_{25}$, for both precipitation occurrence and quantity.



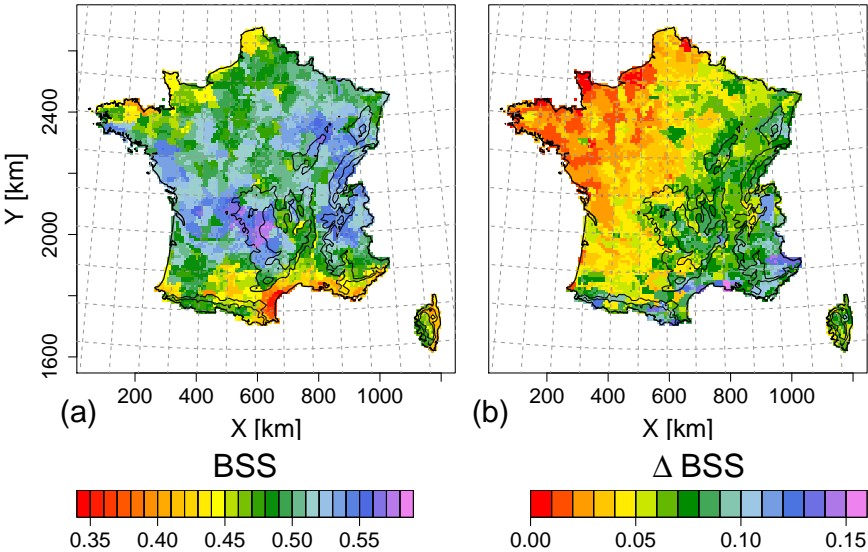

**Figure 3.** (a) BSS obtained with SCAMP. (b) BSS gain obtained with SCAMP compared to $AM_{25}$. Black solid lines correspond to the French borders and the contours around mountainous regions (400- and 800-m elevation) while the dashed lines show the ERA-40 grid mesh.

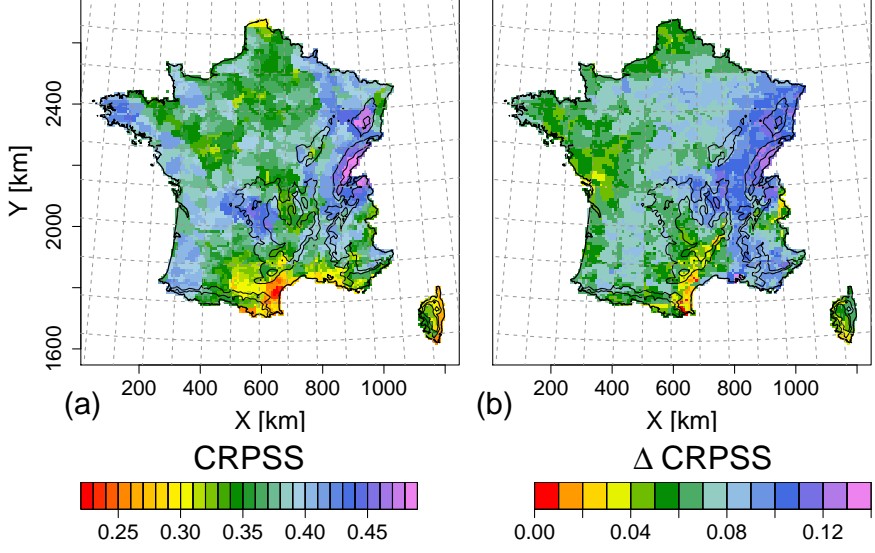

**Figure 4.** (a) CRPSS obtained with SCAMP. (b) CRPSS gain obtained with SCAMP compared to $AM_{25}$.

## 4.3 Characterisation of SCAMP's behaviour

As described in section 3.3, the regression stage of SCAMP is equivalent to updating the empirical distribution obtained from the atmospheric analogs directly. For some prediction days, the regression stage may be however only partly activated, for





either occurrence or quantity. It may be even not activated at all. In these cases, the prediction is fully or partly obtained from the backup model $AM_{25}$.

The situation where both precipitation and quantity are predicted with GLMs (case 4) is very frequently observed (Fig. 5). It corresponds to more than 85 % of the days except in south-eastern France where only 60 % of the days are concerned. Case 2, where only occurrence probability is predicted by a GLM appears between 5 and 15 % of the time, except in south-eastern France where up to 35 % of the days are concerned. All in all, the regression stage of SCAMP is thus very often activated to predict occurrence probability (case 2 + 4) and the situation where $AM_{25}$ is used as backup for this variable is very rare (case 1+3). In the failing full-updating cases, $AM_{25}$ is classically used to backup precipitation quantity prediction (case 1+2). Case 3, where the only quantity is predicted by a GLM, is rather rare, appearing less than 5 % of the time excepted in the very western part of Brittany. Case 1, where the whole prediction is obtained with $AM_{25}$ is finally very rare. For a large majority of the grid cells, it is met less than 35 times in the 20-year period considered (around 5‰).

Figure 6 presents the mean precipitation anomaly for each of the previous cases, i.e. the ratio between the mean amount obtained for all days belonging to the considered case and the overall mean precipitation amount. An anomaly greater (resp. lower) than 1 indicates days that are rainier (resp. drier) than usual. The different cases correspond clearly to different precipitation configurations. The mean precipitation amount of days in case 4 is close to the overall mean. Days in cases 1 and 2 are very dry. Days in case 3 are very wet with a mean precipitation three times greater than the overall mean.

For a given prediction day, the precipitation state of its analog days is actually roughly similar to that of the day. This thus explains the SCAMP's behavior described above. In case 1, almost all analog days are found to be very dry and the observed precipitations extracted from analog days are almost all equal to 0. The proportion of wet analog days is next too low to allow for robust estimation of the occurrence GLM. As a consequence also, the number of humid analog days is also often too small to robustly model the precipitation quantity. This leads to use the backup $AM_{25}$ for the prediction. Case 2 is similar but the proportion of wet day is higher than in case 1. A robust occurrence GLM can thus be estimated in this case. In case 3, prediction days are very humid and are characterized by a very large number of humid analog days. In this case, the proportion of wet analog days is too high to allow for robust estimation of the occurrence GLM. A robust GLM estimation can be conversely achieved for precipitation quantity. In case 4, a balanced proportion of dry and wet analog days is found. This allows for the estimation of both the quantity model and the occurrence model.

The CRPSS gain achieved with SCAMP's results from the updated prediction of both precipitation occurrence and quantity. To assess the relative effects of these updates on the gain, we further compared the four following prediction experiments:

**Exp. 1:** The prediction of both the occurrence and the quantity is achieved with $AM_{25}$ for all prediction days. This corresponds to the results given by Chardon et al. (2014, cf. Fig. 3).

**Exp. 2:** When possible, the precipitation occurrence probability is updated with the occurence GLM. The non-zero precipitation quantity is always predicted with $AM_{25}$.

**Exp. 3:** When possible, the precipitation quantity is updated with the quantity GLM. The precipitation occurrence probability is always predicted with $AM_{25}$.





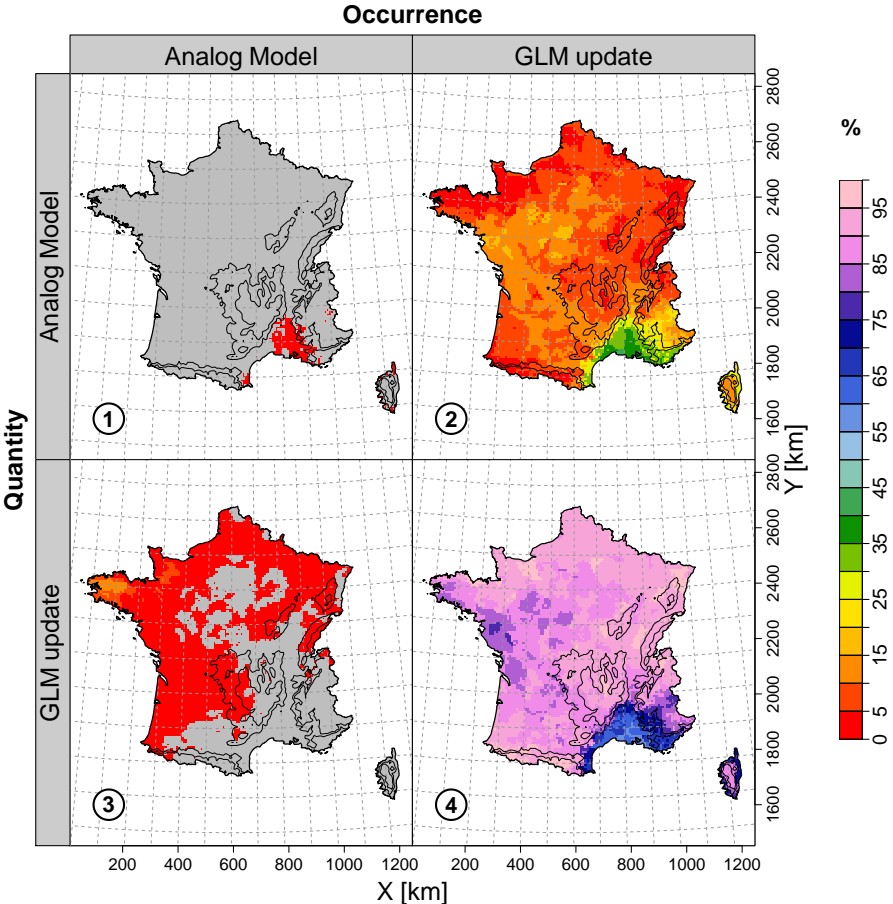

**Figure 5.** Percentage of days where 1) no updates are applied, 2) only the precipitation occurrence is updated, 3) only the precipitation quantity is updated and 4) the occurrence and the quantity of precipitation are updated. Grids with gray colors correspond to grid cells where the corresponding case has been met less than 35 times over the 20-year evaluation period.

**Exp. 4:** When possible, both precipitation occurrence probability and quantity are updated with the occurence and quantity GLMs. This corresponds to the hybrid configuration already evaluated previously.

The CRPSS gain obtained between Exp.1 and 2, between Exp. 1 and 3 and between Exp.1 and 4 are presented in Fig. 7 (the results for Exp.4, already presented in Fig. 4, are presented again for the ease of comparison).

5    For a large majority of grid cells, the CRPSS gain obtained with an updated prediction of the occurrence probability (from 0 to 0.05) is significantly lower than that obtained with an updated prediction of quantity (from 0.03 to 0.1). The CRPSS gain obtained in the latter case is additionally close to that obtained with the full hybrid model. The CRPSS gain obtained by SCAMP in Fig. 7c is thus explained in most cases by the updated prediction of precipitation quantity. The scheme is somehow different in the south of France along the Mediterranean coast and in the Cevennes-Vivarais mountains. In those regions, the





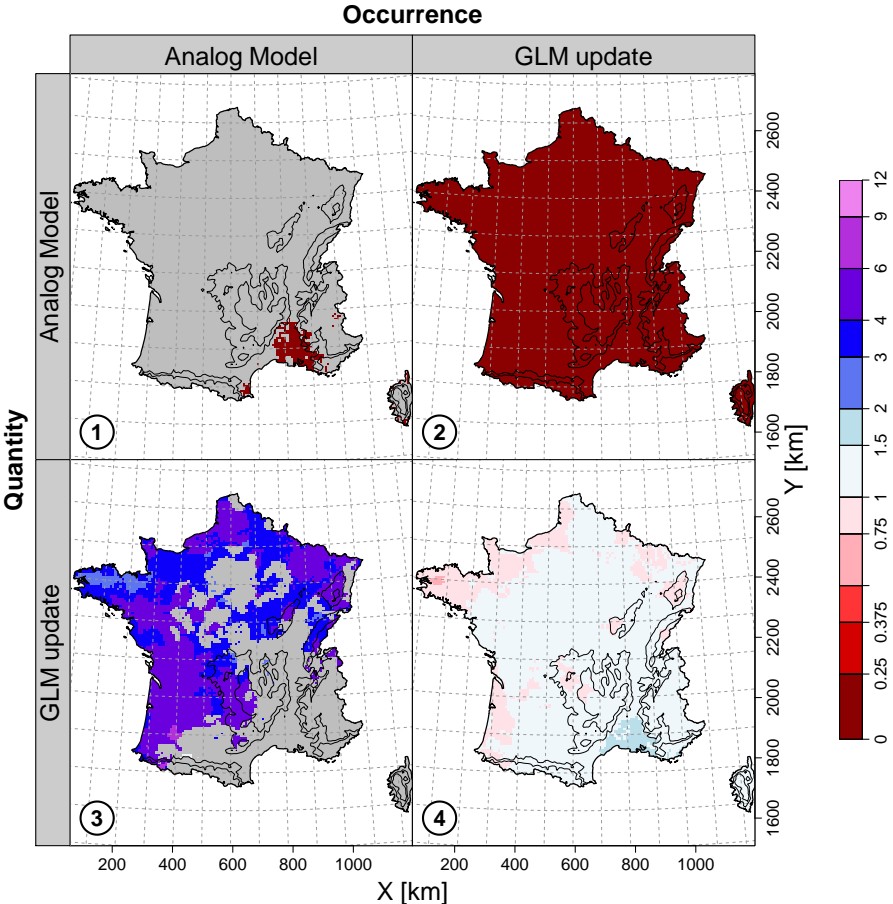

**Figure 6.** Ratio between the mean amount obtained for all days belonging to a given case and the overall mean precipitation amount. Gray grids: same as in Fig. 5.

CRPSS gain obtained by SCAMP is mostly explained by the updated prediction of the occurrence probability. Updating only the quantity leads to fairly no CRPSS gain. The quantity predictor set, optimized for the whole of France, is likely not really optimal for this region.

# 5 Discussion

The sets of predictors used in SCAMP for the prediction of precipitation occurrence and quantity have been listed in section 4.1. For each set, the number of predictors is equal to four. All four predictors are not necessary retained for the GLM. For a given prediction day, a GLM with a single predictor or a combination of several predictors among the four may be selected. All combinations which are possible in our context, denoted as "regressive structures" in the following, are listed in Table 2.





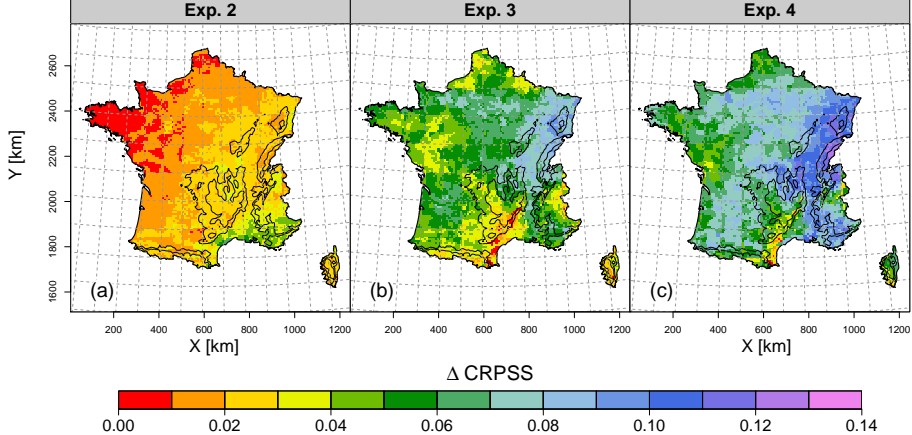

**Figure 7.** Gain in CRPSS for different prediction experiments (see section 4.3 for details) compared to the performance of $AM_{25}$. (a) Exp.2: only the precipitation occurrence probability is updated (when possible), (b) Exp.3: only the precipitation quantity is updated (when possible), (c) Exp.4: both occurrence probability and quantity of precipitation are updated (when possible).

For a given prediction day, the regressive structure selected by SCAMP for the precipitation occurrence or for the quantity are supposed to contain the best information for the prediction. In the following, we assess the frequency each structure has been selected. This allows us to give some insight in the atmospheric information really used for the regression stage and how this information may vary in time.

Figure 8 and Fig. 9 present the percentage of times that the 15 regressive structures and the backup $AM_{25}$ are used for the prediction of precipitation occurrence and quantity, respectively. As in Fig. 5, gray cells indicate that the regression structure has been retained less than 35 times over the 20-year evaluation period.

For occurrence (Fig. 8), the most often selected structure is Str. nº1, which is only based on $R_{700}$ (more than 25 % for the whole of France). $R_{700}$ was actually found to give the highest predictive power when used in a single predictor configuration (see section 4.1). Another structure which is also often selected (more than 15 % for a high number of grids) is Str. nº7 which combines $R_{700}$ with $Occ-1$. Secondary structures – as for example Str. nº6 and nº13 combining $R_{700}$ to $W_{700}$ and $Occ-1$ – can be selected more than 10 % of the days for some given regions. Other structures are seldom selected and some of them (Str. nº8, nº11, nº14 and nº15) are almost never selected. The selection frequency of the structures is also rather region depend and strongly influenced by topography.

Similar results are obtained for quantity. The selected regressive structures gather one principal structure, Str. nº3 which only includes $W_{700}$, and some secondary structures (Str. nº1, nº6 ,nº8 and nº13 including the other predictors). Str. nº9, nº12, nº14 and nº15 are almost never selected.

Note that for the selection of the best regression structure for a given prediction day, all these 15 regressive structures have been in turn tested. The results above suggest that this systematic test is not necessary and that it could be reasonable to consider only the few structures which are retained a reasonable fraction of the days. This would be however not necessary





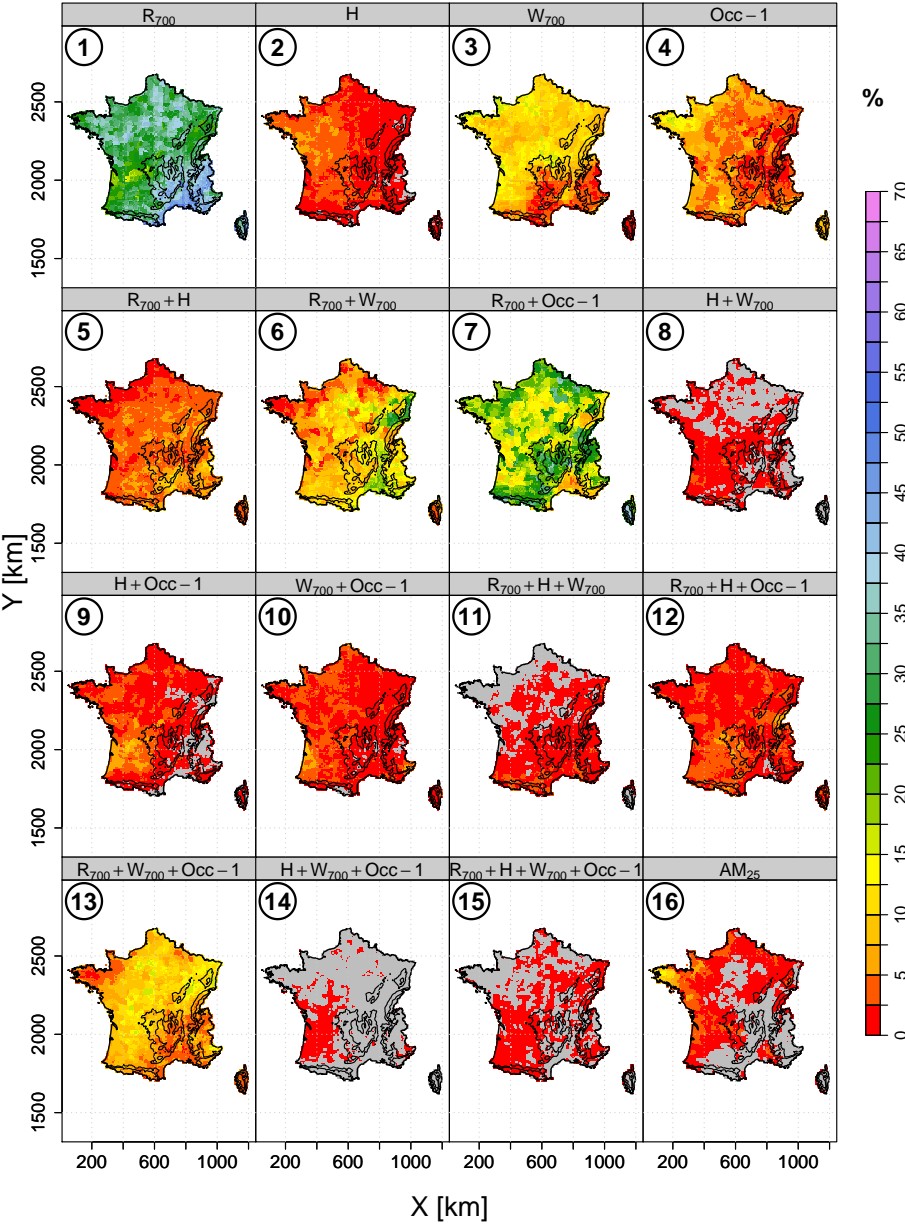

**Figure 8.** Prediction of occurrence probability: selection frequencies (%) of the 15 regression structures and of the backup model $AM_{25}$. Predictors involved are indicated in graphs headers and index of the regressive structure in top left corners. Gray grids: same as in Fig. 5.

relevant nor desirable. The selection frequency of a given structure actually varies with the seasons and/or the encountered synoptic situation and some secondary regressive structures can be retained frequently for specific situations. For a given cell located in north-western France, Fig. 10 illustrates the absolute difference of selection frequency for each season and for





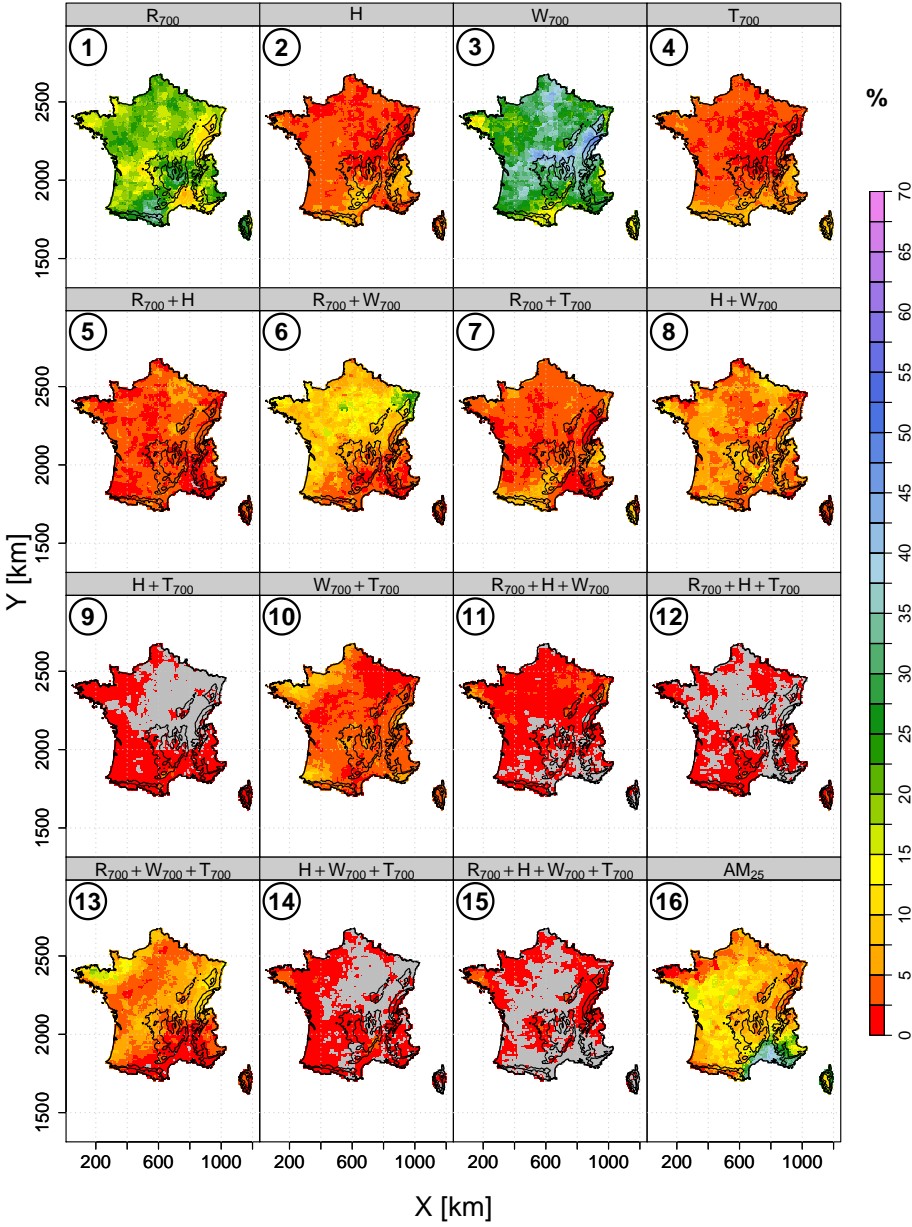

**Figure 9.** Same as Fig. 8 for the probabilistic prediction of precipitation quantity.

several weather patterns (WP, defined in Table 3, Garavaglia et al., 2010) compared to the one obtained for the whole period. A positive selection frequency difference indicates that the considered regressive structure is more selected compared to usual (and conversely in case of a negative difference).





In Fig. 10a, the selection of the main regressive structures previously identified in Fig. 8 (i.e. Str. n°1 and n°7 respectively based on $R_{700}$ and $R_{700} + Occ - 1$) and of the $AM_{25}$ greatly varies (from -15 % to 15 %) compared to usual. This can be also noticed in Fig. 10b for the main regressive structures characterizing the precipitation quantity (i.e. Str. n°1 and n°3). The reduced selection of a given main regressive structure for a given season or WP can lead to preferentially retain some secondary

regressive structure. An example can be seen in Fig. 10b for WP2, where the regressive Str. n°8 based on $W_{700} + H$ is 10 % more selected than usual.

Figure 11 illustrates the extra- and reduced-selection frequency obtained for three WPs with two regressive structures when used for the prediction of precipitation quantity for the whole of France. These results are illustrative of those obtained for the other regressive structures and WPs (both for the precipitation occurrence and quantity).

The atmospheric circulation of the three WPs is characterized in Fig. 11a by the corresponding mean geopotential height at 1000 hPa. For each of these WPs, the absolute selection frequency difference (in %) compared to usual is plotted in Fig. 11b for the quantity regressive Str. n°3 and n°8 (i.e. $W_{700}$ and $W_{700} + H$). It appears that:

– A regressive structure could be preferentially retained for a given WP for a wide region. As an example, Str. n°3 is selected much more often for WP7 for the overall grid cells (more than +15 % compared to usual). This highlights the

importance of the vertical velocity $W_{700}$ for this special WP for the prediction of precipitation quantity.

– For a same WP, the extra-selection of a preferential regressive structure depends on the considered region and on the topographical barriers. For example, the structure only based on $W_{700}$ is more selected along the Atlantic coast and in the north of France for WP2. For these regions, adding the predictor $H$ to $W_{700}$ leads also to a larger selection of Str. n°8. Conversely, these two structures are less selected in the southeast which benefits to the use of the alternative $AM_{25}$.

This region – protected by the Massif Central mountain – does not usually obtain precipitation for WP2 (cf. Fig. 3 of Garavaglia et al., 2010) which justify the use of the alternative $AM_{25}$ (cf. section 4.3). A similar result is observed for WP8 corresponding to an Anticyclonic situation.

– For some few frequent WPs, a regressive structure could be preferentially selected at a given specific location. In Fig. 11b, regressive Str. n°8 based on $W_{700} + H$ is much more selected (around +15 %) in the Cevennes-Vivarais regions

(south-eastern part of the Massif Central) and in the pre-alpine mountains (western part of the Alps). The combination of $W_{700}$ and $H$ seems thus to be very informative in these locations for this really few frequent WP (4 % of the 20-year period).

In Fig. 8, 9 and Fig. 11b, a noticeable point is finally also that the extra-selection of the regressive structures appears to be spatially coherent even for less frequent WPs as WP7. This suggests the spatial robustness of the selection of the regression

structures.




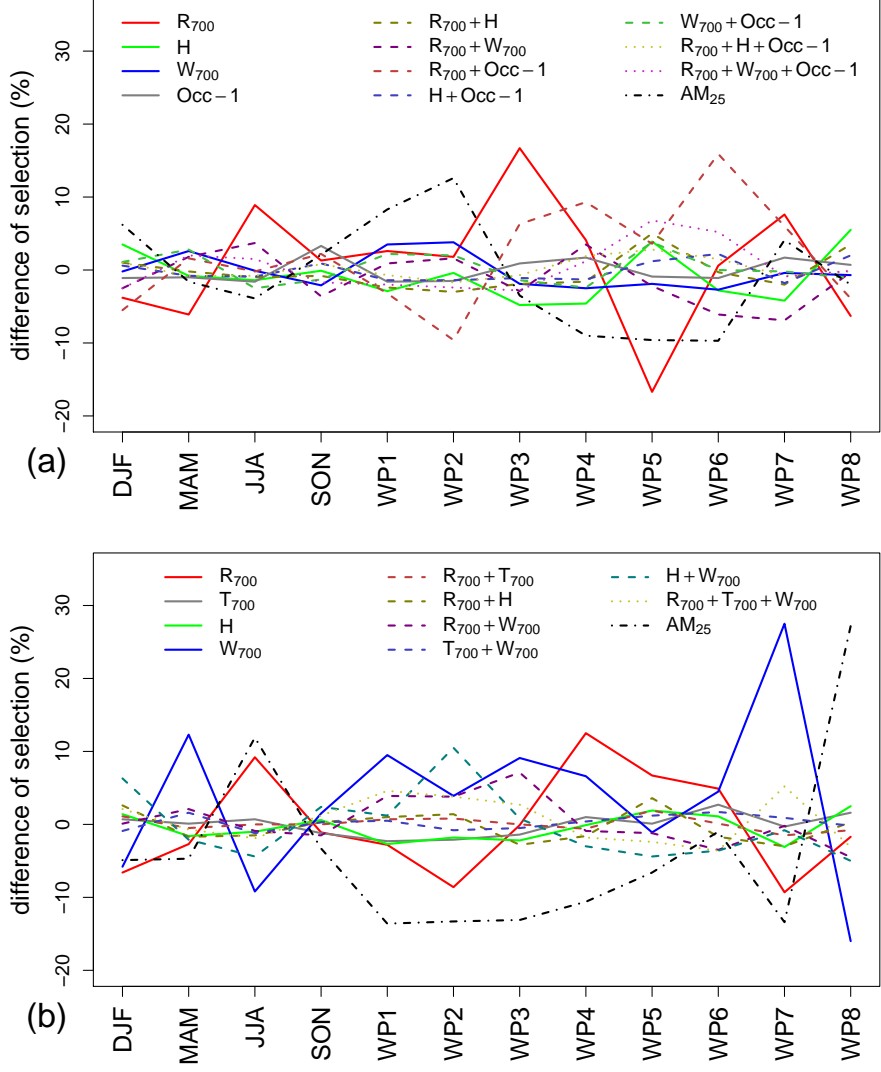

**Figure 10.** For each season and weather type, difference (%) in selection frequency with the all-days case for different regression structures. Results for the prediction of (a) occurrence and (b) quantity. Results are displayed for a grid cell located in the north-west of France. For a clearer illustration, the three or four regressive structures that are almost never selected are not displayed.

## 6 Conclusions

The interest of a hybrid analog/regression model has been explored in this study for the probabilistic prediction of precipitation over France. Atmospheric analogs of the prediction day are identified to estimate a two-part regression model further applied for the prediction. The regression model consists of a logistic GLM for the prediction of precipitation occurrence and a logarithmic

5   GLM for the prediction of precipitation quantity. The prediction obtained with this hybrid approach actually updates the



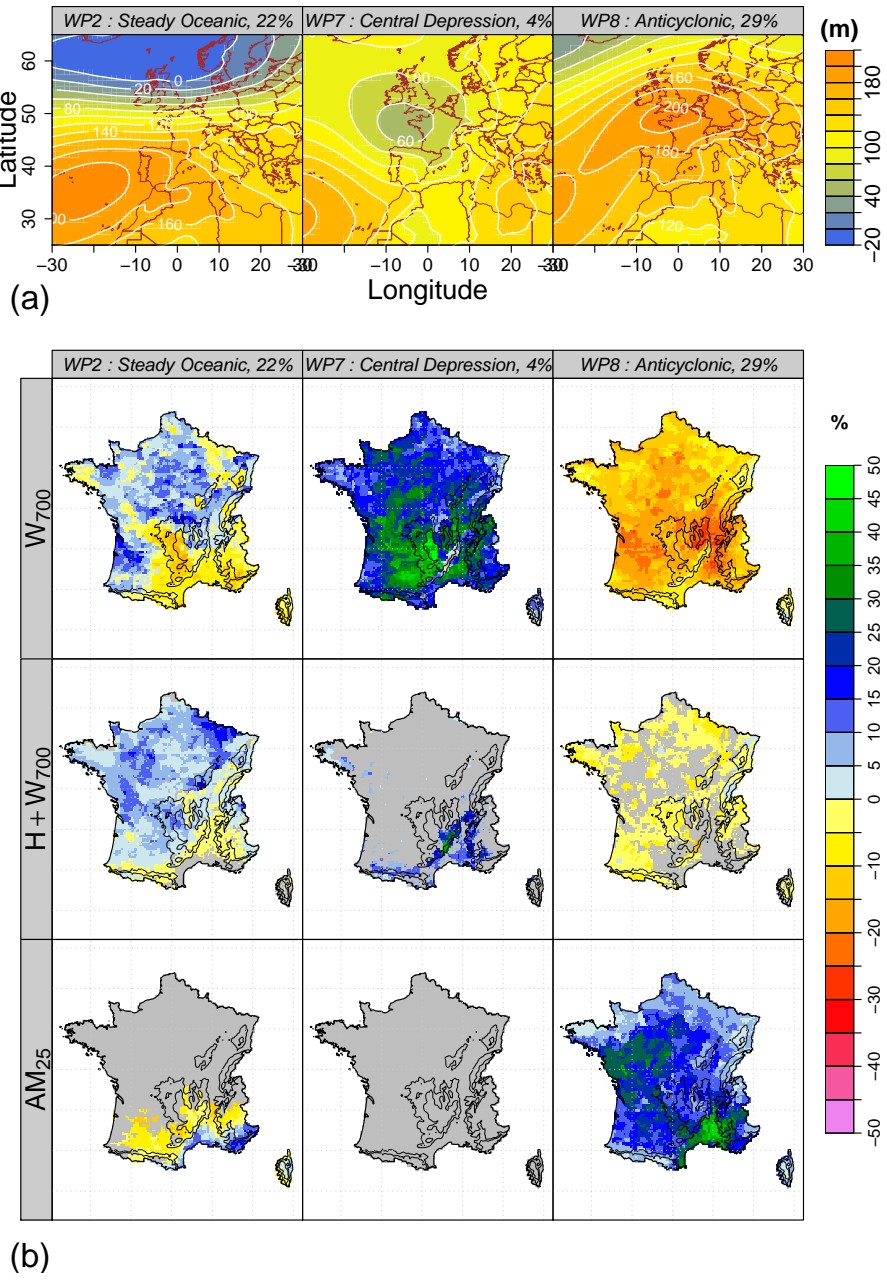

**Figure 11.** (a) Mean geopotential field at 1000 hPa for three WPs (Garavaglia et al., 2010). (b) For each WP, difference (%) in selection frequency with the all-days case. Results for two regression structures ($W_{700}$ and $W_{700} + H$) and for $AM_{25}$. Predictand is precipitation quantity. A positive difference indicates an extra-selection, while a negative difference to a reduced selection. Gray grids: same as in Fig. 5.



predictive distribution that would have been achieved directly with a one stage analog model based on atmospheric circulation analogs. The hybrid approach makes the downscaling model adaptive: as the analog days are identified for each prediction day, the predictors and regression coefficients of the regression models vary from one day to the other.

The regression stage allows a non-negligible prediction skill gain compared to the prediction that could be directly obtained from the atmospheric analogs of the analog stage (gain up to 0.1 points for both the BSS and the CRPSS skill scores). The CRPSS gain is mainly achieved thanks to the regression model estimated for the precipitation quantity. The introduction of local scale predictors such as relative humidity is obviously crucial there. The adaptive nature of the model and thus the possibility to tailor the downscaling relationship (both predictors and regression coefficients) to the current prediction day seems also to be decisive. The CRPSS gain obtained with the hybrid approach is actually twice higher than the one obtained by Chardon et al. (2014) with a two level analog model where a unique and same second level analogy variable (namely humidity) is considered for all days.

The prediction skill and adaptability of this hybrid approach was illustrated for the prediction of both the precipitation occurrence and quantity in a simplified configuration where four predictors, selected in a preliminary analysis from a large ensemble of potential predictors, are used in the regression stage. The predictors used for precipitation occurrence are the relative humidity and vertical velocity at 700 hPa, the helicity integrated from 1000 hPa to 500 hPa and the occurrence of the previous day. A similar set of predictors is used for the precipitation quantity (the occurrence of the previous day is replaced by the 700 hPa temperature). For each prediction day, the estimated regression models for precipitation occurrence and quantity are next retained among two respective lists of 15 regression structures, each corresponding to a single predictor or to a combination of them. Most of the time, regressive structures are only composed of one or two predictors including the main informative variables (i.e. the relative humidity $R_{700}$ and the vertical velocity $W_{700}$ for precipitation occurrence and amount respectively). Some combinations of predictors, almost never used in general, appear to be more frequently retained for some specific weather patterns and/or locations in France, revealing their potential interest for these situations.

For the sake of simplicity and to limit the degrees of freedom in our analysis, we considered a unique set of four potential predictors for all SAFRAN grid cells. This obviously leads to a sub-optimal prediction configuration. The main meteorological processes driving precipitation in France obviously differ from one region to the other. The most informative predictors are thus expected to be region-dependent and the set of predictors to be considered in the regression stage could be refined on a regional basis. This is expected to improve the skill of the prediction.

A number of atmospheric variables have been considered as potential predictors in similar downscaling studies. The predictors found to be of interest are classically few. They are roughly the same than those considered in the preliminary analysis of the present work. However, as in the present work, the analyses classically carried out to identify these informative predictors are potentially misleading. The selection of a variable is indeed classically based on its predictive power, estimated with some prediction skill score in an all-days evaluation framework. As highlighted in the present work however, some predictors are likely to be informative for very few meteorological situations. An all-days evaluation is expected to reveal robust predictors. It however very likely misses important situation-specific predictors. The hybrid approach here estimates the statistical downscaling link from a homogeneous set of days that are moreover atmospheric analogs to the prediction day. This hybrid





approach has thus the potential to reveal the predictive power of very specific predictors, suited for very specific meteorological configurations. It leaves very likely room for significant improvements of the prediction skill in this context. It gives likely also the opportunity to better understand the atmospheric factors under play in a number of non-frequent and atypical meteorological situations. Notwithstanding the technical limitations that may hamper such analyses, a much larger exploration of a much larger diversity of predictors would be thus definitively worth in this context.

*Author contributions.* This study is part of J. Chardon's PhD thesis. B. Hingray and A-C. Favre supervised the PhD. All authors contributed to the designed experiments to the writing of the document. J. Chardon developed the model code and performed the simulations.

*Competing interests.* The authors declare that they have no conflict of interest.

*Acknowledgements.* The authors especially thank Charles Obled and Isabella Zin for fruitful discussions on the analog method. The authors thank also the Grenoble University High Performance Computing centre CIMENT (https://ciment.ujf-grenoble.fr/wiki-pub/index.php/Welcome_to_the_CIMENT_site!) for their help and the large computing resource they provide.



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





**Table 1.** Large-scale potential variables retained for the establishment of GLMs.

| Acronym | Predictor description |
| --- | --- |
| $R_{850}$ | Relative humidity at 850 hPa |
| $R_{700}$ | Relative humidity at 700 hPa |
| $R_{850}TCW$ | Product of $R_{850}$ and the Total Column Water ($TCW$) |
| $FR_{700}$ | Humidity flux intensity at 700 hPa |
| $\nabla FR_{700}$ | Divergence of $FR_{700}$ |
| $T_{700}$ | Temperature at 700 hPa |
| $B_{700}$ | Baroclinicity at 700 hPa |
| $Z_{1000}$ | Geopotential height at 1000 hPa |
| $H$ | Helicity integrated from 1000 to 500 hPa |
| $W_{700}$ | Vertical velocity at 700 hPa |
| $PV_{400}$ | Potential vorticity at 400 hPa |
| $\Delta\theta$ | Potential temperature gradient between 925 and 700 hPa |
| $Occ-1$ | Precipitation occurrence of the day before the prediction day |





**Table 2.** Possible regressive structures (i.e. combination of predictors) for the modeling of precipitation occurrence and quantity.

| Structure index | Precipitation occurrence | Precipitation quantity |
|---|---|---|
| Str. n°1 | $R_{700}$ | $R_{700}$ |
| Str. n°2 | $H$ | $H$ |
| Str. n°3 | $W_{700}$ | $W_{700}$ |
| Str. n°4 | $Occ\text{-}1$ | $T_{700}$ |
| Str. n°5 | $R_{700} + H$ | $R_{700} + H$ |
| Str. n°6 | $R_{700} + W_{700}$ | $R_{700} + W_{700}$ |
| Str. n°7 | $R_{700} + Occ\text{-}1$ | $R_{700} + T_{700}$ |
| Str. n°8 | $H + W_{700}$ | $H + W_{700}$ |
| Str. n°9 | $H + Occ\text{-}1$ | $H + T_{700}$ |
| Str. n°10 | $W_{700} + Occ\text{-}1$ | $W_{700} + T_{700}$ |
| Str. n°11 | $R_{700} + H + W_{700}$ | $R_{700} + H + W_{700}$ |
| Str. n°12 | $R_{700} + H + Occ\text{-}1$ | $R_{700} + H + T_{700}$ |
| Str. n°13 | $R_{700} + W_{700} + Occ\text{-}1$ | $R_{700} + W_{700} + T_{700}$ |
| Str. n°14 | $H + W_{700} + Occ\text{-}1$ | $H + W_{700} + T_{700}$ |
| Str. n°15 | $R_{700} + H + W_{700} + Occ\text{-}1$ | $R_{700} + H + W_{700} + T_{700}$ |





**Table 3.** Names of the weather patterns (WP) defined in Garavaglia et al. (2010) and related frequency for the 01 August 1982-08-01 to 2001-07-31 period.

| Index | Denomination | Annual frequency (%) |
|-------|--------------|----------------------|
| WP1 | Atlantic Wave | 8 |
| WP2 | Steady Oceanic | 22 |
| WP3 | Southwest Circulation | 8 |
| WP4 | South Circulation | 17 |
| WP5 | Northeast Circulation | 6 |
| WP6 | East Return | 6 |
| WP7 | Central Depression | 4 |
| WP8 | Anticyclonic | 29 |