# Peer review of "An adaptive two-stage analog/regression model for probabilistic prediction of small-scale precipitation in France."

_Hydrology and Earth System Sciences, 2017_

## Referee Comment (RC1) · Anonymous Referee #1 · 19 May 2017

**1   Topic and general comments**

**1.1   Topic**

The paper presents a new two-stage hybrid perfect prognosis SDM called SCAMP. SCAMP was applied to a large number of grid points in France and was proven to be adaptive to different weather types and seasons which is illustrated nicely with visually appealing figures. The method seems very interesting given the issues encountered with some other very popular downscaling- or bias correction methods (e.g. lack of variance for pure transfer functions or physical inconsistency that easily occurs with

quantile mapping and related techniques). There are a couple of issues though that I think should be addressed before publication. Some of them might be just a matter of clarification, but some might be more fundamental depending on the intended use of the method. These issues are outlined in the following.

**1.2 What is the intended use of the method?**

In the introduction you mention regional climate studies of present, past and future climate as well as numerical weather prediction (NWP) but without being very clear for which of these cases SCAMP is actually made for. Given that you downscale from 1.125 degree resolution to a 8km grid I suppose that SCAMP is not designed to do NWP, given that the ECMWF global deterministic model runs at 9km resolution and most national weather services in Europe operationally run limited area models at 1-2km resolution and limited area ensembles at 2-10km resolution. If however that is the intended use, please explain in which context and for which users you think it could be useful. What made me doubting that SCAMP is intended for regional climate studies, is the use of the word "prediction" throughout the paper. If the intended use are regional climate studies, I would recommend to either use "simulation" rather than "prediction" or to precisely define what "prediction" means in this context. The same applies to section 3.4.

**1.3 Manuscript organization and conciseness**

1. The introduction is to my mind rather long and could be written more concisely. In addition it should contain some more precise statement on the intended use of SCAMP (see section 1.2).

2. I don't understand why the description of the analog stage (stage 1, section 3.2 and 3.3) comes after the description of the GLM stage (stage 2, section 3.1).

In my view this should be reversed. The first part of section 3 (page 5) should contain a concise outline of SCAMP. There is a start at page 5 line 10-12 that should be completed with one or two sentences on the backup model.

3. The last paragraph of section 3.2 could go in a tightened section 3.3 as well (The AM as benchmark and backup model). Its last two sentences are already a very concise summary of section 3.3.

4. I wonder about sections 2 and 4.1 as well: I found it somewhat difficult to figure out which potential predictors were actually used during the first read. There are a few things said in section 2, during section 3 things are quite vague (concerning predictors) and only in section 4.1 things became more clear. If you consider 4.1 to be a central result of the study the information in this subsection should be split into a "methods part" right after or included in section 2 and a "results part" remaining in section 4. If this is not the case I'd suggest to entirely include section 4.1 after or into section 2, but rewritten (together with section 2 from the fourth paragraph on, page 4 line 17 et seq.) in a much more concise manner. For example saying first what you used in the end and then concisely explain why. I think this would allow to be more specific and to use more precise wording in section 3. With a more clear structure lengthy transitions, such as the page 5 last sentence or page 11 lines 4-6, might not be necessary any more.

**1.4 Language issues**

Please check your paper thoroughly for language/grammar issues during the revision, especially

1. tenses
   - stick to simple past for things you did

- avoid future tense for things you finally did, otherwise it induces unnecessary doubt.

2. reduce the use of modal verbs (may, could etc.) where possible in order to be more precise and quantitative.

3. prepositions

4. word order in the context of adjectives and adverbs

5. remove superfluous adverbs for more clarity

6. add missing definite articles

7. mind French to English translation pitfalls

See the technical correction section for examples.

**2  Specific comments**

1. Is SCAMP an abbreviation for something? (I'm just curious)

2. In the introduction (first paragraph) SDM and post-processing are used synonymously. Are they? And if yes, in which context?

3. some references seem slightly out of context. For example:

   (a) Page 2 line 12-13: Maraun et al 2010 review paper already cited at Page 1 line 25

   (b) Page 2 line 28: Citation Maraun et. al 2010: please cite something more specific in this context.

(c) Page 7 line 2: Radanovics et al., 2013: isn't this one more on predictor domains?

4. Page 3 line 24-26: The last sentence of the paragraph is unclear. Please rewrite.

5. Do you think that the selected predictors may depend on the data set used, or its resolution? Please comment.

6. Page 4 line 20: How meaningful are quantities describing instability at 1.125 degrees resolution? and related, if the aim is to do downscaling of climate model outputs or reconstructions how well are the instability and humidity variables simulated by these models, and could the quality of this simulations be an issue for SCAMP? Please comment.

7. Page 4 line 20: Be more specific on the predictors used. For example by referring to table 1 here.

8. Figure 1: The caption text is unclear. What is highlighted in black? Is there a reason to use "quantity" and not "amount"? (Same for figures 2 and 9, page 11 line 32, page 13 lines 3, 21,25 and 26, page 15 line 2, page 22 line 6)

9. Page 5: I'd suggest to add "SCAMP" to the section title of section 3.

10. Page 7 line 6: What does "+12h and +24h UTC" refer to? are this lead times? but then UTC is strange, because time differences don't have a time zone. Or does it refer to the time of the day? But if so, for which hour is the simulation?

11. Section 3.2: What is the archive length used for the analog model?

12. Section 3.2: Which period was used for the optimization of the predictor domains? Is it the same as for the simulation in this work? What are the implications?

13. Section 3.3 first line: Please specify briefly what the significance conditions are.

14. Section 3.1: are there discrete values drawn from the Gamma distribution for the final prediction? And if so, how?

15. Page 9: I think it is a good thing to look at the skill with respect to climatology as you do, especially for comparison with other studies or methods, but you could have used the $AM_{25}$ benchmark as $P_\phi$ as well, right? Would that be equivalent to your $\Delta BSS$ or $\Delta CRPSS$? If not, what is the difference and which one should be preferred under which circumstances?

16. In section 4.1 you describe several steps of restrictions applied in terms of the candidate predictors for the sake of robustness and clarity of the article. I appreciate these goals, but at present the description is a bit confusing and it remains unclear which of these restrictions are a feature of SCAMP and would be kept for a general application of SCAMP and which ones aren't and what would be the potential impact on robustness and skill.

17. Page 13 line 8: The phrase is very unclear. Please rewrite.

18. Page 13 second paragraph: What exactly causes the GLM to "fail" in the southeast for the occurrence? Are there not enough wet analogues to estimate the occurrence probability or does it fail the significance test for the parameters? please comment.

19. Page 15 line 2-3: The predictor set optimized for the whole of France? I thought they were optimized for each grid cell and time step. Is this only for this experiment or in general? This is confusing and will hopefully get more clear with a restructured version of sections 2 and 4.1.

20. Page 16 line 20: Please quantify which proportion of days you would consider as "reasonable".

21. Looking at figure 9, I wonder if the high frequency of the $AM_{25}$ model in the south-east might be related to the Gamma distribution being a suboptimal approximation of the precipitation amount distribution in this region. Did you test this?

22. Why is figure 10 a line chart? There is no order in the WPs, is there? I'd recommend to transform this in a series of bar charts (one for each WP). This would further avoid all the colors and line types and thus solve the issue with the invisible (probably yellow) dotted line for R700+H+Occ-1 in a) and R700+T700+W700 in b).

23. Depending on the intended use of SCAMP, the temporal structure of the simulated precipitation might be relevant. I suppose that a detailed analysis of the representation of the annual cycle, the autocorrelation and the interannual variability in both SCAMP and $AM_{25}$ is beyond the scope of the present paper, especially since this is not straight forward for probabilistic simulations, and you might have a look at this in future work, but could you make a statement on the overall variance of the SCAMP simulations as compared to the benchmark and the observations? Typically analog models reproduce the observed variance quite well while deterministic regression models suffer from reduced variance. Since SCAMP is a hybrid model it would be interesting to know which characteristics it "inherits".

24. page 22 lines 16-20: This part is not clear, please rewrite.

25. page 22 lines 28-32: I don't understand what "classically" means in this part. please use some more precise wording.

26. page 22 line 35: This sentence is not clear to me. In what sense is the set of days homogeneous?

27. page 23 line 2: The sentence is not clear to me. Which context? and who leaves room for improvement?

28. Is SCAMP transferable to other regions or countries? To what extent? Under which circumstances would it be necessary/unnecessary to redo the predictor selection? Please comment.

29. It would be helpful to mark or highlight the predictors that were preselected for the occurrence and amount models respectively in table 1.

**3 Technical corrections**

1. page 1 line 4: ...and the regression link are likely to also vary in time. → ... and the regression link are likely to vary in time too.

2. page 1: does SDM stand for statistical downscaling **methods** (abstract) or statistical downscaling **models** (Introduction, e.g., line 24)? Please unify.

3. Page 2 line 1: Perfect Prog → Perfect Prognosis

4. Page 2 line 8-9: Another possible reference in the reconstruction context: Caillouet et al. (2016)

5. Page 2 line 9: weather numerical models → numerical weather prediction models

6. Page 3 line 6: ...a set of parameters is classically estimated for each weather-type... → ...a set of parameters is estimated for each weather-type...

7. Page 3 line 12: could consist → consists

8. Page 4 line 1: The paper structures... → The paper is structured...

9. Page 4 line 1: You forgot section 3 in the list.

10. Page 4 line 5: metropolitan French territory → mainland of France (also Page 10 line 11. "metropolitan territory" could be misunderstood as areas occupied by large cities)

11. Page 4 line 10: 6-hour → 6-hourly

12. Page 4 line 12: predictors are 1000 and 500 HGT geopotential fields... → predictors are the 1000hPa and 500hPa geopotential height fields...

13. Page 4 line 19: ...the water atmosphere content... → the atmospheric water content (or total column water content?)

14. Page 4 line 20: we also consider the occurrence of precipitation for the previous day. → we also considered the occurrence of precipitation on the previous day.

15. Page 4 line 29: it's "dependent on" but "independent of"

16. page 7 line 5: the similarity is assessed by the TWS → the similarity is assessed using the TWS

17. Page 7 line 6: applied to the geopotential 1000 hPa and 500 hPa respectively at +12h and +24h UTC → applied to the geopotential height at 1000 hPa and 500 hPa at +12h and +24h UTC respectively.

18. Page 7 line 7: low and high pressures → low and high pressure systems

19. Page 7 line 12: → A different analog model was thus considered for each of the 8981 SAFRAN grid cells.

20. Page 7 line 15: we use the 100 → we used the 100

21. Page 7 line 15: identified with AM → identified with the AM

22. Page 7 line 17: we will also consider → we also consider

23. Page 7 line 20: This analog model ... will give a benchmark → This analog model ... is used as a benchmark

24. Page 7 line 20-21: They will be additionally used → In addition they were used

25. Page 8 line 12: and possibly benchmark → and benchmark

26. Page 6 line 3: pi is modeled → pi was modeled

27. Page 6 line 6: we use a GLM → we used a GLM

28. Page 6 line 7: is here expressed as → is therefore expressed as

29. Page 6 line 32: $F_Y$ is finally obtained → $F_Y$ is obtained

30. Page 8 line 19: to evaluate some Ensemble Prediction System → to evaluate Ensemble Prediction Systems

31. Page 9 line 3: y is here practically described → y is here described

32. Some more in section 4.1

33. Page 11 line 21: Figure 4a similarly presents → Figure 4a shows

34. page 11 line 26: Similarly to what was obtained with the BSS gain → similarly to the BSS gain

35. Page 11 line 30: dependency to regional features → dependency on regional features

36. Page 12 line 1: no apostrophe in SCAMPs behavior. (Same page 13 line 27)

37. Page 12 line 2: I don't understand "to updating" in this context.

38. Page 13 line 6: predict occurrence probability → predict the occurrence probability

39. page 13 line 11: it is met → it occurs

40. page 13 line 19: the proportion ... is next too low → the proportion ... is therefore too low

41. page 15 line 6: not necessary → not necessarily

42. page 16 line 2: we assess the frequency each structure has been selected → we assess how often each structure has been selected

43. page 16 line 3: This allows us to give some insight → this allows for some insight

44. page 16 line 3: information really used for the regression stage → information used in the regression stage

45. page 16 line 13: region depend → region dependent

46. page 16 line 20: This would be however not necessary relevant or desirable." Strange sentence, that could be deleted starting the next sentence with "However" for example.

47. page 19 line 23: For some few frequent WPs → for some less frequent WPs

48. page 19 line 24: is much more selected → is more frequently selected for WP7

49. page 19 line 26: for this really few frequent WP → for this rare WP7

50. page 19 line 29: less frequent WPs as WP7 → less frequent WPs such as WP7

51. page 19 line 29: this suggests the spatial robustness → this suggests spatial robustness or this suggests some spatial robustness

52. page 20 line 2: The interest of a hybrid model → the relevance? of a hybrid model

53. page 20 line 3: analogs of the prediction day are identified to estimate a two-part regression model → analogs of the prediction day were identified to estimate the parameters of a two-part regression model

54. page 20 line 5: this hybrid approach actually updates → this hybrid approach updates

55. page 22 line 1: that would have been achieved directly with a → from a

56. page 22 line 4: compared to the prediction that could be directly obtained from the atmospheric analogs of the analog stage → compared to the reference analog model

57. page 22 line 5: CRPSS skill scores → CRPSS

58. page 22 line 6: thanks to → due to

59. page 22 line 8-9: seems also to be decisive → seems to be decisive as well

60. page 22 line 9: twice higher then → either it is "twice as high as" or "two times higher than" which is by the way equivalent to "three times as high as"

61. table 1: baroclinicity → baroclinity

62. Make sure that especially the figures 2, 7, 8, 9 and 11 are rendered large enough in the final version, that is at least not smaller than now.

**References**

CailLouet, L., Vidal, J.-P., Sauquet, E., and Graff, B. (2016). Probabilistic precipitation and temperature downscaling of the twentieth century reanalysis over france. *Climate of the Past*, 12(3):635–662.

---

## Referee Comment (RC2) · Anonymous Referee #2 · 1 Jun 2017

**Big Picture**

The authors present and explore a methodology to simulate precipitation intensities. Yet, neither time series and/or spatial fields of simulated precipitation intensities are shown nor compared to observations (in a probabilistic manner as the title might suggest). While the methodology might be beautiful, I think this is the biggest missing thing in this paper.

I am not a specialist in analog methods. I did my best to understand what is done here. Ideally my potential failings help to detect shortcomings in the paper and lead

to improvements. Besides the analog part, I tried to help with general statistical - hydrological comments.

**"Hybrid" Approach**

The authors want to predict a variable (e.g., precipitation) for a given day (say, for the example of this review, May 30th 2018) at a given location (within France). Then they look at all 30-Mays in the past when precipitation amounts were recorded.

- Where exactly do the authors look? - at the closest measurement station? Is an interpolation performed? What kind of spatial dependence between observations (and simulated values) is assumed / considered?

- On p22 l1ff you write that "the predictors and regression coefficients of the regression models vary from one day to the other? – How much do they vary? And how much do they vary in neighbouring cells? Is there some kind of relationship between the variations in neighbouring cells? Can you show this?

- What if the observed time-series is not stationary? Are there any checks performed? Is stationarity assumed? How strong of an assumption is it?

- The authors claim that values outside the range of observations can be simulated via "extrapolation" (p2 line 20ff.) – some background / assumptions / limitations of this extrapolation methodology is required.

- The previous statement seems contradictory to what is said on p2 lines 29ff.:

- the author's method is able of extrapolation? - is there any evidence of the quality of the extrapolation?

- p2 line 28: I am not sure how a linear model can be "extended" to non-Gaussian data. If this is not to be a reference to what Maraun et al. (2010) did, but the authors rather claim that their method is capable of simulating non-Gaussian data, then there is some more extensive explanation required: What kind of non-Gaussian-ness is observed in

the data and how can linear models mimic this kind of non-Gaussian data? How and where is this non-Gaussianness seen in the data and how is the model describing it?

- From the abstract it did not become clear to me, what is meant with an _hybrid_ ("having two kinds of components that produce the same or similar results") approach – the title is worded more suitably. On the other hand "local" could be confused with "small scale"

**Setup and Language**

At various places within the paper (see comments below) parts of the methodology are explained. I suggest that the introduction is reworded and a section of the introduction is established that clearly and concisely explains what is done in one paragraph. This should also include an explicit statement of the goal and the novelty of the research.

**Major Comments**

**Section 2 - Data**

- Here, there is a distinction between "analog stage" and "regression stage" – are these two stages what is mean when the authors refer to as a _hybrid_ approach? This gets back to my original question: In the analog stage, are the authors looking for all May-30's in the past or only those May-30's where the pattern of the geopotential field was similar on the May-29's? How was this similarity determined?

- why 13 predictors? Is this enough? For what goal?

**3 - The hybrid analog/regression model - the approach of using a distribution function with a portion of zeros is clear.**

- what is not so clear, is how the parameters are estimated and why this is treated independently?

- should the amount of precipitation not be a random variable drawn from the distribution depicted in Figure 1? - It could then be either zero or some precipitation intensity

other than zero.

- why is \pi estimated separately from the parameters of the distribution function? (I am assuming parameters, even though Figure 1 suggests the use of an empirical distribution) Can those parameters not be estimated jointly?

- now, it seems like currently \pi is estimated via a GLM, which seems to be an improved multiple regression with the secondary variables going into xˆo (Eq.2).

- it is not clear what the difference between superscript o and superscript q is in Eqs 2 and 3.

- How does the Gamma distribution come into the game? Are you using this type of distribution to model the non-zero part of the distribution? Why Gamma? Also, the logic in p6 lines 13,14 is off. I think you should use a distribution that fits somewhat well to the data and then fit its parameters to the data.

- what determines how "near" an analog is to the predicted day? (likely this is answered in Sect. 3.2).

- why is the threshold for precipitation 0.1mm?

- p6 lines 23 ff. are difficult to understand. Say again you are trying to predict May-30 2018 in one grid location of France. Then you are searching for the "nearest" geopotential conditions for all May-29 in the past and then estimate \pi based on the precipitation occurrences in those days. The "nearest conditions" could be different for a neighbouring cell? What does this say about consistency and spatial dependence structure of precipitation fields. Also for Jun-1 2018, again a potentially very different set of "nearest conditions" could be used? Or am I understanding this wrongly, and there are more constraints?

- Why are you using the BIC (and not another criterium)?

- I would suggest a more careful wording when the word "significance" is employed.

Arguably, a predictor can be significant at a certain level, but not plainly not significant (p6 line 26ff) – what level of significance did you choose?

- p8, l21 you start to use a differently typeset "P" after the abbreviation "ESP" – please explain.

- Figure 2: top right panel: should there not be dots on the black line? At least for the part "within" \pi?

**4 - Results**

- p11 l12ff: you write that the BSS gain is "very sensitive to topography". The coasts along the Mediterranean (E portion of southern coast of France) and the Atlantic (W portion of northern coast of France) have opposite BSS gains (Fig 3b). How does that fit to your explanation?

- p11 l32: what do you mean by "greatly and thus significantly"?

**5 - Discussion**

Generally, this section reads as a strung together explanation of what is shown on several figures. What does it mean remains more unclear than the authors probably think...

- can the selection of structures (what is visualised by Figures 8 and 9) be done in a more quantitative way (contribution of each variable to the prediction)?

- p19 l1: Please describe first what your point is, then what is visualised on Figure 10).

- p22 l4ff: you write that the gain is "non-negligible". Then you write that it is "up to 0.1" – can you quantify how much of a gain this really is?

**Minor Comments**

- p 16, last word: necessarily

- p2 line 35: remove "obviously" or explain how this is obvious.

- throughout the paper: frequent use of "classic" and its funnily sounding adverb. What is classic in the sense of analog hydrological methods?

- The authors mention multiple times (e.g. p3 l7, p3 l14) a relation of the presented methodology to physical (maybe deterministic?) - Is the goal of the presented approach to be "physically realistic"?

- How / in what sense does this lead to something "more relevant and robust" (p3 l14/15)?

- Table 1: H, W, PV: of what variables?

- p11 l4: "two" or "four" or something else?

- p13 l19: what does "next too low" mean?

- Style: there are many abbreviations, and it's easy to forget what they all mean.

- sometimes you write "metropolitan French territory", sometimes "France". It seems like you never looked at Paris or the major cities specifically, hence I suggest to use "France" everywhere.

- p20 l2: I don't think the interest has been explored. Rather, the model itself has been explored?

---

## Author Comment (AC1) · 29 Jun 2017

[hess, manuscript]copernicus

*We thank the referee for this thorough review and for the numerous constructive suggestions that we will consider for incorporation in the modified manuscript. We give here the detailed responses to all his comments and questions.*

**1 Topic and general comments**

**1.1 Topic**

The paper presents a new two-stage hybrid perfect prognosis SDM called SCAMP. SCAMP was applied to a large number of grid points in France and was proven to be adaptive to different weather types and seasons which is illustrated nicely with visually appealing figures. The method seems very interesting given the issues encountered with some other very popular downscaling- or bias correction methods (e.g. lack of variance for pure transfer functions or physical inconsistency that easily occurs with quantile mapping and related techniques). There are a couple of issues though that I think should be addressed before publication. Some of them might be just a matter of clarification, but some might be more fundamental depending on the intended use of the method. These issues are outlined in the following.

**1.2 What is the intended use of the method?**

In the introduction you mention regional climate studies of present, past and future climate as well as numerical weather prediction (NWP) but without being very clear for which of these cases SCAMP is actually made for. Given that you downscale from 1.125 degree resolution to a 8km grid I suppose that SCAMP is not designed to do NWP, given that the ECMWF global deterministic model runs at 9km resolution and most national weather services in Europe operationally run limited area models at 1-2km resolution and limited area ensembles at 2-10km resolution. If however that is the intended use, please explain in which context and for which users you think it could be useful. What made me doubting that SCAMP is intended for regional climate studies, is the use of the word "prediction" throughout the paper. If the intended use are regional climate studies, I would recommend to either use "simulation" rather than "prediction"

or to precisely define what "prediction" means in this context. The same applies to section 3.4.

*A widely used argument for the development of statistical downscaling models (SDMs) is that they allow producing local scale weather scenarios. We obviously agree that high resolution ensembles are operationally available from most national weather services. SCAMP would not be of any interest in respect to this point.*

*As mentioned in the manuscript, another important argument for the development/use of SDMs is that the outputs of GCM and/or NWP models are generally 1) biased and, from a statistical point of view, 2) not reliable (the ensembles are often underdispersive – (see for instance, Leutbecher and Palmer, 2008). In a number of cases however, impact studies require unbiased and reliable meteorological scenarios. This is for instance a critical requirement for hydrological impact studies as a result of the strong linearities in the hydrological response of river basins to meteorological forcings.*

*In the present work, we did not select a given context for the application of SCAMP. SCAMP could be used for either forecasting, reconstruction or simulation. We will precise this in the new manuscript version. Some specific requirements would apply for each context. For instance, the temporal transferability of the model in a modified climate context would be required for the development of climate projections. The quality of large scale predictors would have to be checked for reconstructions over the XXth century or for climate prediction (as often reported, thermodynamic predictors are of lower quality than dynamic ones – see questions + responses to 'specific comments' 5 and 6 below).*

*We will precise what the word "prediction" means in this context. We find this word more suited than "simulation" because this latter suggests that times series of precipitation are produced. This is here not the case (although some postprocessing generation process could be used for this but this is out of the scope of the present*

*work) as we issue for each day the statistical distribution of precipitation amount (thus a probabilistic prediction).*

1.3   Manuscript organization and conciseness

1. The introduction is to my mind rather long and could be written more concisely. In addition it should contain some more precise statement on the intended use of SCAMP (see section 1.2).

*We will adapt the introduction as suggested and clarify the intended use of SCAMP.*

2. I don't understand why the description of the analog stage (stage 1, section 3.2 and 3.3) comes after the description of the GLM stage (stage 2, section 3.1). In my view this should be reversed. The first part of section 3 (page 5) should contain a concise outline of SCAMP. There is a start at page 5 line 10-12 that should be completed with one or two sentences on the backup model.

*We thank the referee for these suggestions. We will complete the outline of SCAMP and we will describe the analog stage before the GLM one as suggested.*

3. The last paragraph of section 3.2 could go in a tightened section 3.3 as well (The AM as benchmark and backup model). Its last two sentences are already a very concise summary of section 3.3.

*The Analog Model can be indeed presented as a benchmark and backup model. We will merge the last paragraph of section 3.2 with section 3.3 as suggested.*

4. I wonder about sections 2 and 4.1 as well: I found it somewhat difficult to figure out which potential predictors were actually used during the first read. There are a few things said in section 2, during section 3 things are quite vague (concerning predictors) and only in section 4.1 things became more clear. If you consider 4.1 to be a central result of the study the information in this subsection should be split into a "methods part" right after or included in section 2 and a "results part" remaining in section 4. If this is not the case I'd suggest to entirely include section 4.1 after or into section 2, but rewritten (together with section 2 from the fourth paragraph on, page 4 line 17 et seq.) in a much more concise manner. For example saying first what you used in the end and then concisely explain why. I think this would allow to be more specific and to use more precise wording in section 3. With a more clear structure lengthy transitions, such as the page 5 last sentence or page 11 lines 4-6, might not be necessary any more.

*We thank the referee for these different suggestions. As suggested we will include section 4.1 into section 2 and modify the text and transitions consequently. This will make indeed the paper more clear.*

1.4   Language issues.

Please check your paper thoroughly for language/grammar issues during the revision, especially

1. tenses stick to simple past for things you did avoid future tense for things you finally did, otherwise it induces unnecessary doubt. 2. reduce the use of modal verbs (may, could etc.) where possible in order to be more precise and quantitative.

3. prepositions

4. word order in the context of adjectives and adverbs

5. remove superfluous adverbs for more clarity

6. add missing definite articles

7. mind French to English translation pitfalls

See the technical correction section for examples.

*We will carefully check for these different issues and a native English person will read the paper. Thank you for these recommendations.*

**2  Specific comments**

1. Is SCAMP an abbreviation for something? (I'm just curious)

*In the previous work of Raynaud et al. (2016), we first worked on a multivariate Analog version, for multivariate prediction (precipitation, temperature and radiation). SCAMP is the abbreviation defined in this previous work and stands for Sequential Constructive atmospheric Analogs for Multivariate weather Prediction. We kept this abbreviation for the present work even if we are in a monovariate configuration. This will be clarified.*

2. In the introduction (first paragraph) SDM and post-processing are used synonymously. Are they? And if yes, in which context?

*SDM and post-processing are sometimes rather synonymous for instance when SDM are used to produce local weather scenarios from GCM output data (their ability to do some bias correction is an important feature here as mentioned previously). We agree that this is not always the case. Other applications of SDM are possible as those mentioned in the second paragraph of the introduction (weather generation, climate change attribution...). We will reformulate the text to avoid the confusion.*

3. some references seem slightly out of context. For example:

(a) Page 2 line 12-13: Maraun et al 2010a review paper already cited at Page 1 line 25

*This reference has been indeed already mentioned in a previous paragraph and will be removed here.*

(b) Page 2 line 28: Citation Maraun et. al 2010b: please cite something more specific in this context.

*As mentioned by referee 1, we agree that a more specific reference can be given in this context. Moreover these lines were somehow confusing. Generalized Linear Models (GLM) are regression models specifically introduced by statisticians to model non-gaussian data (see Nelder and Wedderburn, 1972). They were first used by Stern and Coe (1984) for the generation of precipitation. The vector generalized linear models (VGLM, Yee and Wild 1996), closely related to the class of GLMs, are the most general class of linear regression models available. The work of Maraun et al. (2010b) is "just" one recent application of VGLMs for the case of precipitation. We will simplify this section and remove the mention to VLGMs, which is not necessary here.*

(c) Page 7 line 2: Radanovics et al., 2013: isn't this one more on predictor domains?

*Radanovics et al., 2013 is indeed one work focusing on France where the issue of the predictor domains has been explored. The iterative process followed to identify the best predictor domain is however slightly different from that used in the present work. It is repeated 5 times with different initial conditions whereas in the present work, we do it only one. This is the reason why we did not mention it for this domain optimization issue.*

4. Page 3 line 24-26: The last sentence of the paragraph is unclear. Please rewrite.

*Our point is that the type of model used in the work of Ibarra-Berastegi et al. (2011) is not really optimal. A linear regression model is indeed not suited to the non-gaussian nature of precipitation amounts. The approach of Ibarra-Berastegi et al. (2011) would thus benefit from using a model suited to precipitation. We will reformulate the sentence to make it clear.*

5. Do you think that the selected predictors may depend on the data set used, or its resolution? Please comment.

*Different studies have shown that the predictors depend on the predictand. For precipitation, predictors can differ from one location to the other (e.g. Cavazos and Hewitson, 2005; Timbal et al., 2009; Chardon et al., 2014). They are also not necessary the same for precipitation, radiation or other surface weather variables (e.g. Raynaud et al. 2016). We could also expect that the predictors depend on the dataset used, for the atmospheric reanalyses especially. To our knowledge this analysis has not been carried out yet. Some dependence to the resolution is also probably to expect. A higher*

*resolution would definitively allow for a better description of the shapes of geopotential fields. It would also allow for a more relevant simulation of thermodynamic processes. It would likely lead in turn to have higher quality variables for some quantity such as air instability (as mentioned in the following question). The quality of simulation does however not only depend on the resolution of reanalyses but also on the quality of the model and of the observed data available for assimilation. We could thus expect that data with higher resolution do not necessary always lead to better quality predictions.*

*These issues are obviously very interesting and would be worth specific analyses in the future. A comment will be introduced in the perspectives of the modified manuscript.*

6. Page 4 line 20: How meaningful are quantities describing instability at 1.125 degrees resolution? and related, if the aim is to do downscaling of climate model outputs or reconstructions how well are the instability and humidity variables simulated by these models, and could the quality of this simulations be an issue for SCAMP? Please comment.

*We agree that atmospheric variables describing instability do not give a very good picture of instability when available atmospheric variables are at $1.125°$ resolution. To our opinion however, they can have some predictive power as a "proxy" of the instability.*

*The quality of such predictors in climate model outputs or reconstruction is obviously an issue. When applied in a reconstruction context for the whole XXth century, we indeed found that the added value of such predictors was much smaller than when applied with the recent reanalyses available for the last decades. We will add a comment in the discussion on this issue.*

7. Page 4 line 20: Be more specific on the predictors used. For example by referring

to table 1 here.

*Thank you for the suggestion. We will precise the text as suggested.*

8. Figure 1: The caption text is unclear. What is highlighted in black? Is there a reason to use "quantity" and not "amount"? (Same for figures 2 and 9, page 11 line 32, page 13 lines 3, 21,25 and 26, page 15 line 2, page 22 line 6)

*We have used "quantity" to specifically refer to the positive values of precipitation. "amount" is often used to describe all precipitation values, including zeroes. Using "quantity" allows us to make a rather clear link between the "quantity model" and the non-zero amounts it is used for. For us, a "precipitation amount" model would have been more confusing as it could have referred to both the zero amounts and the non-zero amounts.*

*In figure 1, we use equation 1 to decompose $F_{AM}(y)$ into two parts. We highlight in black the contribution of the empirical cdf of the non-zero precipitation amount to the overall cdf. As expressed in equation 1, this last pdf is "weighted" by the probability of occurrence of probability. We will reformulate the caption of Figure 1 to make it clearer.*

9. Page 5: I'd suggest to add "SCAMP" to the section title of section 3.

*This will be done.*

10. Page 7 line 6: What does "+12h and +24h UTC" refer to? are this lead times? but then UTC is strange, because time differences don't have a time zone. Or does it refer to the time of the day? But if so, for which hour is the simulation?

*For any given day, SAFRAN precipitation amounts correspond to the cumulated precipitation from 6h00 UTC of the day to 6h00 UTC of the next day, or in other words they correspond to precipitation from 6h00 UTC to 6h00 UTC +24h (where UTC stands for Coordinated Universal Time). We acknowledge that the times used for the geopotential were confusing. We should not have used the "+" symbol. The atmospheric circulation of the current day is described with the geopotential fields at two different UTC times, namely 12h UTC and 24h UTC of that day. The two fields retained to compare days and identify circulation analogs are thus centred over the temporal window used to determine daily precipitation. We will correct the text accordingly*

11. Section 3.2: What is the archive length used for the analog model?

*The archive length is 1982-2001 which corresponds to the period considered for the predictions with SCAMP and / or the backup AM25 model. We will precise it in the revised version*

12. Section 3.2: Which period was used for the optimization of the predictor domains? Is it the same as for the simulation in this work? What are the implications?

*The period used for the optimization of the predictor domains is 1982-2001. The period is thus the same as the period of the predictions. The prediction skill of SCAMP presented in our manuscript may therefore be slightly overestimated.*

*To assess the influence of the optimization period, we could have followed a leave-one-out approach, where for instance, the best analogy domain would have been identified from all years except that of the current prediction day. This would have required*

*much larger computing resource than those already used for the work presented in the manuscript. The process used for the optimization of the analogy domain is indeed rather long (as mentioned in the manuscript, it is first iterative where different spatial domains of increasing size and considered in turn. The identification of the best analogs days for a given analogy domain is also rather time consuming as a result of the similarity criterion used to compare days). It had also to be applied successively for the 8,981 grid cells of France. This optimization plus the re-estimation for each prediction day of the different regression models considered in the regression stage of SCAMP already required the use of the Grenoble University High Performance Computing centre CIMENT (https://ciment.ujf-grenoble.fr/wiki-pub/index.php/). A leave-one-out approach would have required too many computing resource and was thus not applied here.*

*We agree that the optimal domain may depend on the period used for the optimization of the method. We however expect that the domains would be rather similar when obtained from different periods and that their influence on the main results we present in our work would be limited. In a recent work carried out with an Analog Model similar to AM25, we have actually shown that slightly different domains may lead to identify – for a given prediction day, rather different sets of analog dates. We have however also shown that this does not lead to a significant difference in the prediction skill (Chardon et al., 2014). For the context of the present manuscript, an interesting work would be to explore if analogs from different but similar analogy domains would influence the choice of the predictors in the regression stage and/or also if the coefficients of the regression would change. This could contribute to assess the robustness of the approach. We will mention this perspective work in the discussion.*

13. Section 3.3 first line: Please specify briefly what the significance conditions are.

*We used the 5% significance level for each predictor. This information will be added in the new version of the paper.*

14. Section 3.1: are there discrete values drawn from the Gamma distribution for the final prediction? And if so, how?

*We aim to model the distribution of precipitation amount, its day dependency, and to further use this distribution as probabilistic prediction. For any given prediction day, we do thus not draw some realization from the distribution.*

15. Page 9: I think it is a good thing to look at the skill with respect to climatology as you do, especially for comparison with other studies or methods, but you could have used the AM25 benchmark as P as well, right? Would that be equivalent to your BSS or CRPSS? If not, what is the difference and which one should be preferred under which circumstances?

*Yes, we could have chosen the AM25 model as reference for the evaluation of the combined model. We have preferred to use the climatological reference as this allows for normalized scores which can be compared, as mentioned by the referee, with those obtained in other studies.*

*We compared the combined approach and the AM25 model with gains in skill scores estimated for both approach with respect to the climatology. Such gains, given in terms of BSS or CRPSS percentage points, are also widely used to compare different prediction models. They thus present the advantage to be rather easy to interpret.*

16. In section 4.1 you describe several steps of restrictions applied in terms of the candidate predictors for the sake of robustness and clarity of the article. I appreciate these goals, but at present the description is a bit confusing and it remains unclear

which of these restrictions are a feature of SCAMP and would be kept for a general application of SCAMP and which ones aren't and what would be the potential impact on robustness and skill.

*The issue of predictor restrictions is an interesting point which was not easy to tackle. The main goal of those restrictions was indeed to improve the clarity of the article. The manuscript does thus not present a definitive configuration of SCAMP but more a proof of concept for an adaptive model which could use a much larger set of potential predictors, when relevant.*

*The impact of fewer restrictions on the robustness of the method is potentially an important issue and would be worth a detailed analysis. This will be suggested as a perspective of the work.*

17. Page 13 line 8: The phrase is very unclear. Please rewrite.

*Thanks for this remark. We will rewrite the phrase.*

18. Page 13 second paragraph: What exactly causes the GLM to "fail" in the southeast for the occurrence? Are there not enough wet analogues to estimate the occurrence probability or does it fail the significance test for the parameters? please comment.

*We agree that the text of the second paragraph of p. 13 is somehow confusing. As mentioned in the paragraph, the GLM which models the occurrence does actually almost never fail, even in the southeast. From the sum of the frequencies obtained for the two cases "case 2" and "case 4" of Figure 5, we can see that, whatever the region, the GLM which models the occurrence is indeed activated most of the time (more than*

*97% of the days) (remember that case 2 corresponds to (Success of GLM modeling the occurrence + Failure of GLM modeling the quantity); case 4 corresponds to (Success of GLM modeling the occurrence + Success of GLM modeling the quantity). Consequently, and whatever the region, the situation where AM25 is used as backup for the prediction of the occurrence probability is very rare (see the sum of the frequencies obtained for case 1 and case 3).*

*Nevertheless, Figure 5 indeed highlights a very specific behavior in the southeast when compared to the remaining of France. Case 2 is activated much more often in this region (increase of 30% percentage point) than elsewhere and, in a symmetric way, Case 4 is activated much less often than elsewhere (decrease of 30% percentage points).*

*The reason underlying this result is to be related to the much higher proportion of dry days in southeast as illustrated in figure R1a below. For a number of predictions days, the number of analog days that are wet is indeed to be small in the southeast. This is obviously not a difficulty for the estimation of the GLM modeling the occurrence. This is conversely likely one for the estimation of the GLM modeling the quantity. For days for which the number of wet analogs is small, the size of the dataset available to fit the GLM modeling the quantity can be too small to allow for a fit with significant parameters. This very likely explains the spatial disparities in both graphs "Case 2" and "Case 4" of Figure 5.*

*In our work, a GLM (GLM modeling the occurrence or GLM modeling the quantity) was said to fail for a given prediction day when the significance test failed for the parameters. The link of these failures with the number of wet/dry analog days is not as direct as we could have presented it in the text. It is however strongly suggested from Figure R1a and from what is described in the different graphs of Figure 6. We will reformulate this section for clarification.*

Figure R1a. Probability of a dry days over the 1982-2001 period (percentage of days which are dry) (see supplementary material)

19. Page 15 line 2-3: The predictor set optimized for the whole of France? I thought they were optimized for each grid cell and time step. Is this only for this experiment or in general? This is confusing and will hopefully get more clear with a restructured version of sections 2 and 4.1.

*We agree that the text is somehow confusing. For the sake of clarity, a single set of potential predictors was used for all grid points of France (see section 4.1). The most interesting set may be however rather different from one region to the other. This is a possible reason for which we obtain no gain in southern France when we activate the quantity model. This should become clearer with the restructured version of the manuscript.*

20. Page 16 line 20: Please quantify which proportion of days you would consider as "reasonable".

*We had actually not in mind to suggest a "reasonable" proportion of days, which could be used to retain a reduced number of regression sets. Considering a reduced number of regression sets would obviously allow for reducing the computational time required for the model identification/evaluation.*

*As mentioned in the next sentence of the manuscript, this may however limits the possibility to achieve a better prediction for some (rare) events which would activate very unusal predictors. This is what is highlighted with some of the graphs in the discussion section. We will clarify this point in the revised manuscript version.*

21. Looking at figure 9, I wonder if the high frequency of the AM25 model in the southeast might be related to the Gamma distribution being a suboptimal approximation of the precipitation amount distribution in this region. Did you test this?

*Thank you very much for this input. The distribution of non-zero precipitation amounts is indeed rather different in the southeast region than that of the other regions of France. The main reason is the existence of much more frequent and intense heavy precipitations in southeastern France (this is the reason why a lot of works have focused in the last decades on precipitation extremes in southeastern France). This is suggested with the much higher Coefficient of Variation of daily precipitation in this region (cf. figure R1b below).*

Figure R1b. Variation coefficient of daily precipitation (see supplementary material)

*The gamma distribution is obviously flexible and widely used in the hydro-meteorology literature to model strictly positive precipitation. It may be however not optimal in this specific context and a distribution with a heavy tail would be probably more appropriate (e.g. the extended GPD distribution introduced by **?**). This may improve the prediction but this may also lead to estimation difficulties as a more complex distribution would require more data for a robust fit. Here, the number of data is voluntarily limited to the number of analog dates considered for the fit. A more flexible model (with more parameters) would require considering more analog dates. This may be detrimental for the prediction skill as poorer analogs would be integrated in the set of dates used for the estimation of the regression. Another possibility would be to fit a more flexible model with the same number of analog dates than in the present case. Due to the estimation problems mentioned above, this would however likely lead to a much more frequent use of the backup analog prediction model in our case. Such an analysis would be worth to do and this issue will be mentioned in the perspective section.*

[Figure]

22. Why is figure 10 a line chart? There is no order in the WPs, is there? I'd recommend to transform this in a series of bar charts (one for each WP). This would further avoid all the colors and line types and thus solve the issue with the invisible (probably yellow) dotted line for R700+H+Occ-1 in a) and R700+T700+W700 in b).

*We agree that a series of bar charts would have been more relevant and we will change the figure for this representation.*

23. Depending on the intended use of SCAMP, the temporal structure of the simulated precipitation might be relevant. I suppose that a detailed analysis of the representation of the annual cycle, the autocorrelation and the interannual variability in both SCAMP and AM25 is beyond the scope of the present paper, especially since this is not straight forward for probabilistic simulations, and you might have a look at this in future work, but could you make a statement on the overall variance of the SCAMP simulations as compared to the benchmark and the observations? Typically analog models reproduce the observed variance quite well while deterministic regression models suffer from reduced variance. Since SCAMP is a hybrid model it would be interesting to know which characteristics it "inherits".

*In a context where time series have to be simulated, additional criteria would be actually relevant to evaluate / compare the different modelling approaches; they should especially include as mentioned by the referee, the ability to reproduce the observed variance of precipitation from daily to interannual time scales.*

*We will add a comment on this in the discussion section.*

*Deterministic regression models are indeed known to underestimate the variance of*

*daily precipitation. However, scenarios obtained with deterministic regression models disregard the variance of the residuals in the regression. However, regression models can be also used in a stochastic simulation framework, where a random variable is drawn from the statistical distribution associated to the regression. In ? for instance, such a generation process was used, to first identify if the prediction day was wet or dry (based on a random variable compared to the occurrence probability obtained from a first occurrence model) and to next generate some precipitation amount (based on a random realization within the gamma distribution used to model the distribution of precipitation amount in case of a wet day). The observed variance of daily precipitation was well reproduced.*

*If time series would have to be simulated with SCAMP, a similar stochastic generation process would be followed for the regression stage. We thus would expect that the variance of observed daily precipitation would be rather well reproduced as in ?. The regression stage is also not expected to increase the variance that would be obtained for the benchmark analog (in a configuration where one of the k-nearest analogs is randomly sorted each day and used as weather scenario for the day). This is one of the preliminary results we obtained for a similar work we currently develop in western Switzerland.*

24. page 22 lines 16-20: This part is not clear, please rewrite.

*We will reformulate this paragraph.*

25. page 22 lines 28-32: I don't understand what "classically" means in this part. please use some more precise wording.

*The word "often" is indeed more suited there.*

26. page 22 line 35: This sentence is not clear to me. In what sense is the set of days homogeneous?

*The days are homogeneous with respect to their large-scale atmospheric circulation configuration. We will clarify this.*

27. page 23 line 2: The sentence is not clear to me. Which context? and who leaves room for improvement?

*The context we wanted to refer is that of very specific meteorological configurations that may be observed from time to time and for which the usual predictors are sub-optimal. The analog/regression approach presented here is expected to allow for the identification of better suited predictors and thus for an improved prediction skill for the prediction day under consideration. We will rephrase this sentence in order to make it clearer.*

28. Is SCAMP transferable to other regions or countries? To what extent? Under which circumstances would it be necessary/unnecessary to redo the predictor selection? Please comment.

*SCAMP is indeed transferable to other regions. We will add a comment on this in the discussion. The development of SCAMP for another context would obviously first require the identification of the best predictor set. In a number of previous studies, the most relevant predictors to be used for statistical downscaling are indeed found to be region dependent (e.g. Cavazos and Hewitson, 2005; Timbal et al., 2009; Chardon*

*et al., 2014). Similar conclusions have likely to be expected for SCAMP also. Part of these conclusions may also result from the occurrence frequency of weather regimes observed for the different regions. They can actually vary a lot from one region to the other. In such a case, the best predictors identified for a given region could partly be a result of the fact that the weather situations for which they are more relevant are more frequently observed for that region. This issue would be worth future investigations.*

29. It would be helpful to mark or highlight the predictors that were preselected for the occurrence and amount models respectively in table 1.

*Thank you for the suggestion. This will be marked in table 1.*

**3   Technical corrections**

*We thank the reviewer for the very careful reading of the paper and for all the technical corrections pointed out here. The text will be corrected accordingly.*

Cavazos T, Hewitson BC. 2005. Performance of NCEP-NCAR reanalysis variables in statistical downscaling of daily precipitation. Climate Research 28: 95–107.

Chardon J, Hingray B, Favre AC, Autin P, Gailhard J, Zin I, Obled C. 2014. Spatial similarity and transferability of analog dates for precipitation downscaling over France. J. Clim. 27: 5056–5074, doi: 10.1175/JCLI-D-13-00464.1.

Leutbecher, M.; Palmer, T. N. 2008. Ensemble forecasting. JOURNAL OF COMPUTATIONAL PHYSICS. 227(7). 3515-3539.

Maraun, D., et al. 2010a: Precipitation downscaling under climate change: Recent developments to bridge the gap between dynamical models and the end user, Rev. Geophys., 48,RG3003, 20 doi:201010.1029/2009RG000314, 2010.

Maraun, D., Rust, H. W., and Osborn, T. J.: 2010b. Synoptic airflow and UK daily precipitation extremes Development and validation of a vector generalised linear model, Extremes, 13, 133–153, doi:10.1007/s10687-010-0102-x.

Mezghani, A. and Hingray, B., 2009. A combined downscaling-disaggregation weather generator for stochastic generation of multisite hourly weather variables in complex terrain. Development and multi-scale validation for the Upper Rhone River Basin. J. Hydrology. 377 (3-4) : 245-260.

Naveau, Philippe; Huser, Raphael; Ribereau, Pierre; et al. 2016. Modeling jointly low, moderate, and heavy rainfall intensities without a threshold selection. WATER RESOURCES RESEARCH. 52(4) 2753-2769

Radanovics, S., Vidal, J.-P., Sauquet, E., Ben Daoud, A., and Bontron, G.: Optimising predictor domains for spatially coherent precipitation 5 downscaling, Hydrol. Earth Syst. Sci., 17, 4189–4208, doi:10.5194/hess-17-4189-2013, 2013.

Raynaud, D., Hingray, B., Zin, I., Anquetin, S., Debionne, S., Vautard, R. 2016. Atmospheric analogs for physically consistent scenarii of surface weather in Europe and Maghreb. Int. J. Climatology. doi :10.1002/joc.4844.

Timbal B, Fernandez E, Li Z. 2009. Generalization of a statistical downscaling model to provide local climate change projections for Australia. Environ. Model. Software 24(3): 341–358.

Yee, TW; Wild, CJ. 1996. Vector generalized additive models. JOURNAL OF THE ROYAL STATISTICAL SOCIETY SERIES B-METHODOLOGICAL. 58(3) 481-493

[Figure]

**Supplement:**

[Figure]

*Figure R1a : Probability of a dry days over the 1982-2001 period (percentage of days which are dry).*

[Figure]

*Figure R1b. Variation coefficient of daily precipitation*

---

## Author Comment (AC2) · 29 Jun 2017

[hess, manuscript]copernicus

Anonymous Referee 2;; doi:10.5194/hess-2017-62

*We thank the referee for this thorough review and for the numerous constructive suggestions that we will consider for incorporation in the modified manuscript. We give here the detailed responses to all his comments and questions.*

[Figure]

**1 Big Picture**

1. The authors present and explore a methodology to simulate precipitation intensities. Yet, neither time series and/or spatial fields of simulated precipitation intensities are shown nor compared to observations (in a probabilistic manner as the title might suggest). While the methodology might be beautiful, I think this is the biggest missing thing in this paper. I am not a specialist in analog methods. I did my best to understand what is done here. Ideally my potential failings help to detect shortcomings in the paper and lead to improvements. Besides the analog part, I tried to help with general statistical hydrological comments.

*The downscaling model first aims to issue probabilistic predictions of local precipitation at any given grid point of the SAFRAN grid. This prediction results in a probabilistic distribution function for each prediction day for each grid point. A times series representation, where (probabilistic) simulations and observations are compared, is thus not really convenient. The evaluation is here done with the CRPSS, which is frequently used for the evaluation of probabilistic predictions in a framework where we have to compare one value (the observation) with a whole distribution.*

*We also agree that the prediction of precipitation fields is an important issue. It was not in the scope of our work but further work should consider this issue. We will include a comment on this in the discussion section as a perspective.*

**2 Hybrid Approach**

2. The authors want to predict a variable (e.g., precipitation) for a given day (say, for the example of this review, May 30th 2018) at a given location (within France). Then they look at all 30-Mays in the past when precipitation amounts were recorded. Where exactly do the authors look? at the closest measurement station? Is an interpolation performed? What kind of spatial dependence between observations (and simulated values) is assumed / considered?

*In the present work, we want to predict local precipitation only. We do not thus need any assumption on the spatial dependence between stations.*

*For each prediction target location (each target grid box), we only use precipitation data that were estimated within the SAFRAN reanalysis system at this exact location. Note that the SAFRAN reanalysis give an estimate of daily precipitation at each location from the closest measurement stations and from some information on the weather type for each day. To date, these SAFRAN estimates provide the only high resolution reanalysis of local precipitation over France for the 60 past years. We consider these estimates as pseudo-observations.*

3. On p22 l1ff you write that "the predictors and regression coefficients of the regression models vary from one day to the other? – How much do they vary? And how much do they vary in neighbouring cells? Is there some kind of relationship between the variations in neighbouring cells? Can you show this?

*Thank you for this very interesting point. We actually did not estimate these day-to-day variations. We will change this formulation for "the predictors and regression coefficients of the regression models can thus vary from one day to the other". We have*
*actually an indirect estimation of the variation in the predictors with the different sampling frequencies obtained for the different weather types. If not straightforward, a more formal / direct evaluation would be probably worth. This will be suggested as a perspective of the work.*

*We agree that the regional consistency of the predictors is another very interesting issue. An evaluation of this consistency is indeed expected inform on the robustness of the downscaling relationship. From the results shown in the manuscript, we have a partial idea of this with the rather high spatial coherency of the selection frequency obtained for each regression structure considered in our work. This is illustrated in Figures 8, 9 and 11. In Figure 11, this coherency is also shown to be high (even if with different spatial patterns) for different weather types.*

*This spatial coherency is also suggested from other analyses not shown in the manuscript. Figure R2a below presents the percentage of time (over the 20 years used for the analysis) where the regression structure selected for a given location is the same than the regression structure used for a reference location. As illustrated, this percentage of time with same structures - varies from 35% to more than 50% in the close neighborhood of the reference location. Some regional consistency is also found for the regression coefficient obtained for the predictors. We find that the spatial pattern of this consistency depends on the weather situation. This is illustrated in Figure R2b below for the regression coefficient estimated for W in the regression structure no 4.*

*We will add a comment on this in the new manuscript version.*

*Figure R2a: Percentage of time (over the 20 years used for the analysis) where the regression structure selected for a given location is the same than that used for a reference location (the reference location is indicated with the blue dot in each figure). Results are shown for 16 different reference locations. (see supplementary material)*

*Figure R2b: Spatial distribution of the mean regression coefficients obtained for the predictor W and the regression structure n° 4. Results are presented for the different seasons and for the 8 weather types considered in this work. (see supplementary material)*

4. What if the observed time-series is not stationary? Are there any checks performed? Is stationarity assumed? How strong of an assumption is it?

*No hypothesis of stationarity is assumed in the present work. The time series of precipitation can be non-stationary. In such a case, this non-stationarity would be expected to be reproduced as a result for instance of a change in frequency of weather types or as a result of a non-stationarity (i.e. temporal evolution) in some of the predictor variables.*

*An indirect illustration of this is the ability of the method to reproduce the interannual variability of annual precipitation (or of seasonal precipitation). This was illustrated by Lafaysse (2011) for three different downscaling approaches similar to the present one.*

*Figure 10.1 extracted p122 in Lafaysse(2011). Time series of winter (DJF) and summer (JJA) precipitation obtained with an Analog Prediction model (100 scenarios) for the Upper Durance basin in South-Eastern France (red = observation, green : median scenario, blue : first analog scenario) (see supplementary material)*

5. The authors claim that values outside the range of observations can be simulated via "extrapolation" (p2 line 20ff.) – some background / assumptions / limitations of this extrapolation methodology is required. The previous statement seems contradictory to what is said on p2 lines 29ff.:

*In this paragraph, we wanted to highlight the issue of simulating non observed values. This obviously refers to values that are outside the range of observed values (including rare values): this is indeed extrapolation. This also refers to all values that are in-between two consecutive observed ones. In our context, precipitation is estimated as a function of different predictors. For a given prediction day, the statistical model give a prediction which is a kind of interpolation from what has been observed in configurations that are close to the configuration of the prediction day (in terms of predictors). We have mentioned these two different prediction contexts in our initial text but we understand that too much emphasize was given to extrapolation. We will modify it in the new manuscript version.*

*Using a statistical model for this simulation/extrapolation exercise is obviously current practices in statistical analyses. This is actually based on the assumption that the model is well suited for the data. We here follow the same assumption, with the difference that the statistical model is reestimated each prediction day based on what was observed for a set of analogs for this day. The limitations of this simulation/extrapolation are mainly related to the quality of the statistical model used to model the observations. We here consider the gamma model for precipitation amounts, model which has been widely used in the hydrological literature to model precipitation amounts. We have assessed the quality of the fit using qqplots for several days.*

*As mentioned in our response to question 21 of referee 1, other distributions could be considered in our approach to model this nonzero part (e.g. the extended GPD distribution used by Naveau et al. (2016). This may improve the prediction but this may also lead to estimation difficulties as more complex distribution models would require more data for a robust fit. Here, the number of data is voluntarily limited to the number of analog dates considered for the fit. A more flexible model (with more parameters) would require considering more analog dates. This may be detrimental for the prediction skill as poorer analogs would be integrated in the set of dates used for the*

*estimation of the regression. Another possibility would be to fit a more flexible model with the same number of analog dates than in the present case. Due to the estimation problems mentioned above, this would however likely lead to a much more frequent use of the backup analog prediction model in our case. Such an analysis would be worth to do and this issue will be mentioned in the perspective section.*

*Note finally also that there is no contradiction between the above mentioned two paragraphs. We are going to improve the writing to make it clearer. The main point we wanted to highlight is the difference between both analog and transfer function approaches in their ability/inability to give predictions that we not already observed.*

6. The author's method is able of extrapolation? is there any evidence of the quality of the extrapolation?

*In the present work, we do not use our model for simulation but for the probabilistic prediction of precipitation. We do thus not do extrapolation although it would be an option in a simulation context.*

*As mentioned in Question5, the quality of the extrapolation would result from the quality of the model used for the considered prediction day. In our work, the quality of the regression models (for occurrence first and for non-zero amount next) is checked via the significance of the different predictors used in the regression. Extrapolation would be only possible with the model in the case of regression models with coefficients significantly different from zero. In the opposite case we use the backup analog prediction model as explained in section 3.3.*

7. p2 line 28: I am not sure how a linear model can be "extended" to non-Gaussian data. If this is not to be a reference to what Maraun et al. (2010a) did, but the authors rather claim that their method is capable of simulating non-Gaussian data, then

there is some more extensive explanation required: What kind of non-Gaussian-ness is observed in the data and how can linear models mimic this kind of non-Gaussian data? How and where is this non-Gaussianness seen in the data and how is the model describing it?

*We agree that these lines are somehow confusing. Generalized Linear Models (GLM) are regression models specifically introduced by statisticians to model non-gaussian data (see Nelder and Wedderburn, 1972). GLMs are an extension of linear regression models. They represent a large family of different statistical models which can all be described within a same theory. They were first used by Stern and Coe (1984) for the generation of precipitation. Another important application for precipitation was presented by Chandler et Wheater, 2002. The vector generalized linear models (VGLM, Yee and Wild 1996), closely related to the class of GLMs, are the most general class of linear regression models available. The work of Maraun et al. (2010b) is just one recent application of VGLMs for the case of precipitation. We will simplify this section and remove the mention to VLGMs, which is not necessary here.*

*Daily precipitation data are indeed non-gaussian. They are positive, have a mass in zero (the probability of null precipitation is strictly positive and high) and have classically a skewed distribution for non – zero amounts. To model precipitation, two different GLMs are generally used, one for the probability occurrence of precipitation, another for the distribution of non-zero precipitation. The occurrence of precipitation is classically modelled with a GLM using a logistic link function and a binomial probability distribution. The distribution of non-zero precipitation is often modelled with a GLM using a log-link function and a gamma probability distribution. We follow this two part modelling approach in our work with these two different GLM configurations.*

8. From the abstract it did not become clear to me, what is meant with an hybrid(having two kinds of components that produce the same or similar results) approach. The title

is worded more suitably. On the other hand 'local' could be confused with 'small scale'

*We thank the referee for this comment. The hybrid approach refers to the fact that two different methods, often used alone for the prediction, (the analog approach and the regression approach) are used in our approach as a combination. As suggested, we will also use "two-stage analog/regression model" in the abstract.*

*We also agree that "small scale" precipitation is more suited in the present context than "local". We will change it in the manuscript.*

**3  Setup and Language**

9. At various places within the paper (see comments below) parts of the methodology are explained. I suggest that the introduction is reworded and a section of the introduction is established that clearly and concisely explains what is done in one paragraph. This should also include an explicit statement of the goal and the novelty of the research.

*We thank the referee for those suggestions. We will modify the introduction accordingly.*

**4    Major Comments**

**4.1    Section 2 Data**

10.  Here, there is a distinction between 'analog stage' and 'regression stage' – are these two stages what is mean when the authors refer to as a hybrid approach? This gets back to my original question: In the analog stage, are the authors looking for all May30's in the past or only those May-30's where the pattern of the geopotential field was similar on the May-29's? How was this similarity determined?

*As mentioned in Question8, the "hybrid" approach indeed refers to the two stage analog/regression approach.*

*Let consider that the prediction has to be done for May-30's, 2018.  In the analog stage, we only consider days that are analogs in term of atmospheric circulation to the atmospheric circulation state of this day. As stated in the manuscript, the analog days are identified within a restricted pool of candidate days, namely all days of the archive that are included in a calendar window of $\pm$ 30 calendar days centered on the prediction day. In the present example, all May 1st to all June 30's from all years of the archive period (1982-2001 ; 20 years) are considered as candidate days (this corresponds to 1200 candidates among which only 100 days will be selected). The prediction day (May-30's, 2018) and its 5 preceding and following days are excluded from the candidates. The similarity is measured via the Teweless Wobus score which*

*compare the shapes of the geopotential fields.*

*We will adapt the text to make this analog selection step clearer.*

11. why 13 predictors? Is this enough? For what goal?

*The 13 predictors used for the regression stage gather most predictors considered in previous studies over Europe (e.g. Hanssen-Bauer et al., 2005; Wetterhall et al., 2009; Horton et al., 2012; Raynaud et al., 2016). They include predictors characterizing the thermal state of the atmosphere, its dynamics, the water atmosphere content, its thermo-dynamical instability.*

*As mentioned in the conclusion, the predictors classically used in similar downscaling works were selected owing to their prediction skill. This skill is classically the mean prediction skill evaluated for all days of a given time period. We show that some predictors could have no or fairly no prediction skill for most of the days but could be informative for very specific location/situations. Further works could thus indeed consider the interest of using other predictors, possibly non-conventional ones, as they may reveal, for very specific situations, to be very informative.*

*This point is already mentioned/suggested in the discussion but we will clarify it further.*

4.2   section 3 The hybrid analog/regression model.

12. The hybrid analog/regression model the approach of using a distribution function with a portion of zeros is clear.what is not so clear, is how the parameters are estimated

and why this is treated independently?

*As mentioned in the manuscript, (line 26 p6), the parameters are estimated using the Iterative Re-weighted Least Squares algorithm (IRLS, Nelder and Wedderburn, 1972). The significance of the regression coefficients is assessed by the Z-test (resp. the Student t-test). Because of the mass in zero, the precipitation distribution is modelled in two parts: the probability of occurrence and the distribution of non-zero amounts. The estimation of the parameters is indeed done independently, for precipitation occurrence probability first and for non-zero precipitation amounts next. This is the way the estimation is classically done (e.g. Stern and Coe, 1984; Chandler and Wheater, 2002). This allows also the selected predictors to differ for the two variables to predict.*

13. Should the amount of precipitation not be a random variable drawn from the distribution depicted in Figure 1? It could then be either zero or some precipitation intensity other than zero.

*The amount of precipitation is indeed a random variable with a given distribution which varies from one day to the other. The objective of our approach is to model this distribution and its day dependency and to further use it as probabilistic prediction. For any given prediction day, we do thus not draw some realization from the distribution. If a single scenario would be required for the prediction day, a random realization could be indeed drawn from the distribution leading to either zero or some non-zero precipitation intensity.*

14. Why is npi estimated separately from the parameters of the distribution function? (I am assuming parameters, even though Figure 1 suggests the use of an empirical distribution) Can those parameters not be estimated jointly? Now, it seems like currently

npi is estimated via a GLM, which seems to be an improved multiple regression with the secondary variables going into xĔĘo (Eq.2).

*The probability of occurrence pi is indeed estimated with a GLM where the x'o are the explanatory variables. In eq. 2, pi is the probability of occurrence estimated with the prediction model for the target prediction day. This prediction results from 1) the downscaling relationship estimated for this day from the n-analog dates (it thus depends on the predictors retained for this day and on the corresponding parameters estimated for this day) and it results also from 2) the values of the different predictors observed for the prediction day (used in a second step for the prediction with the downscaling relationship). pi is thus necessarily estimated after the parameters of the distribution have been estimated.*

*Figure 1 actually suggests that the distribution function is empirical. It is indeed empirical when it is estimated with the AM25 backup analog model. Figure 2 clearly shows however that this distribution can be updated thanks to a parametric model.*

*This will be clarified in the future manuscript version.*

15. it is not clear what the difference between superscript o and superscript q is in Eqs 2 and 3.

*The predictors identified for precipitation occurrence (o) can differ from those identified to predict the non-zero precipitation distribution. The different superscripts refer to the fact that the two sets of predictors are specific to the occurrence (o) or to the quantity (q). This notation will be clarified in the paper.*

16. How does the Gamma distribution come into the game? Are you using this type of

distribution to model the non-zero part of the distribution? Why Gamma?

*The gamma is indeed used to model the non-zero part of the distribution. This distribution is a widely used distribution in the hydrological literature for the non-zero amounts (e.g. Stern and Coe, 1984, Chandler and Weather 2002). It has the advantage to be rather robust and it does require only 2 parameters to be estimated. Other distributions could be considered in our approach to model this nonzero part. See Question6 for further comments on this point.*

17. Also, the logic in p6 lines 13,14 is off. I think you should use a distribution that fits somewhat well to the data and then fit its parameters to the data.

*We agree that lines 13-14 are not well written. We use the gamma distribution for the strictly positive precipitation. The method of moments is used to estimate the shape parameter of the distribution. Equation (4) expressed the way the variance is computed. This paragraph will be rewritten in the new version of the paper.*

18. what determines how "near" an analog is to the predicted day? (likely this is answered in Sect. 3.2).

*As indeed mentioned in section 3.2, analog are identified based on their similarity in terms of the shapes of geopotential fields. The Teweless Wobus score is used to measure this similarity (see also no 10).*

19. why is the threshold for precipitation 0.1mm?

*This threshold is often used in the hydro-meteorological literature to define wet / dry days. It corresponds in France to the precision of the bucket capacity used in the raingauge devices used to measure precipitation.*

20. p6 lines 23 ff. are difficult to understand. Say again you are trying to predict May-30 2018 in one grid location of France. Then you are searching for the "nearest" geopotential conditions for all May-29 in the past and then estimate npi based on the precipitation occurrences in those days.

*Thank you for the suggestion. As suggested, we will clarify the process of analog selection.*

21. The "nearest conditions" could be different for a neighbouring cell? What does this say about consistency and spatial dependence structure of precipitation fields.

*Yes, the referee is right. The "nearest conditions" could be different for a neighbouring cell. They are however not as different, as illustrated in Chardon et al. (2014). In a configuration where the analog model is optimized for each location, Chardon et al. (2014) show that, for a given prediction day, the analog dates are very similar from one grid cell to the next. This close similarity covers rather large domains (up to a few 100's of kilometers) excepted in regions with significant relief (the "nearest conditions" can be actually rather different on the western side and eastern side of the "Massif Central" mountainous region for instance).*

*The similarity between analog dates makes possible the development of relevant spatial scenarios (which are especially coherent in terms of spatial structure) as a given analog date can be used as scenario for all locations of a given spatial region.*

22. Also for Jun-1 2018, again a potentially very different set of "nearest conditions" could be used? Or am I understanding this wrongly, and there are more constraints?

*As we consider each prediction day independently from the previous ones, the nearest conditions of Jun-1 2018 could be indeed very different from those of May-31 2018. This is however not the case due to the strong persistence of the large scale atmospheric dynamics which makes one day often rather similar from the previous one. Note again that we do not aim to develop time series scenarios of precipitation and that we did thus not introduce any specific constraint to cope with this temporal issue.*

23. Why are you using the BIC (and not another criterium)?

*In our application we have also tested the Akaike information criterion (AIC). Both criteria give the same results.*

24. I would suggest a more careful wording when the word "significance" is employed. Arguably, a predictor can be significant at a certain level, but not plainly not significant (p6 line 26ff) – what level of significance did you choose?

*We used the 5% significance level. This information will be added in the new version of the paper.*

25. p8, l21 you start to use a differently typeset "P" after the abbreviation "ESP" – please explain.

*Thank you for the remark. Yes the "typeset P" should appear the first time we introduce the abbreviation of a given Ensemble Prediction System, named P.*

26. Figure 2: top right panel: should there not be dots on the black line? At least for the part "within" npi?

*Thank you for the comment. You are right and we will correct it.*

4.3   section Results

27. p11 l12ff: you write that the BSS gain is "very sensitive to topography". The coasts along the Mediterranean (E portion of southern coast of France) and the Atlantic (W portion of northern coast of France) have opposite BSS gains (Fig 3b). How does that fit to your explanation?

*We agree that our formulation was confusing. A first result is that the BSS gain presents a high space variability. The gain is higher in the mountainous areas (Pyrenees, Massif Central, Alpes, Vosges) but topography is indeed not the only factor that influence the gain as important gains are also obtained for the whole Mediterranean coast. We will reformulate the text accordingly.*

28. p11 l32: what do you mean by "greatly and thus significantly"?

*We agree that the formulation was awkward as we did not evaluate in a statistical meaning the significance of the gains. We will reformulate the text accordingly.*

4.4    section Discussion

29. Generally, this section reads as a strung together explanation of what is shown on several figures.  What does it mean remains more unclear than the authors probably think...

*The section "results" shows and discusses the improved prediction skill obtained with the hybrid approach.  Within the "discussion" section, we wanted to give some illustration on the adaptive behavior of the model, both in space (with different selection frequencies of given regression structure from one place to the other) and time (with the differences from one weather type to the other for instance).  We acknowledge that the meaning of the results presented in the figures of this section is not as clear and further work would be worth for a deeper analysis of the model's behavior. What seems to be clear however, is that the most interesting predictors cannot be considered the same all the year and everywhere. We are going to comment this point in the new version of the paper.*

30. can the selection of structures (what is visualised by Figures 8 and 9) be done in a more quantitative way (contribution of each variable to the prediction)?

*Another possibility would have been indeed to present the percentage of prediction*

*days each predictor has been selected. This however prevents to identify which combinations of predictors are more often selected if any. We thought however that it was important to understand which predictors were associated for the prediction. This is the reason why we have preferred to present the selection frequency of the different structures.*

31. p19 l1: Please describe first what your point is, then what is visualised on Figure 10).

*We will reformulate the text as suggested. Our point is that one single regression structure is not necessarily the best one for all large scale situations. This is what is illustrated in Fig10 with important differences, from one weather type to the other, of the selection frequency of the regression structures considered here.*

32. p22 l4ff: you write that the gain is "non-negligible". Then you write that it is "up to 0.1". – Can you quantify how much of a gain this really is?

*Thank you for the remark. The gain is up to 10 percentage points (or in relative value up to 0.1). The best value of both the BSS and CRPSS score is 100 percentage points (or in relative value, 1). The gain is here thus non-negligible indeed.*

**5  Minor Comments**

[Figure]

33. p 16, last word: necessarily

*This will be done*

34. p2 line 35: remove "obviously" or explain how this is obvious.

*This will be done*

35. throughout the paper: frequent use of "classic" and its funnily sounding adverb. What is classic in the sense of analog hydrological methods?

*This refers either to a standard approach or frequently used approach. We will use more appropriate terms in the revised manuscript.*

36. The authors mention multiple times (e.g. p3 l7, p3 l14) a relation of the presented methodology to physical (maybe deterministic?) Is the goal of the presented approach to be "physically realistic"?

*The goal of our work is of course not to obtain a physically realistic approach. We will reformulate the text to avoid such confusing statement.*

*Statistical downscaling models have to extract a statistical relationship between some large scale variables and local scale precipitation. From a physical point of view, the main physical processes, responsible or partly responsible of precipitation, can change from one weather type to the other. The most relevant predictors are also expected to change. We just wanted to point out that the statistical relationship which has to be identified is expected to be more relevant if it is estimated for a subset of days which*

*are similar in terms of large scale atmospheric configuration. This is actually expected to allow for a better identification of the most important driving large-scale variables.*

37. How / in what sense does this lead to something "more relevant and robust" (p3 l14/15)?

*The relevance is expected to be improved as discussed above.*

*From a statistical point of view, exploring the large scale- local scale downscaling relationship on a subset of days which are similar in terms of large scale atmospheric configuration is expected to allow for the estimation of a better quality statistical model. The prediction skill of the model is thus also expected to improve. We also think that the hybrid approach increases the robustness of the approach. This is however impossible to prove from a statistical point of view. We will therefore remove the notion of 'robustness'.*

38. Table 1: H, W, PV: of what variables?

*The vertical velocity at 700hPa is the vertical wind at the 700hPa pressure level. This predictor is particularly useful to locate synoptic fronts which are characterised by strong upward winds leading to condensation and precipitation. The 700hPa level is usually chosen as vertical velocities reach a maximum in the mid-troposphere.*

*The helicity quantifies how the horizontal wind vector changes in intensity and direction with the altitude. As an example, the following figures called hodograph presents how to compute the helicity. V0, V1, V2 are the horizontal wind vectors at different pressure levels (from 1000hPa to 500hPa in our case). This helicity is simply the area marked out by these vectors. In addition to the usefulness of this parameter for convective*

*systems, a brutal change in helicity over a short distance can help detect synoptic fronts.*

*Figure R3 Scheme for the computation of wind helicity from wind vectors (see supplementary material)*

*Finally, the potential vorticity is a parameter that can highlight the areas conductive to the development of low pressure systems. In our case, looking as the PV field at 400hPa, it is rather used to locate some potential instruction of stratospheric air (used called PV anomaly) which can trigger some strong vertical velocities and precipitation.*

*The full mathematical definitions of these 3 parameters can be found in Holton and Hakim (2012).*

*We will modify the description of these 3 variables for :*

*- potential vorticity of the atmosphere at 400hPa*

*- vertical velocity (vertical component of wind) at 700hPa pressure level*

*- helicity of horizontal wind integrated from 1000 to 500 hPa pressure levels*

39. p11 l4: "two" or "four" or something else?

*We have two predictor sets with four predictors each. This will be clarified.*

40. p13 l19: what does "next too low" mean?

*We agree that the text was confusing. It should have read "thus too low" and not "next too low". This will be modified.*

*When all analog days are dry (no wet days in the set of analogs retained for the prediction day), it is obviously impossible to fit any occurrence probability model. The same applies when only a few analog days are wet. There is actually no fixed value for this "too low proportion of wet analog days" for which no model can be estimated. It roughly corresponds to 0 to 20% of the days but it depends on the value of the predictors corresponding to those days. As mentioned in the manuscript, the regression model is retained or not according to the significance of the regression parameters.*

41. Style: there are many abbreviations, and it's easy to forget what they all mean.

*We agree that we use a lot of abbreviations in the paper. We will add an appendix with a glossary*

42. sometimes you write "metropolitan French territory", sometimes "France". It seems like you never looked at Paris or the major cities specifically, hence I suggest to use "France" everywhere.

*This will be accounted for.*

43. p20 l2: I don't think the interest has been explored. Rather, the model itself has been explored?

*We agree and will reformulate the text accordingly.*

**6 references**

Cavazos T, Hewitson BC. 2005. Performance of NCEP-NCAR reanalysis variables in statistical downscaling of daily precipitation. Climate Research 28: 95–107.

Chandler, RE and Wheater, HS. 2002. Analysis of rainfall variability using generalized linear models: A case study from the west of Ireland. Wat. Res. Res. (38)10.

Chardon J, Hingray B, Favre AC, Autin P, Gailhard J, Zin I, Obled C. 2014. Spatial similarity and transferability of analog dates for precipitation downscaling over France. J. Clim. 27: 5056–5074, doi: 10.1175/JCLI-D-13-00464.1.

Hanssen-Bauer I, Achberger C, Benestad RE, Chen D, Forland EJ. 2005. Statistical downscaling of climate scenarios over Scandinavia. Climate Research 29: 255–268.

Holton, J. R., Hakim, G. J. (2012). An introduction to dynamic meteorology (Vol. 88). Academic press.

Lafaysse, Matthieu. 2011. Changement climatique et régime hydrologique d'un bassin alpin. Génération de scénarios sur la Haute-Durance, méthodologie d'évaluation et incertitudes associées. Thèse LTHE, Université Paul Sabatier, Toulouse, 250p. + annexes.

Leutbecher, M.; Palmer, T. N. 2008. Ensemble forecasting. JOURNAL OF COMPUTATIONAL PHYSICS. 227(7). 3515-3539.

Maraun, D., et al. 2010a: Precipitation downscaling under climate change: Recent developments to bridge the gap between dynamical models and the end user, Rev. Geophys., 48,RG3003, 20 doi:201010.1029/2009RG000314, 2010.

Maraun, D., Rust, H. W., and Osborn, T. J.: 2010b. Synoptic airflow and UK daily precipitation extremes Development and validation of a vector generalised linear model, Extremes, 13, 133–153, doi:10.1007/s10687-010-0102-x.

Mezghani, A. and Hingray, B., 2009. A combined downscaling-disaggregation weather generator for stochastic generation of multisite hourly weather variables in complex terrain. Development and multi-scale validation for the Upper Rhone River Basin. J. Hydrology. 377 (3-4) : 245-260.

Naveau, Philippe; Huser, Raphael; Ribereau, Pierre; et al. 2016. Modeling jointly low, moderate, and heavy rainfall intensities without a threshold selection. WATER RESOURCES RESEARCH. 52(4) 2753-2769

Radanovics, S., Vidal, J.-P., Sauquet, E., Ben Daoud, A., and Bontron, G.: Optimising predictor domains for spatially coherent precipitation 5 downscaling, Hydrol. Earth Syst. Sci., 17, 4189–4208, doi:10.5194/hess-17-4189-2013, 2013.

Raynaud, D., Hingray, B., Zin, I., Anquetin, S., Debionne, S., Vautard, R. 2016. Atmospheric analogs for physically consistent scenarii of surface weather in Europe and Maghreb. Int. J. Climatology. doi :10.1002/joc.4844.

Timbal B, Fernandez E, Li Z. 2009. Generalization of a statistical downscaling model to provide local climate change projections for Australia. Environ. Model. Software 24(3): 341–358.

Wetterhall F, Halldin S, Xu CY. 2005. Statistical precipitation downscaling in central Sweden with the analogue method. Journal of Hydrology 306: 174–190. doi:10.1016/j.jhydrol.2004.09.008.

Yee, TW; Wild, CJ. 1996. Vector generalized additive models. JOURNAL OF THE ROYAL STATISTICAL SOCIETY SERIES B-METHODOLOGICAL. 58(3) 481-493

**Supplement:**

[Figure]

*Figure R2a: Percentage of time (over the 20 years used for the analysis) where the regression structure selected for a given location is the same than that used for a reference location (the reference location is indicated with the blue dot in each figure). Results are shown for 16 different reference locations.*

[Figure]

*Figure R2b: Spatial distribution of the mean regression coefficients obtained for the predictor W and the regression structure n°4. Results are presented for the different seasons and for the 8 weather types considered in this work.*

[Figure]

*Fig10.1 extracted p122 in Lafaysse(2011). Time series of winter (DJF) and summer (JJA) precipitation obtained with an Analog Prediction model (100 scenarios) for the Upper Durance basin in South-Eastern France (red = observation, green : median scenario, blue : first analog scenario)*

[Figure]

Figure R3. Scheme of the way wind helicity is estimated from wind vectors. source: NOAA

---

## Author Response (AR1)

**Revised Manuscript version of: An adaptive two-stage analog/regression model for probabilistic prediction of local precipitation in France**

**By Jérémy Chardon, B.Hingray and A.C. Favre**

Dear Editor,

We are pleased to send you the revised version of our manuscript. We made significant modifications to our initial work to account for comments and suggestions from both reviewers. They allowed strengthening the analysis we presented in the former manuscript version. They also allowed improving the pedagogical content of the paper.

As mentioned to both reviewers, the analog/regression approach we present here could be used for either forecasting, reconstruction or simulation in a future climate. We did not select any given application context for the present paper. Depending on the definitive intended use of SCAMP, some specific issues would obviously apply, calling for specific focused analyses and developments. These context specific issues are not considered here. Our main objectives are indeed to present the principles of the two-stage analog/regression approach for the context of small scale precipitation prediction, to assess its predictive power for both precipitation occurrence probability and amount, and to give some insight on its adaptive behavior and thus on the temporal variability of the downscaling link.

The main modifications of the manuscript are the following:

- We made the introduction more focused. We added a number of relevant references to recent works on statistical downscaling models. We have clarified the objective of our work (and especially the intended use of SCAMP), and the novelty of the approach.
- We fully rewrote some sections / paragraphs. For instance, the issue of the predictor selection is now considered in the "data" section only. A concise outline of SCAMP is given in the first part of section 3. The analog stage is described before the regression stage and the description of the backup model follows. The process for the selection of atmospheric analogs, the way the GLM parameters are estimated is clarified. The "discussion" is more focused on the adaptability / variability of the large-to-small scale downscaling relationship.
- We provide in a "Supplementary Material" a new figure (map over France of the probability of dry days) which allows explaining part of the results described in the manuscript.
- In the conclusion we mention additional issues and perspectives to be considered in future works (possible dependence of the results to the dataset used for the analysis and especially to the quality of the predictors, robustness of the adaptive downscaling link, possible extensions of the model for the prediction of precipitation fields).
- We finally made extensive editorial changes and reformulated some sentences that were sometimes clumsy and/or not clear.
- Note that we did not make any changes to figure 2 and figure 10 as suggested by the reviewers as the readability was not improved.

In addition to the new version of the manuscript, the revised manuscript using track changes has been attached. Changes are only highlighted for the major changes listed above, not for editorial changes. With all these elements, we really hope that you will be convinced by the value of our work and that it will be considered suitable for publication in a forthcoming issue of your journal.

Best regards,
B.Hingray

**Final response to Anonymous Referee #1**

*We thank the referee for this thorough review and for the numerous constructive suggestions that we will consider for incorporation in the modified manuscript.*

*We made significant modifications to our initial work to account for comments and suggestions from both reviewers. They allowed strengthening the analysis we presented in the former manuscript version. They also allowed improving the pedagogical content of the paper. A summary of those changes are given in the cover letter to the editor.*

*The detailed response to the comments of Referee #1 and the changes made accordingly to the manuscript are given below.*

*In addition to the new version of the manuscript, the revised manuscript using track changes has been attached. Changes are only highlighted for the major changes listed above, not for editorial changes.*

**1 Topic and general comments**

1.1 Topic The paper presents a new two-stage hybrid perfect prognosis SDM called SCAMP. SCAMP was applied to a large number of grid points in France and was proven to be adaptive to different weather types and seasons which is illustrated nicely with visually appealing figures. The method seems very interesting given the issues encountered with some other very popular downscaling- or bias correction methods (e.g. lack of variance for pure transfer functions or physical inconsistency that easily occurs with quantile mapping and related techniques). There are a couple of issues though that I think should be addressed before publication. Some of them might be just a matter of clarification, but some might be more fundamental depending on the intended use of the method. These issues are outlined in the following.

1.2 What is the intended use of the method? In the introduction you mention regional climate studies of present, past and future climate as well as numerical weather prediction (NWP) but without being very clear for which of these cases SCAMP is actually made for. Given that you downscale from 1.125 degree resolution to a 8km grid I suppose that SCAMP is not designed to do NWP, given that the ECMWF global deterministic model runs at 9km resolution and most national weather services in Europe operationally run limited area models at 1- 2km resolution and limited area ensembles at 2-10km resolution. If however that is the intended use, please explain in which context and for which users you think it could be useful. What made me doubting that SCAMP is intended for regional climate studies, is the use of the word "prediction" throughout the paper. If the intended use are regional climate studies, I would recommend to either use "simulation" rather than "prediction" or to precisely define what "prediction" means in this context. The same applies to section 3.4.

*A widely used argument for the development of statistical downscaling models (SDMs) is that they allow producing local scale weather scenarios. We obviously agree that high resolution ensembles are operationally available from most national weather services. SCAMP would not be of any interest in respect to this point.*

*As mentioned in the manuscript, another important argument for the development/use of SDMs is that the outputs of GCM and/or NWP models are generally 1) biased and, from a statistical point of view, 2) not reliable (the ensembles are often underdispersive – see for instance Leutbecher and Palmer, 2008). In a number of cases however, impact studies require unbiased and reliable meteorological scenarios. This is for instance a critical requirement for hydrological impact studies as a result of the strong linearities in the hydrological response of river basins to meteorological forcings.*

*In the present work, we did not select a given context for the application of SCAMP. SCAMP could be used for either forecasting, reconstruction or simulation. We will precise this in the new manuscript version. Some specific requirements would apply for each context. For instance, the temporal transferability of the model in a modified climate context would be required for the development of climate projections. The quality of large scale predictors would have to be checked for reconstructions over the XXth century or for climate prediction (as often reported, thermodynamic predictors are of lower quality than dynamic ones – see questions + responses #5 and #6 below).*

*We have precised what the word "prediction" means in this context.*

*We find this word more suited than "simulation" because this latter suggests that times series of precipitation are produced. This is here not the case (although some postprocessing generation process could be used for this but this is out of the scope of the present work) as we issue for each day the statistical distribution of precipitation amount (thus a probabilistic prediction).*

*Leutbecher, M, and T N. Palmer (2008). Ensemble forecasting. Journal of Computational physics, 227:3515-3539.*

1.3 Manuscript organization and conciseness

1. The introduction is to my mind rather long and could be written more concisely. In addition it should contain some more precise statement on the intended use of SCAMP (see section 1.2).

> *We have adapted the introduction as suggested and clarify the intended use of SCAMP.*

2. I don't understand why the description of the analog stage (stage 1, section 3.2 and 3.3) comes after the description of the GLM stage (stage 2, section 3.1). In my view this should be reversed. The first part of section 3 (page 5) should contain a concise outline of SCAMP. There is a start at page 5 line 10-12 that should be completed with one or two sentences on the backup model.

> *We thank the referee for these suggestions. We have completed the outline of SCAMP and we will describe the analog stage before the GLM one as suggested.*

3. The last paragraph of section 3.2 could go in a tightened section 3.3 as well (The AM as benchmark and backup model). Its last two sentences are already a very concise summary of section 3.3.

> *The Analog Model can be indeed presented as a benchmark and backup model.*

> *We have merged the last paragraph of section 3.2 with section 3.3 as suggested.*

4. I wonder about sections 2 and 4.1 as well: I found it somewhat difficult to figure out which potential predictors were actually used during the first read. There are a few things said in section 2, during section 3 things are quite vague (concerning predictors) and only in section 4.1 things became more clear. If you consider 4.1 to be a central result of the study the information in this subsection should be split into a "methods part" right after or included in section 2 and a "results part" remaining in section 4. If this is not the case I'd suggest to entirely include section 4.1 after or into section 2, but rewritten (together with section 2 from the fourth paragraph on, page 4 line 17 et seq.) in a much more concise manner. For example saying first what you used in the end and then concisely explain why. I think this would allow to be more specific and to use more precise wording in section 3. With a more clear structure lengthy transitions, such as the page 5 last sentence or page 11 lines 4-6, might not be necessary any more.

> *We thank the referee for these different suggestions. As suggested we have included section 4.1 into section 2 and modified the text and transitions consequently. This makes indeed the paper more clear.*

1.4 Language issues. Please check your paper thoroughly for language/grammar issues during the revision, especially

1. tenses • stick to simple past for things you did avoid future tense for things you finally did, otherwise it induces unnecessary doubt. 2. reduce the use of modal verbs (may, could etc.) where possible in order to be more precise and quantitative.

3. prepositions

4. word order in the context of adjectives and adverbs

5. remove superfluous adverbs for more clarity

6. add missing definite articles

7. mind French to English translation pitfalls

See the technical correction section for examples.

*We have carefully checked for these different issues. Thank you for these recommendations.*

**2 Specific comments**

1. Is SCAMP an abbreviation for something? (I'm just curious)

*In a previous work (Raynaud et al. (2016), we first worked on a multivariate Analog version, for multivariate prediction (precipitation, temperature and radiation). SCAMP is the abbreviation defined in this previous work and stands for Sequential Constructive atmospheric Analogs for Multivariate weather Prediction. We kept this abbreviation for the present work even if we are in a monovariate configuration. This has been clarified.*

*Raynaud, D., Hingray, B., Zin, I., Anquetin, S., Debionne, S., Vautard, R. 2016. Atmospheric analogs for physically consistent scenarii of surface weather in Europe and Maghreb. Int. J. Climatology. doi :10.1002/joc.4844.*

2. In the introduction (first paragraph) SDM and post-processing are used synonymously. Are they? And if yes, in which context?

*SDM and post-processing are sometimes rather synonymous for instance when SDM are used to produce local weather scenarios from GCM output data (their ability to do some bias correction is an important feature here as mentioned previously). We agree that this is not always the case. Other applications of SDM are possible as those mentioned in the second paragraph of the introduction (weather generation, climate change attribution…).*

*We have removed this paragraph. There no more possible confusion.*

3. some references seem slightly out of context. For example:

*Those reference issues have been fixed.*

4. Page 3 line 24-26: The last sentence of the paragraph is unclear. Please rewrite.

*Our point is that the type of model used in the work of Ibarra-Berastegi et al. (2011) is not really optimal. A linear regression model is indeed not suited to the non-gaussian nature of precipitation amounts. The approach of Ibarra-Berastegi et al. (2011) would thus benefit from using a model suited to precipitation. We have reformulated the sentence.*

5. Do you think that the selected predictors may depend on the data set used, or its resolution? Please comment.

*Different studies have shown that the predictors depend on the predictand. For precipitation, predictors can differ from one location to the other (e.g. Cavazos and Hewitson, 2005; Timbal et al., 2009; Chardon et al., 2014). They are also not necessary the same for precipitation, radiation or other surface weather variables (e.g. Raynaud et al. 2016). We could also expect that the predictors depend on the dataset used, for the atmospheric reanalyses especially. To our knowledge this analysis has not been carried out yet. Some dependence to the resolution is also probably to expect. A higher resolution would definitively allow for a better description of the shapes of geopotential fields. It would also allow for a more relevant simulation of thermodynamic processes. It would likely lead in turn to have higher quality variables for some quantity such as air instability (as mentioned in the following question). The quality of simulation does however not only depend on the resolution of reanalyses but also on the quality of the model and of the observed data available for assimilation. We could thus expect that data with higher resolution do not necessary always lead to better quality predictions.*

*These issues are obviously very interesting and would be worth specific analyses in the future.*

*A comment has been introduced in the perspectives of the modified manuscript.*

6. Page 4 line 20: How meaningful are quantities describing instability at 1.125 degrees resolution? and related, if the aim is to do downscaling of climate model outputs or reconstructions how well are the instability and humidity variables simulated by these models, and could the quality of this simulations be an issue for SCAMP? Please comment.

*We agree that atmospheric variables describing instability do not give a very good picture of instability when available atmospheric variables are at 1.125° resolution. To our opinion however, they can have some predictive power as a "proxy" of the instability.*

*The quality of such predictors in climate model outputs or reconstruction is obviously an issue. When applied in a reconstruction context for the whole XXth century, we indeed found that the added value of such predictors was much smaller than when applied with the recent reanalyses available for the last decades. A comment has been added in the discussion on this issue.*

7. Page 4 line 20: Be more specific on the predictors used. For example by referring to table 1 here.

*Thank you for the suggestion. We have precised the text as suggested.*

8. Figure 1: The caption text is unclear. What is highlighted in black? Is there a reason to use "quantity" and not "amount"? (Same for figures 2 and 9, page 11 line 32, page 13 lines 3, 21,25 and 26, page 15 line 2, page 22 line 6)

*We now use "amount" for the whole manuscript.*

*In figure 1, we use equation 1 to decompose $F_{AM}(y)$ into two parts. We highlight in black the contribution of the empirical cdf of the non-zero precipitation amount to the overall cdf. As expressed in equation 1, this last pdf is "weighted" by the probability of occurrence of probability. We have reformulated the caption of Figure 1 to make it clearer.*

9. Page 5: I'd suggest to add "SCAMP" to the section title of section 3.

*This has been done.*

10. Page 7 line 6: What does "+12h and +24h UTC" refer to? are this lead times? but then UTC is strange, because time differences don't have a time zone. Or does it refer to the time of the day? But if so, for which hour is the simulation?

*The text has been clarified according to your suggestion.*

11. Section 3.2: What is the archive length used for the analog model?

*The archive length is 1982-2001 is now clarified*

12. Section 3.2: Which period was used for the optimization of the predictor domains? Is it the same as for the simulation in this work? What are the implications?

*The period used for the optimization of the predictor domains is 1982-2001. The period is thus the same as the period of the predictions. The prediction skill of SCAMP presented in our manuscript may therefore be slightly overestimated.*

*To assess the influence of the optimization period, we could have followed a leave-one-out approach, where for instance, the best analogy domain would have been identified from all years except that of the current prediction day. This would have required much larger computing resource than those already used for the work presented in the manuscript. The process used for the optimization of the analogy domain is indeed rather long (as mentioned in the manuscript, it is first iterative where different spatial domains of increasing size and considered in turn. The identification of the best analogs days for a given analogy domain is also rather time consuming as a result of the similarity criterion used to compare days). It had also to be applied successively for the 8,981 grid cells of France. This optimization plus the re-estimation for each prediction day of the different regression models considered in the regression stage of SCAMP already required the use of the Grenoble University High Performance Computing centre CIMENT (https://ciment.ujf-grenoble.fr/wiki-pub/index.php/). A leave-one-out approach would have required too many computing resource and was thus not applied here.*

*We agree that the optimal domain may depend on the period used for the optimization of the method. We however expect that the domains would be rather similar when obtained from different periods and that their influence on the main results we present in our work would be limited. In a recent work carried out with an Analog Model similar to AM25, we have actually shown that slightly different domains may lead to identify – for a given prediction day, rather different sets of analog dates. We have however also shown that this does not lead to a significant difference in the prediction skill (Chardon et al., 2014). For the context of the present manuscript, an interesting work would be to explore if analogs from different but similar analogy domains would influence the choice of the predictors in the regression stage and/or also if the coefficients of the regression would change. This could contribute to assess the robustness of the approach.*

*This perspective work is now mentioned in the discussion.*

13. Section 3.3 first line: Please specify briefly what the significance conditions are.

*We used the 5% significance level for each predictor. This information has been added in the new version of the paper.*

14. Section 3.1: are there discrete values drawn from the Gamma distribution for the final prediction? And if so, how?

*We aim to model the distribution of precipitation amount, its day dependency, and to further use this distribution as probabilistic prediction. For any given prediction day, we do thus not draw some realization from the distribution.*

15. Page 9: I think it is a good thing to look at the skill with respect to climatology as you do, especially for comparison with other studies or methods, but you could have used the AM25 benchmark as P_ as well, right? Would that be equivalent to your _BSS or _CRPSS? If not, what is the difference and which one should be preferred under which circumstances?

*Yes, we could have chosen the AM25 model as reference for the evaluation of the combined model. We have preferred to use the climatological reference as this allows for normalized scores which can be compared, as mentioned by the referee, with those obtained in other studies.*

*We compared the combined approach and the AM25 model with gains in skill scores estimated for both approach with respect to the climatology. Such gains, given in terms of BSS or CRPSS percentage points, are also widely used to compare different prediction models. They thus present the advantage to be rather easy to interpret.*

16. In section 4.1 you describe several steps of restrictions applied in terms of the candidate predictors for the sake of robustness and clarity of the article. I appreciate these goals, but at present the description is a bit confusing and it remains unclear which of these restrictions are a feature of SCAMP and would be kept for a general application of SCAMP and which ones aren't and what would be the potential impact on robustness and skill.

> *The issue of predictor restrictions is an interesting point which was not easy to tackle. The main goal of those restrictions was indeed to improve the clarity of the article. The manuscript does thus not present a definitive configuration of SCAMP but more a proof of concept for an adaptive model which could use a much larger set of potential predictors, when relevant.*

> *The impact of fewer restrictions on the robustness of the method is potentially an important issue and would be worth a detailed analysis. This is now suggested as a perspective of the work.*

17. Page 13 line 8: The phrase is very unclear. Please rewrite.

> *The phrase has been modified*

18. Page 13 second paragraph: What exactly causes the GLM to "fail" in the southeast for the occurrence? Are there not enough wet analogues to estimate the occurrence probability or does it fail the significance test for the parameters? please comment.

> *We agree that the text of the second paragraph of p. 13 is somehow confusing. As mentioned in the paragraph, the GLM which models the occurrence does actually almost never fail, even in the southeast. From the sum of the frequencies obtained for the two cases "case 2" and "case 4" of Figure 5, we can see that, whatever the region, the GLM which models the occurrence is indeed activated most of the time (more than 97% of the days) (remember that case 2 corresponds to (Success of GLM modeling the occurrence + Failure of GLM modeling the quantity); case 4 corresponds to (Success of GLM modeling the occurrence + Success of GLM modeling the quantity). Consequently, and whatever the region, the situation where AM25 is used as backup for the prediction of the occurrence probability is very rare (see the sum of the frequencies obtained for case 1 and case 3).*

> *Nevertheless, Figure 5 indeed highlights a very specific behavior in the southeast when compared to the remaining of France. Case 2 is activated much more often in this region (increase of 30% percentage point) than elsewhere and, in a symmetric way, Case 4 is activated much less often than elsewhere (decrease of 30% percentage points).*

> *The reason underlying this result is to be related to the much higher proportion of dry days in southeast as illustrated in figure R1a below. For a number of predictions days, the number of analog days that are wet is indeed to be small in the southeast. This is obviously not a difficulty for the estimation of the GLM modeling the occurrence. This is conversely likely one for the estimation of the GLM modeling the quantity. For days for which the number of wet analogs is small, the size of the dataset available to fit the GLM modeling the quantity can be too small to allow for a fit with significant parameters. This very likely explains the spatial disparities in both graphs "Case 2" and "Case 4" of Figure 5.*

> *In our work, a GLM (GLM modeling the occurrence or GLM modeling the quantity) was said to fail for a given prediction day when the significance test failed for the parameters. The link of these failures with the number of wet/dry analog days is not as direct as we could have presented it in the text. It is however strongly suggested from Figure R1a and from what is described in the different graphs of Figure 6.*

> **The whole section has been rewritten for clarification. The figure of Pdry is now given in a supplementary material**

[Figure]

*Figure R1a : Probability of a dry days over the 1982-2001 period (percentage of days which are dry).*

19. Page 15 line 2-3: The predictor set optimized for the whole of France? I thought they were optimized for each grid cell and time step. Is this only for this experiment or in general? This is confusing and will hopefully get more clear with a restructured version of sections 2 and 4.1.

> *We agree that the text is somehow confusing. For the sake of clarity, a single set of potential predictors was used for all grid points of France (see section 4.1). The most interesting set may be however rather different from one region to the other. This is a possible reason for which we obtain no gain in southern France when we activate the quantity model.*

> *This has been clarified with the restructured version of the manuscript.*

20. Page 16 line 20: Please quantify which proportion of days you would consider as "reasonable".

> *We had actually not in mind to suggest a "reasonable" proportion of days, which could be used to retain a reduced number of regression sets. Considering a reduced number of regression sets would obviously allow for reducing the computational time required for the model identification/evaluation.*

> *As mentioned in the next sentence of the manuscript, this may however limits the possibility to achieve a better prediction for some (rare) events which would activate very unusal predictors. This is what is highlighted with some of the graphs in the discussion section.*

> *This has been clarified in the revised manuscript version.*

21. Looking at figure 9, I wonder if the high frequency of the $AM_{25}$ model in the south-east might be related to the Gamma distribution being a suboptimal approximation of the precipitation amount distribution in this region. Did you test this?

> *Thank you very much for this input.*

> *To our opinion, the main reason is very likely the much larger number of dry days in the Southeastern region, as already discussed in our response to comment 18 and figure R1a. (consider that the sum of frequency for case 1 and 2 in figure 5) exactly corresponds to what is shown in figure 9 for the selection frequency obtained for the backup model (structure 16)).*

> *This now mentioned in the manuscript.*

*Of course, the choice of a same distribution model for the whole country is also an issue. The distribution of non-zero precipitation amounts is indeed rather different in the southeast region than that of the other regions of France. The main reason is the existence of much more frequent and intense heavy precipitations in southeastern France (this is the reason why a lot of works have focused in the last decades on precipitation extremes in southeastern France). This is suggested with the much higher Coefficient of Variation of daily precipitation in this region (cf. figure R1b below).*

[Figure]

*Figure R1b. Variation coefficient of daily precipitation*

*The gamma distribution is obviously flexible and widely used in the hydro-meteorology literature to model strictly positive precipitation. It may be however not optimal in this specific context and a distribution with a heavy tail would be probably more appropriate (e.g. the extended GPD distribution introduced by Naveau et al., 2016). This may improve the prediction but this may also lead to estimation difficulties as a more complex distribution would require more data for a robust fit. Here, the number of data is voluntarily limited to the number of analog dates considered for the fit. A more flexible model (with more parameters) would require considering more analog dates. This may be detrimental for the prediction skill as poorer analogs would be integrated in the set of dates used for the estimation of the regression. Another possibility would be to fit a more flexible model with the same number of analog dates than in the present case. Due to the estimation problems mentioned above, this would however likely lead to a much more frequent use of the backup analog prediction model in our case.*

22. Why is figure 10 a line chart? There is no order in the WPs, is there? I'd recommend to transform this in a series of bar charts (one for each WP). This would further avoid all the colors and line types and thus solve the issue with the invisible (probably yellow) dotted line for R700+H+Occ-1 in a) and R700+T700+W700 in b).

*We agree that a series of bar charts would have been more relevant. We tested this representation but this made the figure less clear. We thus kept the initial representation.*

23. Depending on the intended use of SCAMP, the temporal structure of the simulated precipitation might be relevant. I suppose that a detailed analysis of the representation of the annual cycle, the autocorrelation and the interannual variability in both SCAMP and AM25 is beyond the scope of the present paper, especially since this is not straight forward for probabilistic simulations, and you might

have a look at this in future work, but could you make a statement on the overall variance of the SCAMP simulations as compared to the benchmark and the observations? Typically analog models reproduce the observed variance quite well while deterministic regression models suffer from reduced variance. Since SCAMP is a hybrid model it would be interesting to know which characteristics it "inherits".

> *In a context where time series have to be simulated, additional criteria would be actually relevant to evaluate / compare the different modelling approaches; they should especially include as mentioned by the referee, the ability to reproduce the observed variance of precipitation from daily to interannual time scales.*

> *Those additional evaluation criteria would obviously depend on the final application of the model. This has been mentioned in the conclusion.*

> *Deterministic regression models are indeed known to underestimate the variance of daily precipitation. However, scenarios obtained with deterministic regression models disregard the variance of the residuals in the regression. However, regression models can be also used in a stochastic simulation framework, where a random variable is drawn from the statistical distribution associated to the regression. In Mezghani and Hingray (2009) for instance, such a generation process was used, to first identify if the prediction day was wet or dry (based on a random variable compared to the occurrence probability obtained from a first occurrence model) and to next generate some precipitation amount (based on a random realization within the gamma distribution used to model the distribution of precipitation amount in case of a wet day). The observed variance of daily precipitation was well reproduced.*

> *If time series would have to be simulated with SCAMP, a similar stochastic generation process would be followed for the regression stage. We thus would expect that the variance of observed daily precipitation would be rather well reproduced as in Mezghani and Hingray (2009). The regression stage is also not expected to increase the variance that would be obtained for the benchmark analog (in a configuration where one of the k-nearest analogs is randomly sorted each day and used as weather scenario for the day). This is one of the preliminary results we obtained for a similar work we currently develop in western Switzerland.*

24. page 22 lines 16-20: This part is not clear, please rewrite.

> *We have reformulated this paragraph.*

25. page 22 lines 28-32: I don't understand what "classically" means in this part. please use some more precise wording.

> *The word "often" is indeed more suited there.*

26. page 22 line 35: This sentence is not clear to me. In what sense is the set of days homogeneous?

> *The days are homogeneous with respect to their large-scale atmospheric circulation configuration. We have clarified this.*

27. page 23 line 2: The sentence is not clear to me. Which context? and who leaves room for improvement?

> *The context we wanted to refer is that of very specific meteorological configurations that may be observed from time to time and for which the usual predictors are sub-optimal. The analog/regression approach presented here is expected to allow for the identification of better suited predictors and thus for an improved prediction skill for the prediction day under consideration.*

> *We have modified the paragraph.*

28. Is SCAMP transferable to other regions or countries? To what extent? Under which circumstances would it be necessary/unnecessary to redo the predictor selection? Please comment.

*SCAMP is indeed transferable to other regions. We will add a comment on this in the discussion. The development of SCAMP for another context would obviously first require the identification of the best predictor set. In a number of previous studies, the most relevant predictors to be used for statistical downscaling are indeed found to be region dependent (e.g. Cavazos and Hewitson, 2005; Timbal et al., 2009; Chardon et al., 2014). Similar conclusions have likely to be expected for SCAMP also. Part of these conclusions may also result from the occurrence frequency of weather regimes observed for the different regions. They can actually vary a lot from one region to the other. In such a case, the best predictors identified for a given region could partly be a result of the fact that the weather situations for which they are more relevant are more frequently observed for that region. This issue would be worth future investigations.*

29. It would be helpful to mark or highlight the predictors that were preselected for the occurrence and amount models respectively in table 1.

*Thank you for the suggestion. This has been done*

**3 Technical corrections**

*We thank the reviewer for the very careful reading of the paper and for all the technical corrections pointed out here.*

*The text has been corrected accordingly.*

**Final response to Anonymous Referee #2**

*We thank the referee for this thorough review and for the numerous constructive suggestions that we will consider for incorporation in the modified manuscript.*

*We made significant modifications to our initial work to account for comments and suggestions from both reviewers. They allowed strengthening the analysis we presented in the former manuscript version. They also allowed improving the pedagogical content of the paper. A summary of those changes are given in the cover letter to the editor.*

*The detailed response to the comments of Referee #1 and the changes made accordingly to the manuscript are given below.*

*In addition to the new version of the manuscript, the revised manuscript using track changes has been attached. Changes are only highlighted for the major changes listed above, not for editorial changes.*

**Big Picture**

1. The authors present and explore a methodology to simulate precipitation intensities. Yet, neither time series and/or spatial fields of simulated precipitation intensities are shown nor compared to observations (in a probabilistic manner as the title might suggest). While the methodology might be beautiful, I think this is the biggest missing thing in this paper. I am not a specialist in analog methods. I did my best to understand what is done here. Ideally my potential failings help to detect shortcomings in the paper and lead to improvements. Besides the analog part, I tried to help with general statistical hydrological comments.

*The downscaling model first aims to issue probabilistic predictions of local precipitation at any given grid point of the SAFRAN grid. This prediction results in a probabilistic distribution function for each prediction day for each grid point. A times series representation, where (probabilistic) simulations and observations are compared, is thus not really convenient. The evaluation is here done with the CRPSS, which is frequently used for the evaluation of probabilistic predictions in a framework where we have to compare one value (the observation) with a whole distribution.*

*We also agree that the prediction of precipitation fields is an important issue. It was not in the scope of our work but further work should consider this issue.*

**A paragraph has been added in the conclusion on this point.**

**"Hybrid" Approach**

2. The authors want to predict a variable (e.g., precipitation) for a given day (say, for the example of this review, May 30th 2018) at a given location (within France). Then they look at all 30-Mays in the past when precipitation amounts were recorded. Where exactly do the authors look? at the closest measurement station? Is an interpolation performed? What kind of spatial dependence between observations (and simulated values) is assumed / considered?

*In the present work, we want to predict local precipitation only. We do not thus need any assumption on the spatial dependence between stations.*

*For each prediction target location (each target grid box), we only use precipitation data that were estimated within the SAFRAN reanalysis system at this exact location. Note that the SAFRAN reanalysis give an estimate of daily precipitation at each location from the closest measurement stations and from some information on the weather type for each day. To date, these SAFRAN estimates provide the only high resolution reanalysis of local precipitation over*

*France for the 60 past years. We consider these estimates as pseudo-observations. This is now clarified.*

3. On p22 l1ff you write that "the predictors and regression coefficients of the regression models vary from one day to the other? – How much do they vary? And how much do they vary in neighbouring cells? Is there some kind of relationship between the variations in neighbouring cells? Can you show this?

*Thank you for this very interesting point. We actually did not estimate these day-to-day variations. We will change this formulation for "the predictors and regression coefficients of the regression models can thus vary from one day to the other". We have actually an indirect estimation of the variation in the predictors with the different sampling frequencies obtained for the different weather types. If not straightforward, a more formal / direct evaluation would be probably worth.*

*We agree that the regional consistency of the predictors is another very interesting issue. An evaluation of this consistency is indeed expected inform on the robustness of the downscaling relationship. From the results shown in the manuscript, we have a partial idea of this with the rather high spatial coherency of the selection frequency obtained for each regression structure considered in our work. This is illustrated in Figures 8, 9 and 11. In Figure 11, this coherency is also shown to be high (even if with different spatial patterns) for different weather types.*

*This spatial coherency is also suggested from other analyses not shown in the manuscript. Figure R2a below presents the percentage of time (over the 20 years used for the analysis) where the regression structure selected for a given location is the same than the regression structure used for a reference location. As illustrated, this percentage of time with same structures - varies from 35% to more than 50% in the close neighborhood of the reference location. Some regional consistency is also found for the regression coefficient obtained for the predictors. We find that the spatial pattern of this consistency depends on the weather situation. This is illustrated in Figure R2b below for the regression coefficient estimated for W in the regression structure $n^o$ 4.*

[Figure]

*Figure R2a: Percentage of time (over the 20 years used for the analysis) where the regression structure selected for a given location is the same than that used for a reference location (the reference location is indicated with the blue dot in each figure). Results are shown for 16 different reference locations.*

[Figure]

*Figure R2b: Spatial distribution of the mean regression coefficients obtained for the predictor W and the regression structure n°4. Results are presented for the different seasons and for the 8 weather types considered in this work.*

4. What if the observed time-series is not stationary? Are there any checks performed? Is stationarity assumed? How strong of an assumption is it?

*No hypothesis of stationarity is assumed in the present work. The time series of precipitation can be non-stationary. In such a case, this non-stationarity would be expected to be reproduced as a result for instance of a change in frequency of weather types or as a result of a non-stationarity (i.e. temporal evolution) in some of the predictor variables.*

*An indirect illustration of this is the ability of the method to reproduce the interannual variability of annual precipitation (or of seasonal precipitation). This was illustrated by Lafaysse (2011) for three different downscaling approaches similar to the present one.*

[Figure]

*Fig10.1 extracted p122 in Lafaysse(2011). Time series of winter (DJF) and summer (JJA) precipitation obtained with an Analog Prediction model (100 scenarios) for the Upper Durance basin in South-Eastern France (red = observation, green : median scenario, blue : first analog scenario)*

5.  The authors claim that values outside the range of observations can be simulated via "extrapolation" (p2 line 20ff.) – some background / assumptions / limitations of this extrapolation methodology is required. The previous statement seems contradictory to what is said on p2 lines 29ff.:

    *In this paragraph, we wanted to highlight the issue of simulating non observed values. This obviously refers to values that are outside the range of observed values (including rare values): this is indeed extrapolation. This also refers to all values that are in-between two consecutive observed ones. In our context, precipitation is estimated as a function of different predictors. For a given prediction day, the statistical model give a prediction which is a kind of interpolation from what has been observed in configurations that are close to the configuration of the prediction day (in terms of predictors). We have mentioned these two different prediction contexts in our initial text but we understand that too much emphasize was given to extrapolation.*

    *As the development of our model is not justified by this extrapolation issue, we have removed it from the manuscript.*

6.  The author's method is able of extrapolation? is there any evidence of the quality of the extrapolation?

    *The "extrapolation" issue is not more considered in the revised manuscript version (see Q5 above).*

7.  p2 line 28: I am not sure how a linear model can be "extended" to non-Gaussian data. If this is not to be a reference to what Maraun et al. (2010) did, but the authors rather claim that their method is capable of simulating non-Gaussian data, then there is some more extensive explanation required: What kind of non-Gaussian-ness is observed in the data and how can linear models mimic this kind of non-Gaussian data? How and where is this non-Gaussianness seen in the data and how is the model describing it?

    *We agree that these lines are somehow confusing. Generalized Linear Models (GLM) are regression models specifically introduced by statisticians to model non-gaussian data (see Nelder and Wedderburn, 1972). GLMs are an extension of linear regression models. They represent a large family of different statistical models which can all be described within a same theory. They*

*were first used by Stern and Coe (1984) for the generation of precipitation. Another important application for precipitation was presented by Chandler et Wheater, 2002. The vector generalized linear models (VGLM, Yee and Wild 1996), closely related to the class of GLMs, are the most general class of linear regression models available. The work of Maraun et al. (2010) is just one recent application of VGLMs for the case of precipitation.*

*We have simplified this section and remove the mention to VLGMs, which is not necessary here.*

*Daily precipitation data are indeed non-gaussian. They are positive, have a mass in zero (the probability of null precipitation is strictly positive and high) and have classically a skewed distribution for non – zero amounts. To model precipitation, two different GLMs are generally used, one for the probability occurrence of precipitation, another for the distribution of non-zero precipitation. The occurrence of precipitation is classically modelled with a GLM using a logistic link function and a binomial probability distribution. The distribution of non-zero precipitation is often modelled with a GLM using a log-link function and a gamma probability distribution. We follow this two part modelling approach in our work with these two different GLM configurations.*

*Nelder, J.A., Wedderburn, R.W.M., 1972. Generalized linear models. Journal of the Royal Statistical Society A 135 (370–384).*

*Yee, T.W.,Wild, C.J, 1996.: Vector generalized additive models. J. R. Stat. Soc., B 58, 481–493 )*

*Stern, R.D., Coe, R., 1984. A model-fitting analysis of daily rainfall data. Journal of the Royal Statistical Society Series A – Statistics in Society 147, 1–34.*

*Chandler, RE; Wheater, HS. 2002.. Analysis of rainfall variability using generalized linear models: A case study from the west of Ireland.. Water Resources Research 38(10).*

8. From the abstract it did not become clear to me, what is meant with an _hybrid_("having two kinds of components that produce the same or similar results") approach. The title is worded more suitably. On the other hand "local" could be confused with "small scale"

   *We thank the referee for this comment. The hybrid approach refers to the fact that two different methods, often used alone for the prediction, (the analog approach and the regression approach) are used in our approach as a combination.*

   *As suggested, we now use "two-stage analog/regression model" in the abstract and elsewhere.*

   *We also agree that "small scale" precipitation is more suited in the present context than "local". We have changed it in the manuscript.*

**Setup and Language**

9. At various places within the paper (see comments below) parts of the methodology are explained. I suggest that the introduction is reworded and a section of the introduction is established that clearly and concisely explains what is done in one paragraph. This should also include an explicit statement of the goal and the novelty of the research.

   *We thank the referee for those suggestions. We have modified the introduction accordingly.*

**Major Comments**

**Section 2 Data**

10. Here, there is a distinction between "analog stage" and "regression stage" – are these two stages what is mean when the authors refer to as a _hybrid_ approach? This gets back to my original question: In the analog stage, are the authors looking for all May30's in the past or only those

May-30's where the pattern of the geopotential field was similar on the May-29's? How was this similarity determined?

*As mentioned in Q#8, the "hybrid" approach indeed refers to the two stage analog/regression approach.*

*Let consider that the prediction has to be done for May-30's, 2018. In the analog stage, we only consider days that are analogs in term of atmospheric circulation to the atmospheric circulation state of this day. As stated in the manuscript, the analog days are identified within a restricted pool of candidate days, namely all days of the archive that are included in a calendar window of ± 30 calendar days centered on the prediction day. In the present example, all May 1st to all June 30's from all years of the archive period (1982-2001 ; 20 years) are considered as candidate days (this corresponds to 1200 candidates among which only 100 days will be selected). The prediction day (May-30's, 2018) and its 5 preceding and following days are excluded from the candidates. The similarity is measured via the Teweless Wobus score which compare the shapes of the geopotential fields.*

*We have adapted the text to make this analog selection step clearer.*

11. why 13 predictors? Is this enough? For what goal?

*The 13 predictors used for the regression stage gather most predictors considered in previous studies over Europe (e.g. Hanssen-Bauer et al., 2005; Wetterhall et al., 2009; Horton et al., 2012; Raynaud et al., 2016). They include predictors characterizing the thermal state of the atmosphere, its dynamics, the water atmosphere content, its thermo-dynamical instability.*

*As mentioned in the conclusion, the predictors classically used in similar downscaling works were selected owing to their prediction skill. This skill is classically the mean prediction skill evaluated for all days of a given time period. We show that some predictors could have no or fairly no prediction skill for most of the days but could be informative for very specific location/situations. Further works could thus indeed consider the interest of using other predictors, possibly non-conventional ones, as they may reveal, for very specific situations, to be very informative.*

*This point is already mentioned/suggested in the discussion but we have clarified it further.*

**3 The hybrid analog/regression model the approach of using a distribution function with a portion of zeros is clear.**

12. what is not so clear, is how the parameters are estimated and why this is treated independently?

*As mentioned in the manuscript, (line 26 p6), the parameters are estimated using the Iterative Re-weighted Least Squares algorithm (IRLS, Nelder and Wedderburn, 1972). The significance of the regression coefficients is assessed by the Z-test (resp. the Student t-test). Because of the mass in zero, the precipitation distribution is modelled in two parts: the probability of occurrence and the distribution of non-zero amounts. The estimation of the parameters is indeed done independently, for precipitation occurrence probability first and for non-zero precipitation amounts next. This is the way the estimation is classically done (e.g. Stern and Coe, 1984; Chandler and Wheater, 2002). This allows also the selected predictors to differ for the two variables to predict.*

13. Should the amount of precipitation not be a random variable drawn from the distribution depicted in Figure 1? It could then be either zero or some precipitation intensity other than zero.

*The amount of precipitation is indeed a random variable with a given distribution which varies from one day to the other. The objective of our approach is to model this distribution and its day dependency and to further use it as probabilistic prediction. For any given prediction day, we do thus not draw some realization from the distribution. If a single scenario would be required for the prediction day, a random realization could be indeed drawn from the distribution leading to either zero or some non-zero precipitation intensity.*

14. Why is npi estimated separately from the parameters of the distribution function? (I am assuming parameters, even though Figure 1 suggests the use of an empirical distribution) Can those parameters not be estimated jointly? Now, it seems like currently npi is estimated via a GLM, which seems to be an improved multiple regression with the secondary variables going into xˆo (Eq.2).

*The probability of occurrence pi is indeed estimated with a GLM where the x'o are the explanatory variables. In eq. 2, pi is the probability of occurrence estimated with the prediction model for the target prediction day. This prediction results from 1) the downscaling relationship estimated for this day from the n-analog dates (it thus depends on the predictors retained for this day and on the corresponding parameters estimated for this day) and it results also from 2) the values of the different predictors observed for the prediction day (used in a second step for the prediction with the downscaling relationship). pi is thus necessarily estimated after the parameters of the distribution have been estimated.*

*Figure 1 actually suggests that the distribution function is empirical. It is indeed empirical when it is estimated with the AM25 backup analog model. Figure 2 clearly shows however that this distribution can be updated thanks to a parametric model.*

*This has been clarified in the revised manuscript.*

15. it is not clear what the difference between superscript o and superscript q is in Eqs 2 and 3.

*The predictors identified for precipitation occurrence (o) can differ from those identified to predict the non-zero precipitation distribution. The different superscripts refer to the fact that the two sets of predictors are specific to the occurrence (o) or to the quantity (q).*

*This notation has been clarified in the paper.*

16. How does the Gamma distribution come into the game? Are you using this type of distribution to model the non-zero part of the distribution? Why Gamma?

*The gamma is indeed used to model the non-zero part of the distribution. This distribution is a widely used distribution in the hydrological literature for the non-zero amounts (e.g. Stern and Coe, 1984, Chandler and Weather 2002). It has the advantage to be rather robust and it does require only 2 parameters to be estimated. Other distributions could be considered in our approach to model this nonzero part (e.g. the extended GPD distribution used by Naveau et al. (2016). This may improve the prediction but this may also lead to estimation difficulties as more complex distribution models would require more data for a robust fit. Here, the number of data is voluntarily limited to the number of analog dates considered for the fit. A more flexible model (with more parameters) would require considering more analog dates. This may be detrimental for the prediction skill as poorer analogs would be integrated in the set of dates used for the estimation of the regression. Another possibility would be to fit a more flexible model with the same number of analog dates than in the present case. Due to the estimation problems mentioned above, this would however likely lead to a much more frequent use of the backup analog prediction model in our case.*

17. Also, the logic in p6 lines 13,14 is off. I think you should use a distribution that fits somewhat well to the data and then fit its parameters to the data.

*We agree that lines 13-14 are not well written. We use the gamma distribution for the strictly positive precipitation. The method of moments is used to estimate the shape parameter of the distribution. Equation (4) expressed the way the variance is computed.*

*This paragraph has been clarified*

18. what determines how "near" an analog is to the predicted day? (likely this is answered in Sect. 3.2).

*As indeed mentioned in section 3.2, analog are identified based on their similarity in terms of the shapes of geopotential fields. The Teweless Wobus score is used to measure this similarity (see also no 10).*

19. why is the threshold for precipitation 0.1mm?

*This threshold is often used in the hydro-meteorological literature to define wet / dry days. It corresponds in France to the precision of the bucket capacity used in the raingauge devices used to measure precipitation.*

20. p6 lines 23 ff. are difficult to understand. Say again you are trying to predict May-30 2018 in one grid location of France. Then you are searching for the "nearest" geopotential conditions for all May-29 in the past and then estimate npi based on the precipitation occurrences in those days.

*Thank you for the suggestion. We have clarified the process of analog selection.*

21. The "nearest conditions" could be different for a neighbouring cell? What does this say about consistency and spatial dependence structure of precipitation fields.

*Yes, the referee is right. The "nearest conditions" could be different for a neighbouring cell. They are however not as different, as illustrated in Chardon et al. (2014). In a configuration where the analog model is optimized for each location, Chardon et al. (2014) show that, for a given prediction day, the analog dates are very similar from one grid cell to the next. This close similarity covers rather large domains (up to a few 100's of kilometers) excepted in regions with significant relief (the "nearest conditions" can be actually rather different on the western side and eastern side of the "Massif Central" mountainous region for instance).*

*The similarity between analog dates makes possible the development of relevant spatial scenarios (which are especially coherent in terms of spatial structure) as a given analog date can be used as scenario for all locations of a given spatial region.*

*This is now mentioned in the conclusion.*

22. Also for Jun-1 2018, again a potentially very different set of "nearest conditions" could be used? Or am I understanding this wrongly, and there are more constraints?

*As we consider each prediction day independently from the previous ones, the nearest conditions of Jun-1 2018 could be indeed very different from those of May-31 2018. This is however not the case due to the strong persistence of the large scale atmospheric dynamics which makes one day often rather similar from the previous one. Note again that we do not aim to develop time series scenarios of precipitation and that we did thus not introduce any specific constraint to cope with this temporal issue.*

23. Why are you using the BIC (and not another criterium)?

*In our application we have also tested the Akaike information criterion (AIC). Both criteria give the same results.*

24. I would suggest a more careful wording when the word "significance" is employed. Arguably, a predictor can be significant at a certain level, but not plainly not significant (p6 line 26ff) – what level of significance did you choose?

*We used the 5% significance level. This information has been added.*

25. p8, l21 you start to use a differently typeset "P" after the abbreviation "ESP" – please explain.

*Thank you for the remark. Yes the "typeset P" should appear the first time we introduce the abbreviation of a given Ensemble Prediction System, named P.*

*This has been corrected*

**4 Results**

26. p11 l12ff: you write that the BSS gain is "very sensitive to topography". The coasts along the Mediterranean (E portion of southern coast of France) and the Atlantic (W portion of northern coast of France) have opposite BSS gains (Fig 3b). How does that fit to your explanation?

*We agree that our formulation was confusing. A first result is that the BSS gain presents a high space variability. The gain is higher in the mountainous areas (Pyrenees, Massif Central, Alpes, Vosges) but topography is indeed not the only factor that influence the gain as important gains are also obtained for the whole Mediterranean coast.*

*We have reformulated the text.*

27. p11 l32: what do you mean by "greatly and thus significantly"?

*We agree that the formulation was awkard as we did not evaluate in a statistical meaning the significance of the gains.*

*We have reformulated the text.*

**5 Discussion**

28. Generally, this section reads as a strung together explanation of what is shown on several figures. What does it mean remains more unclear than the authors probably think...

*The section "results" shows and discusses the improved prediction skill obtained with the hybrid approach. Within the "discussion" section, we wanted to give some illustration on the adaptive behavior of the model, both in space (with different selection frequencies of given regression structure from one place to the other) and time (with the differences from one weather type to the other for instance). We acknowledge that the meaning of the results presented in the figures of this section is not as clear and further work would be worth for a deeper analysis of the model's behavior. What seems to be clear however, is that the most interesting predictors cannot be considered the same all the year and everywhere.*

*We have modified the text to better highlight this point.*

29. can the selection of structures (what is visualised by Figures 8 and 9) be done in a more quantitative way (contribution of each variable to the prediction)?

*Another possibility would have been indeed to present the percentage of prediction days each predictor has been selected. This however prevents to identify which combinations of predictors are more often selected if any. We thought however that it was important to understand which predictors were associated for the prediction. This is the reason why we have preferred to present the selection frequency of the different structures.*

30. p19 l1: Please describe first what your point is, then what is visualised on Figure 10).

*We have reformulated the text as suggested. Our point is that one single regression structure is not necessarily the best one for all large scale situations. This is what is illustrated in Fig10 with important differences, from one weather type to the other, of the selection frequency of the regression structures considered here.*

31. p22 l4ff: you write that the gain is "non-negligible". Then you write that it is "up to 0.1". – Can you quantify how much of a gain this really is?

*Thank you for the remark. The gain is up to 10 percentage points (or in relative value up to 0.1). The best value of both the BSS and CRPSS score is 100 percentage points (or in relative value, 1). The gain is here thus non-negligible indeed.*

**Minor Comments**

*We have modified the text to account for those different comments when required.*

*Cavazos T, Hewitson BC. 2005. Performance of NCEP-NCAR reanalysis variables in statistical downscaling of daily precipitation. Clim. Res. 28: 95–107.*

*Timbal B, Fernandez E, Li Z. 2009. Generalization of a statistical downscaling model to provide local climate change projections for Australia. Environ. Model. Software 24(3): 341–358.*

*Chardon J, Hingray B, Favre AC, Autin P, Gailhard J, Zin I, Obled C. 2014. Spatial similarity and transferability of analog dates for precipitation downscaling over France. J. Clim. 27: 5056–5074, doi: 10.1175/JCLI-D-13-00464.1.*

*Raynaud, D., Hingray, B., Zin, I., Anquetin, S., Debionne, S., Vautard, R. 2016. Atmospheric analogs for physically consistent scenarii of surface weather in Europe and Maghreb. Int. J. Climatology. doi :10.1002/joc.4844.*

*Mezghani, A. and Hingray, B., 2009. A combined downscaling-disaggregation weather generator for stochastic generation of multisite hourly weather variables in complex terrain. Development and multi-scale validation for the Upper Rhone River Basin. J. Hydrology. 377 (3-4) : 245-260.*

**New manuscript version with main change (tracking mode)**

\begin{document}

\title{An adaptive two-stage analog/regression model for probabilistic prediction of small-scale precipitation in France.}

% \Author[affil]{given_name}{surname}

\Author[1]{J{\'e}r{\'e}my}{Chardon}
\Author[1]{Benoit}{Hingray}
\Author[1]{Anne-Catherine}{Favre}

[revised manuscript text omitted]
 \ref{sec:data} describes the data and section \ref{sec:models} the two-stage downscaling model. Section \ref{sec:results} presents the skill of the model for the prediction of both precipitation occurrence and amount. The adaptive behavior of the model is considered in section \ref{sec:discussion} and section \ref{sec:conclusion} concludes.

\section{Data}
\label{sec:data}

The predictand is the daily small-scale precipitation estimated for the 1982-2001 period over 8,981 grid cells of 8 $\times$ 8 \unit{km\textsuperscript{2}} covering the continental French territory. The predictand is "local" precipitation, i.e. precipitation at a given grid cell. Each of the 8,981 grid cells is thus considered in turn in the following independently of the other cells. In other words, the predictions do not target precipitation fields. Small-scale precipitation data are obtained from the SAFRAN analysis produced for several surface variables at hourly time step by MeteoFrance \citep{quintana-segui_analysis_2008,vidal_50-year_2010}. SAFRAN precipitation estimates are obtained each day from the closest measurement stations. They are considered as pseudo-observations in the following.

Atmospheric predictors are taken from the European Centre for Medium-Range Weather Forecasts (ECMWF) Re-Analysis \citep[ERA-40,][]{uppala_era-40_2005}. This global meteorological re-analysis is available on a 1.125\unit{\textbf{$^{\circ}$}} $\times$ 1.125\unit{\textbf{$^{\circ}$}} grid with a 6-hourly temporal resolution.

For the analog stage, predictors are the 1000hPa and 500hPa geopotential height fields over a large spatial domain (roughly Lat = $10^{\unit{\textbf{$^{\circ}$}}}$, Lon = $8^{\unit{\textbf{$^{\circ}$}}}$) centered on the target location. These predictors have been found to be the most informative large scale predictors to be used in this context for France

\citep[e.g.][]{guilbaud_approche_1998,obled_quantitative_2002,radanovics_optimising_2013}. They also correspond to the best large scale predictors of daily precipitation for different regions in Europe with contrasted meteorological regimes \citep{raynaud_multivariate_2016}.

[revised manuscript text omitted]

\begin{figure}[t]
\includegraphics[width = 8.3cm]{fig01.pdf}
\caption{Cumulative distribution function (cdf) of precipitation amount for a given prediction day (in grey) at a given grid cell. For illustration, the prediction here corresponds to the empirical cdf achieved with the Analog Model (AM) mentioned in \ref{sec: def_backup}. The contribution of the precipitation amount $F_{Q,AM}$ cdf to the overall cdf is highlighted in black (c.f. Eq. \ref{eq:cdf_precip}).}
\label{fig:description_cdf}
\end{figure}

In the present work, the cdf of precipitation is modeled for each grid cell and each prediction day with GLMs \citep{coe_fitting_1982, stern_model_1984}, estimated for this specific day from atmospheric analogs of the day. The probability of precipitation occurrence and the cdf of the non-zero precipitation amount are modeled separately.

In the following, we first describe the Analog Model (AM) used to identify atmospheric analog days (section \ref{sec:def_AM}) and the GLMs applied in the regression stage (section \ref{sec:GLM_description}).

As discussed later, one can face prediction days where the regression stage fails, i.e. wherethe regression parameters are not significantly different from zero at the chosen significance level. For such days, we use the Analog Model as backup prediction model. The backup model can be used for precipitation occurrence probability, for non-zero precipitation amount or for both predictands simultaneously.

The way these different models are combined to finally give, for the current prediction day, a probabilistic prediction of precipitation is presented in section \ref{sec:def_backup}. In the following, this hybrid analog/regression model is further referred to as SCAMP (SCAMP stands for Sequential Constructive atmospheric Analogs for Multivariate weather Prediction and refers to the model presented by \cite{raynaud_multivariate_2016} for the multivariate prediction of precipitation/temperature/radiation/wind).

\subsection{Atmospheric analogs}
\label{sec:def_AM}

The atmospheric analog days retained for the regression stage are identified with an Analog Model defined from the developments of several past studies in France \citep[e.g.][]{obled_quantitative_2002,marty_toward_2012,radanovics_optimising_2013}.

For any given prediction day (e.g. May-31's, 2018), the analog days retained for the regression are the $N_d$ days that are most similar to that day in terms of large scale atmospheric circulation. The similarity is assessed using the Teweless-Wobus Score \citep[TWS, ][]{teweless_verification_1954} applied to the geopotential height at 1000 hPa and 500 hPa at 12h and 24h UTC respectively. The TWS compares the shapes of geopotential fields, and thus informs on the localization of low and high pressure systems and on the origin of air masses. Note that the $N_d$ analog days are identified within a restricted pool of candidate days, namely all days of the archive that are included in a calendar window of $\pm$ 30 calendar days centered on the prediction day (for the prediction of May-31's, 2018, candidates are all May 1st to all June 30's from all years of the archive). The prediction day (May-31's, 2018) and its 5 preceding and following days are excluded from the candidates. In the present work, the archive period corresponds to 1982-2001 (20 years) and we used the 100-nearest atmospheric analog days to estimate the GLMs in the regression stage.

Following \citet{chardon_spatial_2014}, the domain considered to estimate the atmospheric similarity was optimized for each target location. A different analog model was thus considered for each of the 8,981 SAFRAN grid cells. For each prediction day, the analog days thus likely differ from one SAFRAN grid cell to the next \citep[see][for illustration]{chardon_spatial_2014}.

\subsection{Regression stage with GLMs}
\label{sec:GLM_description}

The cdf of precipitation is then modeled for each prediction day with GLMs estimated for this specific day from the atmospheric analogs of the day. GLMs make the cdf depending on some covariates, atmospheric predictors in the present case.

For each prediction day, the probability of precipitation occurrence $\pi$ was modeled with a GLM in the form of a logistic regression as:

\begin{equation}
\log \left( \frac{\pi}{1-\pi} \right) = \mathbf{x^o}^\text{T} \mathbf{\beta^o},
\label{eq:logistic_regression}
\end{equation}

where $\mathbf{x^o}$ is the scalar vector of the $K_o$ predictors $(x^o_1, x^o_2, .. x^o_{K_o})$ and $\mathbf{\beta^o}$ the scalar vector of the $K_o$ corresponding regression coefficients $(\beta^o_1, \beta^o_2, .. \beta^o_{K_o})$. \par

For the non-zero precipitation amount, we used a GLM with the gamma distribution and the log link function. The expected amount $\mu$ of non-zero precipitation is therefore here expressed as:

\begin{equation}
\log \left( \mu \right) = \mathbf{x^q}^\text{T} \mathbf{\beta^q},
\label{eq:logarithmic_regression}
\end{equation}

where $\mathbf{x^q}$ denotes the scalar vector of the $K_q$ predictors $(x^q_1, x^q_2, .. x^q_{K_q})$ and $\mathbf{\beta^q}$ the scalar vector of the corresponding regression coefficients $(\beta^q_1, \beta^q_2, .. \beta^q_{K_q})$ . The shape parameter $\nu$ of the gamma distribution is computed from the variance $\sigma^2$ of non-zero precipitation amounts estimated from Pearson's residuals \citep{mccullagh_generalized_1989} as:

\begin{equation}
\sigma^2 = \frac{1}{\left\{N_q- \left( K_q +1 \right) \right\}} \sum_{i=1}^{N_q} \frac{\left( q_i - \mu \right)^2}{\mu^2},
\label{eq:sigma_gamma}
\end{equation}

where $N_q$ is the number of non-zero precipitation data $q_i$ considered in the analysis. As the shape parameter $\nu$ equals the inverse of the variance $1/\sigma^2$, the gamma distribution $F_Q$ modeling the precipitation amount thus follows a gamma distribution of this type $\Gamma \left(\nu, \alpha = \mu / \nu\right)$.

For any given prediction day, the estimation of both GLM models practically proceeds as follows:

\begin{itemize}
\item The precipitation state (wet or dry), the precipitation amount and the value of the different potential predictors are extracted for the $N_d$ nearest analogs of the day. The precipitation state of a given day is considered to be wet if the precipitation amount for this day is higher or equal to 0.1 mm. It is described with a binary precipitation occurrence variable $\mathbf{O}$, set to 1 for the wet case, 0 for the dry case.

\item For occurrence probability, different sets of predictors are considered in turn. For each set, the parameters of the occurrence GLM are estimated from the predictors/occurrence values available for the $N_d$ analogs.

\item For precipitation amount, different sets of predictors are again considered in turn. For each set, the parameters of the GLM are estimated from the predictors/amount values available from the analog days which are wet ($N_q$, the number of days considered here for the regression, is therefore smaller or equal to $N_d$ and varies a priori from one target day to another).

\end{itemize}

For the considered prediction day, the different sets of predictors considered in turn are built from the four potential predictors identified in the preliminary work (cf. section \ref{sec:data}). For occurrence probability (resp. precipitation amount), the four potential predictors actually allow to build fifteen different sets of predictors, further denoted as "regressive structures" in the following (cf list in Table \ref{tab:regressiv_structure}). For each regressive structure, the regression coefficients of corresponding GLMs are estimated using the Iterative Re-weighted Least Squares algorithm \citep[IRLS,][]{nelder_generalized_1972}. The prediction skills of the different regressive structures are then compared and the regressive structure (predictor set) which minimizes the Bayesian Information Criterion is

retained for the prediction \citep{schwarz_estimating_1978,akaike_new_1974} (only the regressive structures for which all coefficients are significant at a 5% level are compared; the significance is estimated with the $Z$-test (resp. the Student $t$-test)).

The prediction of the occurrence probability (resp. the expected precipitation amount) for the prediction day is finally obtained from the best occurrence (resp. amount) GLM, using the values of the predictors observed for that prediction day. The final distribution of precipitation $F_Y$ is obtained by combining the issued occurrence probability $\pi$ and the amount distribution $F_Q$ according to Eq. \ref{eq:cdf_precip}.

\subsection{The Analog Model as benchmark and backup prediction model}
\label{sec:def_backup}

The $N_d$-nearest analog days identified with the AM can also be directly used, without further regression stage, for a probabilistic prediction. In the following, we also consider predictions obtained with the 25 nearest analog days (for the AM considered here, 25 was found to give the best prediction skill for France by \citet{chardon_spatial_2014}). In this case, the precipitation cdf for the prediction day is simply the empirical distribution of the precipitation values observed for these 25 analogs. The predictions obtained with this analog model, further called AM$_\text{25}$, are used as a benchmark to assess the prediction skill of the hybrid analog/regression approach. In addition, they were used as backup prediction for days for which the regression stage failed in the hybrid approach. One can actually face the situation where no GLM satisfies the significance conditions required for the regression coefficients. This can occur for precipitation occurrence probability, for non-zero precipitation amount or for both predictands simultaneously. In such cases, AM$_\text{25}$ is applied as backup prediction model.

If the significance conditions cannot be satisfied for the precipitation occurrence GLM, the occurrence probability $\pi$ is set to that obtained with AM$_\text{25}$. It thus simply corresponds to the empirical probability $\pi_{\text{AM$_\text{25}$}}$ of precipitation occurrence derived from the 25 analog days of AM$_\text{25}$ as:
\begin{equation}
\pi \equiv \pi_{\text{AM$_\text{25}$}} = \frac{1}{25} \sum_{i=1}^{25} o_i.
\label{eq:pi_AM}
\end{equation}

Similarly, if the significance conditions cannot be satisfied for the precipitation amount GLM, the distribution $F_Q$ is estimated with the empirical distribution $F_{Q,\text{AM$_\text{25}$}}$ derived with AM$_\text{25}$ as:
\begin{equation}
F_Q(q) \equiv F_{Q,\text{AM$_\text{25}$}}(q) = \frac{F_{\text{AM$_\text{25}$}}(q) - \left( 1 - \pi_{\text{AM$_\text{25}$}} \right)}{\pi_{\text{AM$_\text{25}$}}},
\label{eq:FQ_AM}
\end{equation}
where $F_{\text{AM$_\text{25}$}}$ corresponds to the empirical cdf estimated from all precipitations (null and positive) related to the 25 analog days. Note also that if the number $N_q$ of humid analog days is low ($N_q < 10$), the estimation of a GLM is not expected to be robust. When this case appears, $F_Q$ is also set to the cdf obtained with AM$_\text{25}$.

As illustrated in Fig. \ref{fig:description_assembling_cases}, four prediction cases are thus achieved with the hybrid approach. They correspond respectively to cases where AM$_\text{25}$ is used to backup the prediction of the whole precipitation distribution (case 1), where AM$_\text{25}$ is applied to backup the amount cdf prediction (case 2), where AM$_\text{25}$ is used to backup the occurrence probability prediction (case 3) and where the regression stage could be activated for both occurrence and amount (case 4).

Note that the regression stage achieved with GLMs can also be seen as a way to refine the estimation of the cdf that could have been obtained directly with the backup (and benchmark) AM$_\text{25}$ analog model. The refinement leads to update the occurrence probability and/or the cdf of non-zero precipitation amount.

As described previously, the two-stage analog/regression prediction process is repeated for each prediction day in turn. As the analog days vary from one prediction day to another, the predictors selected in the regression stage and the value of the corresponding regression coefficients are expected to vary from one prediction day to the other. The hybrid model SCAMP allows thus for a day-to-day adaptive and tailored downscaling.

```
\begin{figure*}[t]
\includegraphics[width = 8.3cm]{fig02.pdf}
\caption{Illustrations of the four cases met for the issue of $F_Y(y)$ by the hybrid model. Case 1: None of the occurrence and amount GLMs could be retained during the regression stage: AM$_\text{25}$ is used to predict the whole precipitation distribution. Case 2: Only the occurrence GLM could be retained. It gives the estimated occurrence probability. The distribution of non-zero precipitation comes from AM$_\text{25}$. Case 3: Only the amount GLM could be retained. It gives the distribution of non-zero precipitation. The occurrence probability is the empirical occurrence probability from AM$_\text{25}$. Case 4: Both occurrence and amount GLMs could be estimated: they respectively give the occurrence probability and the distribution of non-zero precipitation, to be further combined for the full distribution of precipitation.}
\label{fig:description_assembling_cases}
\end{figure*}
```

```
\subsection{Model evaluation}
\label{sec:evaluation}
```

The prediction skill of the downscaling model is assessed with probabilistic scores usually used to evaluate Ensemble Prediction Systems (EPS). Let us consider a given EPS, denoted as $\mathcal{P}$.

The Brier Score \citep{brier_verification_1950,murphy_new_1973} first evaluates the ability of EPS $\mathcal{P}$ to predict precipitation occurrence. When estimated over $M$ prediction days, the mean Brier Score $\overline{\mathrm{BS}}$ reads:
```
\begin{equation}
\overline{\mathrm{BS}} = \frac{1}{M} \sum_{i=1}^{M} \left[ p_i - o_i \right] ^2,
\label{eq:def_BS}
\end{equation}
```
where, for a given prediction day $i$, $p_{i}$ is the occurrence probability issued by EPS $\mathcal{P}$ and $o_i$ is the effective precipitation occurrence for this day ($o_i=1$ for a wet day, $=0$ otherwise).

The ability of EPS $\mathcal{P}$ to estimate the precipitation amount is evaluated with the Continuous Ranked Probability Score \citep[CRPS,][]{brown_admissible_1974,matheson_scoring_1976}. When estimated over $M$ prediction days, the mean CRPS reads:
```
\begin{equation}
\mathrm{\overline{CRPS}} = \frac{1}{M} \sum_{i=1}^{M}\int_{-\infty}^{+\infty}{ \left[ F_{i}(x)-H_{y_{i}}(x) \right] }^2 dx,
\label{eq:def_CRPS}
\end{equation}
```
where, for a given prediction day $i$, $H_{y_{i}}$ and $F_i$ denote respectively the cdf of the observation $y_{i}$ and the cdf derived from EPS $\mathcal{P}$. $x$ denotes the predictand quantiles of the cdfs. Note that $H_{y_{i}}$ corresponds to the Heaviside function where $H_{y_{i}} = 1$ if $x \ge y_{i}$ and $H_{y_{i}}=0$ otherwise.

[revised manuscript text omitted]
\textsuperscript{o}1, n\textsuperscript{o}6 ,n\textsuperscript{o}8 and n\textsuperscript{o}13 including the other predictors). Str. n\textsuperscript{o}9, n\textsuperscript{o}12, n\textsuperscript{o}14 and n\textsuperscript{o}15 are almost never selected.

\begin{figure*}[t]
\includegraphics[width = 12cm]{fig08.pdf}
\caption{Prediction of occurrence probability: selection frequencies (\%) of the 15 regression structures and of the backup model AM$_\text{25}$. Predictors involved are indicated in graphs headers and index of the regressive structure in top left corners. Gray grids: same as in Fig. \ref{fig:freq_cas}. The selection frequency of AM$_\text{25}$ corresponds to the sum of those obtained for cases 1 and 3 in Fig. \ref{fig:freq_cas}.}
\label{fig:global_selection_occ}
\end{figure*}

\begin{figure*}[t]
\includegraphics[width = 12cm]{fig09.pdf}
\caption{Same as Fig. \ref{fig:global_selection_occ} for the probabilistic prediction of precipitation amount. The selection frequency of AM$_\text{25}$ corresponds to the sum of those obtained for cases 1 and 2 in Fig. \ref{fig:freq_cas}.}
\label{fig:global_selection_qu}
\end{figure*}

Note that for the selection of the best regression structure for a given prediction day, all these 15 regressive structures have been in turn tested. The results above suggest that this systematic test is not necessary and that it could be reasonable to consider only the few structures which are frequently retained or which are retained a "reasonable" fraction of the days. However, the selection frequency of a given structure actually varies with the seasons and/or the encountered synoptic situation and some secondary regressive structures can be retained frequently for specific situations. This is illustrated in Fig. \ref{fig:adapt_temp_404_2321} for a cell located in north-western France. The figure presents how the selection frequency of each regression structure differs in different seasons and weather patterns \citep[WP, defined in Table \ref{tab:WP_definition}, ][]{garavaglia_introducing_2010} from the selection frequency obtained for the all-days situation.

For precipitation occurrence (Fig. \ref{fig:adapt_temp_404_2321}a), the selection of the main regressive structures (i.e. Str. n\textsuperscript{o}1 and n\textsuperscript{o}7 respectively based on $R_{700}$ and $R_{700} + Occ-1$) is up to 15 \% more frequent (resp. less frequent) for WP3 (resp. WP5) compared to the alldays situation. For precipitation amount (Fig. \ref{fig:adapt_temp_404_2321}b), the selection frequency of the main regressive structures (Str. n\textsuperscript{o}1 and n\textsuperscript{o}3 based on $R_{700}$ and $W_{700}$ respectively) can similarly change up to +/-10\%. The reduced selection of a main regressive structure for a given season or WP can lead to preferentially retain some secondary regressive structure. For instance, the regressive Str. n\textsuperscript{o}8 based on $W_{700} + H$ is selected 10 \% more frequently for WP2 than for the all-days situation (Fig. \ref{fig:adapt_temp_404_2321}b).

\begin{figure*}[t]
\includegraphics[width = 12cm]{fig10.pdf}
\caption{For each season and weather type, difference (\%) in selection frequency with the all-days case for different regression structures. Results for the prediction of (a) occurrence and (b) amount. A positive difference indicates that the considered regressive structure is selected more often than for the all-day situation. Results are displayed for a grid cell located in the north-west of France. For a clearer illustration, the three or four regressive structures that are almost never selected are not displayed.}
\label{fig:adapt_temp_404_2321}
\end{figure*}

The preferential (or conversely reduced) selection of some regression structures for given WTs or seasons was estimated for all grid cells of France. In most cases, the preferential (or reduced) selection was found to present a noticeable spatial coherency. Different configurations are observed as illustrated in Figure \ref{fig:adapt_spatiale} and discussed below.

The preferential selection of some regression structures can first be observed over large to very large regions. As an example, the preferential selection of Str. n\textsuperscript{o}3 for the prediction of precipitation amount for days in WP7 (more than +15 \% compared to usual) is obtained for all grid cells in France. Whatever the location, the vertical velocity $W_{700}$ seems thus required in this specific weather pattern. Another example is that of WP8 which corresponds to an Anticyclonic situation. Whatever the location, no precipitation is really expected for this configuration. No predictor is thus required in addition to geopotential heights used in the analog stage. This configuration logically leads to a large preferential selection of the backup AM$_\text{25}$ model.

For a given weather pattern, the preferential selection of a regressive structure can also vary from one region to the other. For WP2 for instance, the structures based on $W_{700}$ or on $W_{700}$ and $H$ are selected much more often along the Atlantic coast and in the north of France. The backup AM$_\text{25}$ model is conversely more selected in the South-East, in the Mediterranean coast especially. For this weather regime, the South-east is actually protected by the Massif Central mountain and does thus usually not receive precipitation \citep[cf. Fig. 3 of ][]{garavaglia_introducing_2010}.

The preferential selection of a regressive structure can be also obtained for rather small and specific regions. In Fig. \ref{fig:adapt_spatiale}b, the regressive Str. n\textsuperscript{o}8 based on $W_{700}+H$ is more frequently selected for WP7 (around +15 \%) in the Cevennes-Vivarais regions (south-eastern part of the Massif Central) and in the pre-alpine mountains (western part of the Alps). The combination of $W_{700}$ and $H$ seems thus to be very informative in those configurations for this really rare WP (4 \% of the 20-year period).

Whatever the configuration, the preferential selection of regression structures presents some spatial coherency, at small or large regional scales. This obviously also suggests the spatial robustness of the informative predictors to be retained for given large scale weather configurations.

\begin{figure*}[t]
\includegraphics[width = 12cm]{fig11.pdf}
\caption{(a) Mean geopotential height at 1000 hPa for three WPs \citep{garavaglia_introducing_2010}. (b) For each WP, difference (\%) in selection frequency with the all-days case. Results for two regression structures ($W_{700}$ and $W_{700}+H$) and for AM$_\text{25}$. Predictand is precipitation amount. A positive (resp. negative) difference indicates an extra-selection (resp. reduced selection). Gray grids: same as in Fig. \ref{fig:freq_cas}.}
\label{fig:adapt_spatiale}
\end{figure*}

\conclusions %% \conclusions[modified heading if necessary]
\label{sec:conclusion}

[revised manuscript text omitted]

\end{acknowledgements}

{\clearpage}
\begin{appendix}[A]
%\appendixtitle{Acronyms}
%------------------------------------------------------------------------------------------------------
\begin{table*}[t]
        \begin{tabular}{rl}
        AM & Analog Model\\
        AM$ \text{25}$ & Analog Model (based on the 25 nearest atmospheric analogs) used as benchmark or backup prediction model. \\

```
        BS & Brier Score\\
        BSS & Brier Skill Score\\
        cdf & cumulative distribution function\\
        CRPS & Continuous Ranked Probability Score \\
        CRPSS & Continuous Ranked Probability Skill Score \\
        EPS & Ensemble Prediction System\\
        GCM & General Circulation Model\\
        GLM & Generalized Linear Model\\
        SAFRAN & 8x8km precipitation reanalysis for France from MeteoFrance\\
        SCAMP & two-stage analog / regression model\\
        SDM & Statistical Downscaling Model      \\
        TWS & Teweless-Wobus Score\\
        WP & Weather Pattern \\
        \end{tabular}
        \label{tab:Acronyms}
\end{table*}
\end{appendix}

%% REFERENCES

%% Since the Copernicus LaTeX package includes the BibTeX style file copernicus.bst,
%% authors experienced with BibTeX only have to include the following two lines:

\bibliographystyle{copernicus}
\bibliography{references}

%% Tables
\clearpage
\begin{table*}[t]
        \caption{Large-scale potential variables considered in the work. Stars: predictors obtained from the best
GLMs identified for the 12 test SAFRAN grid cells (section \ref{sec:data_model_evaluation}). Double stars:
predictors used for the analog stage. Bold text: predictors retained for the SCAMP version presented and
evaluated in this work. See \cite{Holton_introduction_2012} for the definition of the variables.}
        \begin{tabular}{rl}
                \tophline
                Acronym & Predictor description\\
                \middlehline
                $R_{850}$ & *Relative humidity at 850 hPa \\
                $R_{700}$ & *\textbf{Relative humidity at 700 hPa} \\
                $R_{500}$ & Relative humidity at 500 hPa \\
                $TCW$ & Total Column Water\\
                $R_{850}TCW$ & *Product of $R_{850}$ and $TCW$ \\
                $T_{700}$ & *\textbf{Air temperature at 700 hPa} \\
                $B_{700}$ & *Baroclinity at 700 hPa \\
                $\Delta_z$ & 700hPa - 1000hPa thickness of the air column\\
                $Z_{1000}$ & **\textbf{Geopotential height at 1000 hPa} \\
                $Z_{700}$ & Geopotential height at 700 hPa \\
                $Z_{500}$ & **\textbf{Geopotential height at 500 hPa} \\
                $F_{700}$ & Wind speed at 700 hPa \\
                $U_{700}$ & West component of wind speed at 700 hPa \\
                $V_{700}$ & South component of wind speed at 700 hPa \\
                $W_{700}$ & *\textbf{Vertical velocity (vertical component of wind speed) at 700 hPa} \\
                $H$ & *\textbf{Helicity of horizontal wind integrated from 1000 to 500 hPa} \\
                $PV_{400}$ & *Potential vorticity of the atmosphere at 400 hPa \\
                $\Delta \theta$ & *Potential temperature gradient between 925 and 700 hPa \\
                $FR_{700}$ & *Humidity flux at 700 hPa \\
                $FU_{700}$ & West component of humidity flux at 700 hPa \\
                $FV_{700}$ & South component of humidity flux at 700 hPa \\
                $\nabla FR_{700}$ & *Divergence of $FR_{700}$ \\
                $Occ-1$ & *\textbf{Precipitation occurrence of the day before the prediction day}\\\
```

\clearpage
\begin{table*}[t]
    \caption{Possible regressive structures (i.e. combination of predictors) for the modeling of precipitation occurrence and amount.}
    \begin{tabular}{cll}
        \tophline
        Structure index & Precipitation occurrence & Precipitation amount \\
        \middlehline
        Str. n\textsuperscript{o}1 & $R_{700}$ & $R_{700}$ \\
        Str. n\textsuperscript{o}2 & $H$ & $H$\\
        Str. n\textsuperscript{o}3 & $W_{700}$ & $W_{700}$\\
        Str. n\textsuperscript{o}4 & \textit{Occ-1} & $T_{700}$ \\
        Str. n\textsuperscript{o}5 & $R_{700} + H$ & $R_{700} + H$ \\
        Str. n\textsuperscript{o}6 & $R_{700} + W_{700}$ & $R_{700} + W_{700}$\\
        Str. n\textsuperscript{o}7 & $R_{700} + $\textit{Occ-1} & $R_{700} + T_{700}$\\
        Str. n\textsuperscript{o}8 & $H + W_{700}$ & $H + W_{700}$\\
        Str. n\textsuperscript{o}9 & $H + $\textit{Occ-1} & $H + T_{700}$\\
        Str. n\textsuperscript{o}10 & $W_{700} + $\textit{Occ-1} & $W_{700} + T_{700}$ \\
        Str. n\textsuperscript{o}11 & $R_{700} + H + W_{700}$ & $R_{700} + H + W_{700}$\\
        Str. n\textsuperscript{o}12 & $R_{700} + H + $\textit{Occ-1} & $R_{700} + H + T_{700}$ \\
        Str. n\textsuperscript{o}13 & $R_{700} + W_{700} + $\textit{Occ-1} & $R_{700} + W_{700} + T_{700}$\\
        Str. n\textsuperscript{o}14 & $H + W_{700} + $\textit{Occ-1} & $H + W_{700} + T_{700}$\\
        Str. n\textsuperscript{o}15 & $R_{700} + H + W_{700} + $\textit{Occ-1} & $R_{700} + H + W_{700} + T_{700}$\\
        \bottomhline
    \end{tabular}
    \belowtable{} % Table Footnotes
    \label{tab:regressiv_structure}
\end{table*}

\clearpage
\begin{table*}[t]
    \caption{Names of the weather patterns (WP) defined in \citet{garavaglia_introducing_2010} and related frequency for the 01 August 1982-08-01 to 2001-07-31 period.}
    \begin{tabular}{ccc}
        \tophline
        Index & Denomination & Annual frequency (\%)\\
        \middlehline
        WP1 & Atlantic Wave & 8\\
        WP2 & Steady Oceanic & 22 \\
        WP3 & Southwest Circulation & 8 \\
        WP4 & South Circulation & 17 \\
        WP5 & Northeast Circulation & 6 \\
        WP6 & East Return & 6 \\
        WP7 & Central Depression & 4 \\
        WP8 & Anticyclonic & 29\\
        \bottomhline
    \end{tabular}
    \belowtable{} % Table Footnotes
    \label{tab:WP_definition}

\end{table*}

\end{document}